**Subject Category:**
Biology (whole organism)

behaviour/ecology/environmental science

masking, humpback whale, anthropogenic noise, acoustic communication, active space

**Author for correspondence:**
Rebecca A. Dunlop
e-mail: r.dunlop@uq.edu.au

# The effects of vessel noise on the communication network of humpback whales

## Rebecca A. Dunlop

Cetacean Ecology and Acoustics Laboratory, School of Veterinary Science, University of Queensland, Gatton Campus, Gatton, Queensland 4343, Australia

RAD, 0000-0002-0427-6317

Humpback whales rely on acoustic communication to mediate social interactions. The distance to which these social signals propagate from the signaller defines its communication space, and therefore communication network (number of potential receivers). As humpback whales migrate along populated coastlines, they are likely to encounter noise from vessel traffic which will mask their social signals. Since no empirical data exist on baleen whale hearing, the consequences of this are usually assumed, being the modelled reduction in their communication space. Here, the communication space and network of migrating humpback whales was compared in increasing wind-dominated and vessel-dominated noise. Behavioural data on their social interactions were then used to inform these models. In typical wind noise, a signaller's communication space was estimated to extend to 4 km, which agreed with the maximum separation distance between groups that socially interacted. An increase in vessel noise reduced the modelled communication area, along with a significant reduction in group social interactions, probably due to a reduction in their communication network. However, signal masking did not fully explain this change in social behaviour, implying there was also an additional effect of the physical presence of the vessel on signaller and receiver behaviour. Though these observed changes in communication space and social behaviour were likely to be short term and localized, an increase in vessel activity due to tourism and coastal population growth may cause more sustained changes along the humpback whale migration paths.

## 1. Introduction

Vocal communication networks involve a signaller, one or more intended receivers and one or more 'unintended receivers' such as 'eavesdroppers' [1–4]. The size of this communication network is usually defined as the distance radius from the signaller at which any other receiver can detect and interpret the signal. This

depends on the signal level and structure, the propagation environment, levels of background noise and the receiver's ability to detect, recognize and discriminate the signal within noise [5]. In the marine environment, as sound propagates over large distances, cetacean communication networks can be extensive and involve multiple intended and unintended receivers. Cetaceans are likely to use these networks within feeding (e.g. [6,7]) and breeding interactions (e.g. [8,9]), meaning effective communication exchanges are essential for these life functions.

In the underwater environment, there are several natural sources of noise, such as noise from surface breaking waves during periods of high wind. Noise sources can also be anthropogenic in origin, such as from vessel activity, oil and gas exploration, naval sonar activity or construction. These noise sources have the potential to mask signals, meaning they become inaudible to receivers. This will reduce the size of the communication network and, in turn, may have detrimental effects on feeding or breeding interactions within these networks. Important data for predicting the potential impacts of masking include, for the receiver, basic audiograms, critical ratios (CRs) of hearing and identification of mechanisms that allow receivers some release from masking [10]. For the signaller, important data for predicting masking include determining which anti-masking strategies are used and how successful these strategies are [10]. Signalling whales, for example, increase their signal level [11,12], change their signalling behaviour [13], frequency content [14–16] and duration of their signal [17] in increasing noise. It is likely, however, that there is a limit to the success of these strategies, especially if the noise source is anthropogenic in origin and noise levels are above natural noise levels. Therefore, in anthropogenic noise, the receiver's ability to detect and interpret sounds may be reduced over and above that in natural noise, with a higher risk of loss of communication space and reduction in communication network.

Acoustic communication in humpback whales (*Megaptera novaeangliae*) has been extensively studied, making this a good model species for cetacean masking studies. The communication space of social sounds ('vocalizations', blow-hole associated sounds and percussive signals generated at the surface) is in the order of kilometres [18] in a shallow-water environment (less than 50 m). This space includes a network of likely intended and unintended receivers [8,9]. Their repertoire of humpback whale social vocalizations is diverse and spans a fundamental frequency range of less than 40 Hz to over 2 kHz. Sound types range from low-frequency 'grumbles' to high-frequency bird-like 'chirps' and include modulated (e.g. 'moans', 'trumpets' and 'cries'), amplitude-modulated (e.g. 'purrs' and 'growls') and broadband (e.g. 'roars' and 'underwater blows') sound types [19]. These social sounds are commonly heard during breeding [8,9,20,21] and feeding group social interactions [22–25]. They probably function to provide information on group membership, changes in group membership (social information), as well as signaller sex, size, location [26] and motivation [27]. Previous studies found that the communication space of humpback whale signallers extends to approximately 4 km [18]. Signallers maintained this space in increased wind-dominated noise by switching from vocal sounds to surface-generated sounds [13] and increasing their vocal source level [11]. Interestingly, in response to increased vessel-dominated noise, there was no evidence that the signaller used either of these anti-masking strategies [28]. This implies that when vessels are traversing the area, the communication space of a humpback whale signaller is significantly reduced. The consequences of this within the context of breeding interactions, if any, remain unknown.

In baleen whales, apart from anatomical modelling, there are no empirical data on their hearing in noise. Therefore, if predicting the impact of masking on the receiver, the signal propagation is usually modelled in various noise conditions and the distance at which the signal disappears below noise is assumed to equate to the receiver's audible limits. In other words, assumptions of receiver hearing in noise, and/or the receivers's ability to hear (e.g. [29]) or recognize (e.g. [30,31]) the signal must be made. To do this, these studies assume the distance at which the measured received level of a signal-to-noise ratio, or signal excess, becomes a certain value (e.g. 0 dB) delimits the signaller's communication distance. This study will follow the approach outlined above to define the communication space of signalling humpback whales but will use behavioural observations of social interactions to further inform results. Here, communication space will be defined as distance that conspecifics can potentially receive acoustic signals (modelled communication space) and where there is an observable (social) response in the receiving whale (from observational data). To test for effects of anthropogenic noise on this defined space, first, the likely extent of a signalling whale's communication space will be modelled and compared in natural (wind) and anthropogenic (vessel) noise. Second, the study will assess the behavioural consequences of increased vessel noise within the communication network, in terms of changes in the number of available receivers and changes in acoustically mediated social interactions. By combining signal-to-noise measurements at a receiver, with observed group social behaviour within a local communication network, a more informed representation of their important communication space is presented which does not rely on assumptions of receiver hearing.

# 2. Material and methods

## 2.1. Visual and acoustic data collection

Acoustic recordings were made of groups of migrating (from their breeding ground in the Great Barrier Reef, towards their Antarctic feeding grounds) humpback whales (September/October of 2002–2004 and 2008). The study site was Peregian Beach, 150 km north of Brisbane, on the east coast of Australia (26°S, 153°E), where the majority of humpback whale groups migrate close to the shoreline (within 10 km). The array comprised five hydrophone-buoy systems anchored in 20–28 m of water in a 'T' configuration. Buoys 1–3 were anchored in a line 1.5 km offshore, with buoys 4 and 5 extended a further 1 km offshore. Each system included a High Tech HTI-96-MIN hydrophone with built-in +40 dB pre-amplifier, a pre-amplifier (+20 dB) and VHF radio transmitter. Real-time radio transmissions from the buoys were received at a base station using a Yagi antenna attached to a four channel, low noise, VHF receiver (type 8101) and a Winradio receiver. Sound source locations were calculated by *Ishmael* where the mean position of several estimates was calculated over a brief period. A shore-based theodolite survey of the surface buoys was used to obtain accurate positions. Sounds were then attributed to migrating groups (see below). For detailed validation experiments, see Noad *et al.* [32].

Simultaneous land-based tracking of the groups occurred to obtain visual data of migrating humpback whale groups within a 10 km offshore radius from the acoustic array. All groups were tracked regardless of whether or not they were vocalizing. Group positions were recorded using a theodolite linked to *Cyclopes* (a tracking program developed by Eric Kniest, University of Newcastle, Australia) each time the group surfaced. Additional observations on group social composition (number of adults within the group and whether or not there was a calf) and social behaviour (splitting of animals from, and joining of animals to, the group) were recorded at each position using binoculars. The positions of traversing vessels were also recorded by the land-based observation team using the theodolite, meaning vessels were tracked as they moved through the study site. Visual and acoustic positions were then overlaid onto the one Cyclopes file so that the tracking could be matched to groups being visually tracked. Using this overlaid file, the distance of each signalling group to each hydrophone-buoy receiver was then measured for each recorded social sound. Vocalizing groups were recorded out to approximately 5 km from the array in all directions, in water depths of between 20 and 35 m. However, these signalling groups were continuously visually tracked when in visual range (10 km offshore radius from the array), meaning the visual track was longer than the acoustic track (up to approx. 3 h).

## 2.2. Sound measurements

The following analysis was limited to daylight hours, given that is when the visual observations were taking place. Spectrograms of acoustic recordings were produced and viewed using Raven 1.2 (Cornell Lab of Ornothology) with the FFT size set at 4096 samples, Hamming window and 80% overlap. Group recordings started when they first became audible on the array (determined by manual inspection of the spectrogram) and ended when they were no longer audible as the group migrated out of the area (between 20 and 80 min). Social sounds ($n = 1268$ vocalizations) were measured from 31 groups (15 in wind noise and 16 in vessel noise). Each social sound was measured using three of the five hydrophones. These hydrophones were selected to maximize the range of distances from the signalling group at which the same social sound was recorded, while still having enough signal above noise to allow measurement (i.e. at least some of the signal was visible in the spectrogram). This meant the same sound could be measured at an increasing range from the receiver to better quantify the loss in signal energy with increasing distance.

Each social sound was then isolated from three different channels and saved as a separate file along with a sample of background noise either just before or just after the sound. This sample included background singing only if this sound source was also present within the social sound sample. If a sound was not visible on one of the spectrograms, only two channels were used in the analysis.

Estimates of received and source level were made using a custom-made Matlab script [33]. From the wavefile, an FFT was used to calculate the frequency content of the sound (FFT size of 16 384, 1.35 Hz resolution, 50% overlap) and its power spectrum extracted. The power spectrum was calibrated to account for the sensitivity of the hydrophone and the gain of the system. The full recording chain was calibrated post-field. All sounds measured were made within the dynamic limits of the system.

Hydrophone sensitivity was −164 dB *re* 1 V μPa$^{-1}$, which included +40 dB gain for the built-in pre-amplifier. Full system sensitivity varied by 1.5 dB over the frequency range 40–10 000 Hz.

Transmission loss (TL) was measured at the site as described in [34]. A boat and playback of octave band-limited white noise were used as the sources. As the bottom type was variable (sandy with patches of rock), the boat conducted runs along lines radiating to and from the array, from distances of 100 m out to about 10 km. The boat speed was kept constant. Wind speed for that day ranged from 10 to 15 knots, swell was less than 1 m and there were no significant currents in the area. This gave various regression lines of received levels as a function of distance from the source

$$\mathrm{TL} = a + b\log(x),\tag{2.1}$$

where $b$ is the slope of the regression line, $x$ is the distance (m) and $a$ is a constant (which was frequency-dependent). The value of $a$ may have varied with the direction of the boat (approaching or going away from the receivers) as the noise radiated forward probably differed from the noise radiated aft. Even so, for most frequencies, $b$ varied with distance but could be well approximated by two values; one applying to distances less than, and the other greater than, a cross-over value. Absolute values of TL were determined by measuring received levels of octave band-limited noise, projected using a J11 source suspended from a boat (estimating the source level using a hydrophone suspended at a distance of 3 m and corrected to 1 m assuming spherical spreading). This source was projected at three distances (between 200 and 1000 m) from the array. The trend in loss, $b\log(x)$, from the boat runs was then fitted to the absolute values of loss from the J11 measurements to determine the value of $a$ for each octave band, by minimizing the sum of the squares of the differences between $a + b\log(x)$ and the data-points from the J11 measurements. Using equation (2.1), the values were then calculated for each octave band. The signal was then reconstructed and filtered from 35 to 5623 Hz to eliminate low-frequency turbulence noise and high-frequency system noise.

Broadband background noise levels during this time (NL$_{bb}$: actual band 36 Hz–2.8 kHz) were estimated using 10 min files isolated from each hydrophone-buoy. As recordings were divided into 10 min files for storage, the file containing the social sound of interest was used for the background noise measurement. Wind-dominated noise was measured when there were no audible and/or visually tracked vessels in the area (including shipping vessels in the within-sight shipping channel). A previous study at this site found the measured noise, in the absence of singer and vessels, was similar to that observed for wind-dependent noise in Australian waters [35]. In addition, this study found wind speed (ranging from less than 5 to 15 knots) to be significantly correlated with noise levels in all 1/3 octave frequency bands [13], suggesting the dominant noise source was due to wind. Wind-dominated noise levels at the hydrophones could be considered to be similar to levels at the vocalizing whales. These levels varied from approximately 90 to 112 dB *re* 1 μPa in wind speeds ranging from less than 5 knots to approximately 15 knots. The median wind-dominated noise was 100 dB *re* 1 μPa for this area at a wind speed of 12 knots. The vessel noise environment was due to passing fishing vessels which were usually audible on the array for 10–40 min depending on the vessel's trajectory, distance from the array, speed and size. Measured noise levels were above 100 dB *re* 1 μPa with mean and modal measured levels being 105 dB *re* 1 μPa (vessels were approx. 4 km from the array). Given the distance of the vessel, and therefore measured noise, varied across the three hydrophones, the hydrophone used to measure the signal was also the hydrophone used to measure the noise. It should be noted that vessel noise may not have dominated the noise environment for the entire recording as vessel noise would have faded in and faded out as they traversed the area.

The 1/3 octave band containing the most energy was used as the array received level (RL$_{rms}$) and noise levels were then used to calculate array signal-to-noise levels (rSNR) as per Dunlop *et al.* [34]. From the received level of each sound (RL$_{rms}$ over 35–5623 Hz), the source level was estimated as

$$\mathrm{SL}_{rms} = \mathrm{RL}_{rms} + \mathrm{TL}.\tag{2.2}$$

## 2.3. Modelled communication space

First, signals were grouped into 'low frequency' or 'high frequency' (separated by frequency content) using the results of the classification and regression tree (CART) analysis presented in Dunlop [18]. To summarize, low-frequency sounds generally had a centre frequency below 158 Hz and a minimum frequency below 126 Hz, with high-frequency signals above these values.

The communication space has been previously statistically modelled for wind-dominated [18] and vessel-dominated [36] noise. Further details on the statistical modelling procedure and model outputs

are presented in the electronic supplementary material, Methods (electronic supplementary material, figures S1, S2 and table S1). In brief, a two-dimensional smooth surface was fitted to the interaction between background noise and separation distance between the signaller and array. The response variable was the received SNR at the array (termed rSNR). This gave a series of trend lines representing the relationship between the distance of the signalling group from the receiver, broadband noise levels, and measured rSNR for low- and high-frequency vocal sounds. A separate analysis was carried out for groups in wind-dominated noise and groups in vessel-dominated noise and in both models. It was assumed that an rSNR below 0 would equate to reduced ability of receiver humpbacks to discriminate and/or recognize the signal, meaning, for this study, rSNR = 0 was assumed to delimit the signaller communication space. To compare the size of the communication space in wind-dominated noise with vessel-dominated noise, statistical model outputs for the 0 dB rSNR trend lines were selected. Raw data-points were also selected and plotted to indicate the variation around the displayed trend line.

Further statistical models were run to assess the variation in rSNRs at the receiver. Here, signals from groups between 900 and 1100 m from the receiver were measured to limit the data to signals produced from groups approximately 1 km from the receiver ($n = 14$ and $n = 6$ groups for low- and high-frequency models, respectively). Response data (measured rSNRs) were normally distributed. A generalized linear mixed model framework was used in 'R' software with the 'lmer' package for model fitting. Predictor variables were broadband noise with the interaction effect of dominant noise source (wind or vessel). GroupID was included as a random effect as multiple signals were measured from the same group.

## 2.4. Behavioural analysis

The social environment was first quantified and compared between groups measured in wind-dominated noise, with those measured in vessel-dominated noise, to ensure the social environments were comparable (see electronic supplementary material, Methods). For every 10 min of the recording, the mean distance of each neighbour, being the nearest neighbour, second nearest neighbour, and so on, for every group within a 5 km radius of the signalling group was measured. This gave between two and eight measured time periods per group, depending on the length of the recording, and a range of between zero and six neighbours per time period. Social behaviour was then quantified for each neighbour according to whether (1) or not (0) it interacted with the signalling group, as outlined in the electronic supplementary material. An interaction was defined as being joined by, or joining, another group, or another group changing direction and approaching to within approximately 200 m of the signalling group. To determine if groups were less likely to interact with each other in increasing vessel noise compared with wind noise, the response variable (neighbour interaction) was statistically modelled using a generalized linear model, assuming a binomial regression distribution, and including the interaction effects of neighbour distance, noise source and noise level.

## 2.5. Communication area and receiver group distribution models

The communication area (circular area around the signaller) was calculated using the radius from the signaller at which the signal level above noise was equal to 0 dB. This was the most simplistic representation of communication area and likely to be used in the absence of information on propagation loss. Four randomly distributed receiver groups (maximum number observed within this defined communication area) were placed in the area. The communication space for low-frequency signals was then reduced according to the communication space results for vessel-dominated noise (loss of radial distance), and the number of groups left within the area counted. This simulation was repeated 30 times (approximate signaller group sample size) to give an estimated mean (with standard deviation) of the percentage loss of receiver groups due to the reduced area. The same random group distribution of four groups within the area was then used to simulate (30 times) the percentage loss in receiver groups for the high-frequency signal communication space.

Next, the known (taken from the social environment data) percentage loss in the number of receiver groups was determined for each signaller group (per 10 min) and reported as a mean and standard deviation for all signallers. Wind and vessel noise data were combined, given there was no significant difference in receiver group distribution between the two noise environments (electronic supplementary material, Methods).

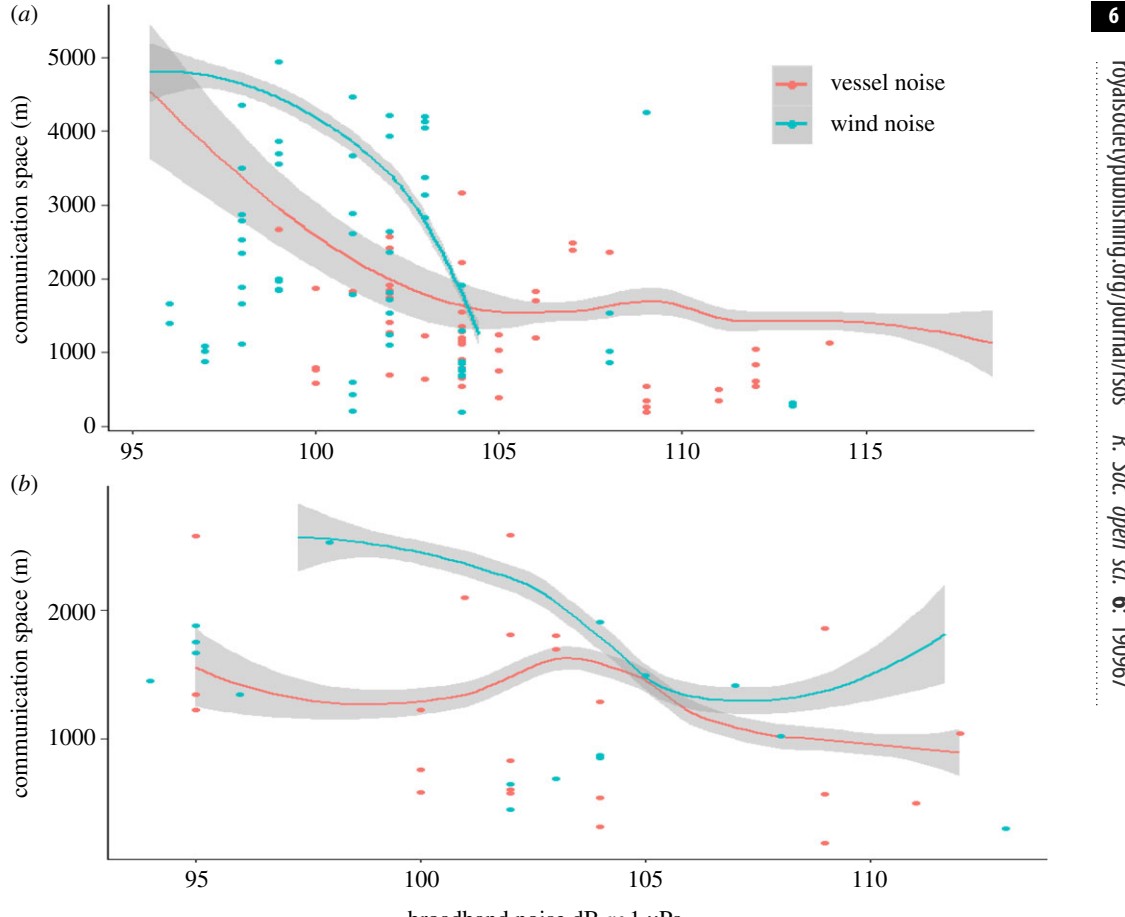

**Figure 1.** The modelled communication space of humpback whale (*a*) low- and (*b*) high-frequency signals in wind-dominated and vessel-dominated noise. Smoothing functions are from the generalized estimate equation outputs (electronic supplementary material, table S1) for the rSNR = 0 relationship (from electronic supplementary material, figures S1 and S2). Raw data-points at which the signal rSNR at the array was between 1 and −1 are also included.

## 3. Results

### 3.1. Social environment

From the visual observations, the majority of signalling groups were observed to have between two and four groups within 5 km ($n = 21$). In both noise environments, the mean distance of nearest neighbour was 2 km, with first and third quartile ranges of 1–3 km. The second nearest neighbour mean distance was approximately 3 km, and the third, approximately 4 km, from the vocalizing group. Signalling groups most commonly interacted with their nearest neighbour (16 out of 19 interactions), and then may have further interacted with their second neighbour ($n = 1$) or third neighbour ($n = 2$).

### 3.2. Modelled communication space

The modelled communication space for low-frequency vocalizations was estimated to be 4 km in received wind-dominated noise levels of 100 dB *re* 1 μPa wind noise, reducing to 3 km in the same received level of vessel-dominated noise (figure 1*a*). These noise levels equated to typical wind conditions and relatively low received levels of vessel noise. The maximum interaction distance of groups was a separation of 4 km, where a neighbour turned to head towards a signalling group from this distance to eventually join together ($n = 2$). This separation distance matched the likely extent of the communication space for low-frequency sounds in typical wind-dominated noise. The size of the communication network for migrating humpback whales, in this shallow-water environment (less than 50 m), was therefore likely to be between two and four other groups based on the distribution of groups from the signaller.

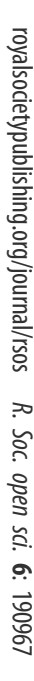

**Figure 2.** The modelled probability of the signalling group interacting with a neighbouring group as a function of the distance of the neighbour from the signaller. Relationships are shown for the wind and vessel noise environments and include 95% confidence intervals (*a*) and separated according to the neighbour position relative to the signalling group in (*b*) wind and (*c*) vessel noise.

For high-frequency vocalizations, the modelled communication space was 2.5 km in typical wind-dominated noise, given they were lower in level (mean of 149 dB *re* 1 µPa @ 1 m compared with 156 dB *re* 1 µPa @ 1 m in low-frequency sounds). This reduced the number of receiving groups within the network to one or two. Groups, in wind-dominated noise, had a +50% chance of interacting if separated by less than 2 km (figure 2) suggesting this distance was likely to be important in terms of acoustically mediated group social behaviour (joining together). In relatively low received levels of vessel-dominated noise (100 dB *re* 1 µPa), the modelled communication space of high-frequency vocalizations reduced to 1.5 km (figure 1*b*).

## 3.3. Effect of increased noise

As received noise levels increased to 105 dB *re* 1 µPa (mean received vessel-dominated noise and above average wind-dominated noise), the modelled communication space decreased to approximately 2 km and

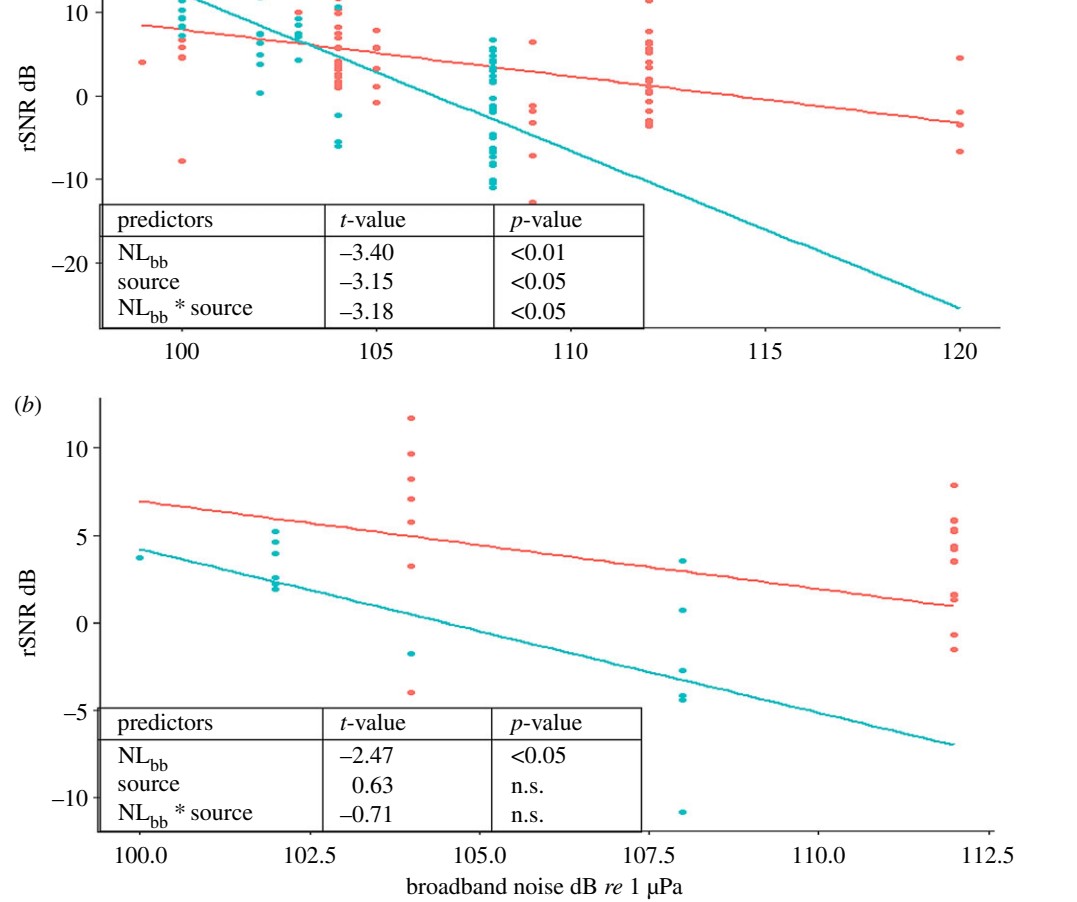

**Figure 3.** The modelled relationship, and significance of the predictor variables, of the rSNR of humpback whale (*a*) low- and (*b*) high-frequency signals in wind and vessel-dominated noise received noise. Data-points from signalling whales between 900 and 1100 m (approx. 1 km) from the array are included.

1 km for low- and high-frequency vocalizations, respectively (figure 2*a*), meaning the network would probably be reduced to one receiver. By received noise levels of 110 dB *re* 1 µPa, which is considered to be high wind noise, or noise from a close-by (within 2 km) vessel, the modelled communication space for both signal types was reduced to only 1 km. This space may or may not have included one receiver group, depending on the distance of the nearest neighbour from the signaller. However, although the reduction in modelled communication space was the same for both noise sources, a neighbour was significantly less likely to interact with the signalling group when there was a vessel in the area ($t = 2.19$, $p = 0.03$; figure 2*a*). In wind-dominated noise, signalling groups had an 80% chance of interacting with another group if it came within 1 km, reducing to a 30% chance if a vessel was present (figure 2). In other words, there was a reduction in the probability that a signalling group would join with its nearest neighbour in the presence of a vessel (figure 2*b*,*c*).

One explanation for this reduction in interaction behaviour would be increased signal masking. However, noise level was not a significant predictor in that increased noise did not explain any reduction in the likelihood of groups to interact. Even in vessel-dominated noise, many received signals would have been above noise at this close-by separation distance (figure 1), suggesting masking was not a significant issue. If using 105 *re* 1 µPa as a generic (average) received vessel-dominated noise level, though the modelled communication space of both vocalization types was significantly reduced, it was still beyond 1 km (figure 1). Using signals at which the rSNR = 0 does not include any measure of variance, in that, signals received by a whale at 1 km are likely to range in rSNRs. If choosing all signals emitted from groups positioned at 1 km from the receiver, the rSNR of low-frequency signals, in received vessel noise of 105 *re* 1 µPa, was up to +13 dB (figure 3*a*), and for high-frequency signals, up to

**Table 1.** The modelled communication area based on the distance from the signaller at which the rSNR = 0. The group numbers within the wind and vessel noise communication areas were estimated assuming a random distribution (of four groups per 50 km$^2$) and known distribution, using visual data of groups around signalling groups. Standard deviations are in parentheses.

| signal type | distance (km) | area (km$^2$) | random distribution | | known distribution | |
|---|---|---|---|---|---|---|
| | | | no. groups | % reduction | no. groups | % reduction |
| low-f | 4 | 50 | 4 | | 2.2 (1.0) | |
| | 2 | 50 | 0.9 (0.9) | 77 (22) | 1.0 (0.8) | 47 (36) |
| high-f | 2 | 12 | 0.9 (0.8) | | 1.0 (0.8) | |
| | 1 | 12 | 0.3 (0.5) | 78 (35) | 0.5 (0.6) | 33 (44) |

+12 dB (figure 3b). In other words, levels of many of the signals at the receiver would have been well above noise. Further, the modelled decrease in the rSNR with increasing measured noise was significantly less in vessel-dominated noise compared with wind-dominated noise (table 1 and figure 3b), in that low-frequency signals were less masked by vessel noise compared with wind noise, as long as the receiver was within 1 km of the signaller. Therefore, at close group-to-group separation distances, the observed reduction in interaction behaviour in vessel noise cannot be explained by modelled signal masking.

## 3.4. Communication area and group distribution

To determine the loss of available receiver groups within the communication area in vessel-dominated noise, first a random distribution of groups within that area (four groups per 50 km$^2$) was assumed using a simple spherical communication area with the signalling group at the centre. In vessel-dominated noise, this equated to an approximately 77–78% reduction in available receivers within the network (table 1) for both signal types. However, the actual distribution of groups at this study site suggests a bias towards the signalling group, where the density of groups (available receivers) was higher closer to the signalling group (figure 1). Using this known distribution, the reduction in available receivers due to loss in communication space was estimated as 47% and 33% for the low- and high-frequency signal communication area, respectively (table 1). Behavioural data of observed social interactions found that, in wind-dominated noise, the signalling group had a 70% chance of interacting with the nearest neighbour if the separation distance was less than 2 km, reducing by 50% in vessel-dominated noise. Therefore, this reduction in interaction behaviour was greater than that predicted by using known group distribution as a proxy for the likelihood of social interactions, but less than using a random distribution of groups within the communication space.

## 4. Discussion

This goal of this study was to acoustically and behaviourally define the communication space and network of signalling humpback whales and determine if, and how, this was reduced during periods of vessel-dominated noise. Here, modelled communication space was defined as distance that conspecifics could potentially send and receive acoustic signals (assuming a signal excess of 0 dB delimited this distance). However, the significance of this study lies in the use of observable (social) responses (from behavioural data) in the receiving whale groups to further inform these acoustic models and assumptions. First, the communication space and network was defined in wind-dominated (natural) noise. Acoustic results suggest that humpback whale low-frequency social signals had a modelled communication space of approximately 4 km with a communication network of two to four other groups. Behavioural data found that humpback whale receivers responded to, and eventually joined, a signalling group from a maximum separation distance of 4 km, suggesting low-frequency sounds may function to mediate longer-range interactions between networks of dispersed groups. However, sighted behavioural interactions were most likely to occur within 2 km, which agreed with the modelled communication space of higher frequency signals. Higher frequency signals were of a lower amplitude compared with low-frequency signals and may be used to mediate closer-range joining interactions. By matching the predictions of communication space from the acoustic data, with observed social behaviour, fewer assumptions on receiver hearing in noise are required

when assessing the extent of a signaller's communication space. This provides a much more accurate tool with which to fully assess the consequences of increased anthropogenic noise.

Next, to test for effects of anthropogenic noise, the likely extent of a signalling whale's communication space was compared in natural (wind) and anthropogenic (vessel) noise. In average vessel-dominated noise (105 dB *re* 1 µPa), the modelled communication space of humpback whale low-frequency vocalizations reduced by half to 2 km. These results are similar to modelled loss in communication space of humpback whale social sounds in vessel noise in Cholewiak *et al.* [30]. The communication space of high-frequency sounds was also reduced by half in vessel-dominated noise, to only 1 km. This translates to a reduction of four times the communication area, assuming a circular area from the signaller for both signal types. In theory, this will reduce the number of available receivers and therefore the number of observed social interactions. To assess this, many studies do not have access to data on receiver group distribution and social behaviour within the communication area. Rather, a random distribution of receiver groups around the signaller is assumed. Here, assuming a random distribution, the number of available receivers reduced from four to one. However, there are two problems with this assumption. First, observational data found that receiving groups were not randomly distributed from the signaller, and second, behavioural data found that signallers were highly likely to interact with the closest receiver. After accounting for these two problems, the reduction in available receivers within the communication space was reduced by 47% and 33% for low- and high-frequency signals, respectively. Observed social interactions in vessel-dominated noise, however, found a 50% reduction in the likelihood of a signalling group interacting with a receiver group within the 2 km interaction zone. Therefore, using the random distribution of receivers over-estimated the likely consequences of the reduction in communication network, whereas accounting for observed receiver distribution, but not signaller/receiver behaviour, underestimated network consequences. This discrepancy, as explained later, is likely to be due to the physical presence of the vessel.

Though this study focused on the acoustic environment and behaviour of receivers, potential signaller anti-masking strategies were also considered. The estimated communication space of low-frequency signals in wind-dominated noise was maintained at approximately 4 km up to received levels of 100 dB *re* 1 µPa. This was probably due to the Lombard response found in humpback whales, where signallers increased their vocal source level in response to increased wind-dominated noise [11]. In the same received levels of vessel-dominated noise, the communication space was found to be smaller. A previous study found no signaller Lombard response to increasing vessel-dominated noise [28]. Vessel noise levels were estimated at the signalling group (in [28]) and were comparable to those measured at the array in this study. In other words, received levels of vessel noise should have been high enough to cause a Lombard response in signalling whales but did not. One explanation put forward in this study was that whales have released from masking, in that they can focus their attention on the vocalizing conspecific, rather than the vessel noise. This implies the receiver groups, if possessing the ability for masking relief, would continue to behave as if there was no masking noise source in the area. Behavioural results presented here showed this was not the case. Further, the reduction in humpback group interaction behaviour in vessel-dominated noise was not fully explained by signal masking. Behavioural interactions were most likely to occur when groups were separated by 1 km or less where most signals, even high-frequency signals, were still well above received vessel noise and unlikely to be masked. Again, this shows the value of using behavioural data to inform acoustic models.

An alternative explanation for the lack of signaller Lombard response put forward in Dunlop [28] was that humpback whales were emitting signals at a relatively low level to avoid detection, similar to how they behave around unintended receivers [8]. In other words, the lack of Lombard response was an acoustic avoidance response to the presence of the vessel. The physical presence of the vessel has been shown to reduce bottlenose dolphin [37] and humpback whale [38] foraging activity as well as bottlenose dolphin resting and socializing behaviour [37]. Other studies, carried out on this population of humpback whales at this study site, found that humpback whale groups significantly changed their migration speed and direction to avoid a slowly (4 knots) moving vessel towing (silent) air gun arrays [39,40]. Therefore, this body of work suggests that not only do migrating humpback whales physically avoid vessels, but also change their acoustic and social behaviour when vessels are present. As many of these social interactions are likely to be breeding interactions [8,9,26], understanding the population consequences of disturbance, in this case reduced potential breeding opportunities, would be the next step in understanding whale and vessel interactions. The population used for this study is robust in that it is increasing at 11% per year [41]. These behavioural interruptions within the communication network, if localized and rare, are unlikely to have a

significant effect on such a robust population. Given this population is also well studied in terms of population dynamics, a population consequences model would not be difficult to produce once the link between reduced group interactions within the communication network and reduced mating opportunities has been established. These models could be extended to other humpback populations and other baleen whale species, such as the highly endangered right whale (*Eubalaena* spp.), where less is known about their breeding behaviour. For now, the most comprehensive model to estimate the loss of communication space for baleen whales in vessel noise would be to account for signal type, receiver group distribution with some knowledge of the local communication network and any additional (e.g. avoidance) effect of the presence of the vessel.

Along the eastern coast of Australia, and probably other populated coastlines, vessel activity is increasing due to the growth in the tourism industry. For cetaceans occupying a coastal habitat, vessel interactions are becoming more of an issue. Current Australian mitigation measures regarding the interaction of vessels and whales include caution zones (within 300 m of the whale) and exclusion zones (within 100 m). The assumption with these measures is that staying more than 300 m away 'reduces the risk of disturbance to natural behaviours'. The results of this study suggest otherwise. What is clear is that the deleterious effects of vessels on whales are not limited to collisions and increased signal masking but potential changes in vital function behaviours, such as breeding interactions, should also be considered. In summary, mitigating vessel activity from a noise and communication space perspective may not mitigate for behavioural changes within their communication network.

Ethics. The project operated under animal ethics approval from the University of Queensland Animal Ethics Committee, a Queensland Government Environment, and Heritage Protection permit and an Australian Government Access to Biological Resources in a Commonwealth Area for Non-Commercial Purposes permit.

Data accessibility. The article's supporting data are available on the Dryad Digital Repository: https://doi.org/10.5061/dryad.65j00m3 [42]. Wind and vessel noise communication models are uploaded as part of the electronic supplementary material.

Competing interests. There are no competing interests in that funding agencies had no input into experimental design, data collection, data analysis or interpretation or the writing of the paper.

Acknowledgements. The author would like to thank everyone involved in the Humpback Acoustic Research Collaboration (HARC; funded by the US Office of Naval Research and the Australian Antarctic Division), in particular the numerous volunteers who donated their time and energy to this project. I also thank David Paton for his invaluable field expertise and Eric Kniest for his continued support in the development of Cyclopes. The author would particularly like to acknowledge Associate Professor Michael Noad for leading the HARC work (without which, this study would not have been possible) and Dr Douglas Cato for his continued support and mentorship.

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
