## [Reviewer comments · Royal Society Open Science]

Review History

RSOS-190967.R0 (Original submission)

Review form: Reviewer 1

Is the manuscript scientifically sound in its present form?

No

Are the interpretations and conclusions justified by the results?

Yes

Is the language acceptable?

Yes

Do you have any ethical concerns with this paper?

No

Have you any concerns about statistical analyses in this paper?

No

Recommendation?

Accept with minor revision (please list in comments)

Comments to the Author(s)

Please see attached comments (Appendix A).

Review form: Reviewer 2

Is the manuscript scientifically sound in its present form?

Yes

Are the interpretations and conclusions justified by the results?

Yes

Is the language acceptable?

Yes

Do you have any ethical concerns with this paper?

No

Have you any concerns about statistical analyses in this paper?

Yes

Recommendation?

Major revision is needed (please make suggestions in comments)

Comments to the Author(s)

This paper promises to present a worth-while discussion, taking the organism perspective in how increases in ambient noise, driven by vessel noise and wind noise, may change behaviour in vocalising and/or interactions between humpback groups. What it does really is just describe the extent of the communication space based on interactions between groups of whales based on different ambient noise regimes. I can't help but think that more can be made from the data that was collected, both acoustic and observational, which would enlighten the discussion further, and more to what was promised from the title/abstract.

It seems like a lot of material is trying to be drawn from the data presented: that baleen whale hearing capacity might be better determined; that masking might be discussed in context of communication space, and that the implication of masking on behavioural interactions may be determined. More focus given to one of these may help with streamlining the paper and making it more directed. Highlighting the decreased communication space and how that has wider implications in conspecific interaction I think is the best way to go here, and is where the author seems to take it but there needs to be some more definitive statements of the study question and findings. As presented the data and analysis would not answer this question.

The use of visual data in conjunction with acoustic recordings can afford some very interesting results and a strength to the conclusions not possible from acoustics alone. A little more could be drawn out the significance of using these to different types of datasets together. There was a lot left to the reader to follow up on (e.g. social calling/vocalisation types, behavioural

categorisation) or that is not clearly defined (like wind dominated ambient noise) or explained (some of the analysis – see comments).

Missing from the discussion is the implications of taking an individual-centric or communication space approach, and how that might play into understanding of behaviour or mitigation actions. Also it could lead to a discussion on how this approach may lead to different/better management actions than are currently in place, but did not.

The paper presents a lot of material, but it feels piecemeal, and like a lot of different pieces of data forced together currently, with it lacking a main focus or directive that is clearly stated to being about cohesiveness.

The paper would benefit from proof-reading and going over to make it more concise, focusing on the follow on of one paragraph to another and the general message to the reader. It doesn't have an easy flow to the material.

Title: suggest rewording the title. The paper presents discussion from vessel and wind noise. The short title almost works better as a descriptor of the material presented (masking of humpback whale signals) as the masking effect is where this paper focuses and how that effects the space over which whales interact.

Abstract

Reads a little awkwardly, some rewording may be helpful here with the flow.

Introduction:

I find the introduction to lack flow within and between sections. There seems to be too much covered. If the how communication space for social sounds, as modeled, in different soundscape settings (wind or vessel noise) affects social interaction between whales became the clear focus some of the less relevant/irrelevant material could be removed and make the section more concise and easier to follow.

Lines 32-42: Don't think this is relevant to the discussion. That there is an acoustic communication network and this network is defined by the propagation of each call type in ambient noise conditions if the underlying basis of this paper. Then go on to discuss the conditions in ambient noise, and additions to the soundscape (both natural and anthropogenic) that might alter this. A description of sources of noise that comprise the full soundscape may also be worthwhile to the reader too, perhaps describing the differences in natural (geophonic) sources and human-derived (anthrophonic) noise sources.

Again in the second paragraph (lines 45-60) – discuss masking first, and its implications on communication, and then what baleen whales have been noted to do to compensate for increased ambient noise. For this paper, as it references so many previous studies the most succinct way to introduce material, without losing the reader, would be to use these previous works as a jumping off point, i.e we saw compensation mechanisms in increased wind noise and not in vessel noise, knowing this we interpret the reduction of communication space to have implications on socialising behaviours (the focus of the study).

Maybe somewhere explicitly say that you are using communication space as the distance that conspecifics can send and receive acoustic messages, and where there is a observable (social) response in the receiving whale.

Line 64: may be nice to describe briefly under what conditions the propagation distances were calculated. Is there a general description for social sounds (in frequency, length, modulation etc.). Some more details here would likely help the reader.

Lines 82-87: some of this detail may be better in the methods/ is in the methods and doesn't need repeating

Line 89: 'will modelled and compared' should be 'will be modeled and compared to conditions of...'

Materials and Methods

There are some details missing that would help the reader, for example, is there a baseline or 'background' noise level defined. What is average wind speed/noise, and what is the definition of 'wind dominated'. What was the process of comparing the visual and the acoustic data?

Line 106: Should read were received at a base station

Line 111: Were groups only tracked from shore?

Line 113: Be sure it is clear: vocalisations are localised to a position using the array, the calls are then attributed to a group using visual recordings, the group positions are tracked using theodolite. Are all call types used, or just social calls. Are there social and travel calls?

Line 115 onwards: more detail could be given on the theodolite tracking methods.

Line 118: At what interval were these positions taken? Is this in time or space?

Line 120: were the visual and acoustic tracks compared? Were there groups that did not vocalise? Under what conditions may this have happened?

I assume that the comparisons between visual and acoustic are limited to day light hours, but this is not specifically stated. Were the comparisons limited to only social calls when you made them? The points made in Lines 119-121 need to be highlighted and more specifically described. Is the distance between the whales and vessels also calculated?

Line 124: A better description of wind noise would be useful to the reader. Is this over a particular wind speed or is it an increased ambient noise level when vessels are not present.

Line 125: Is there a specific SNR/dB level for 'first becomes audible'. I presume this is derived manually through inspection of spectrograms.

Line 127-129: Unclear

Line 131: Is this distance from the array and within the hydrophones receiving the calls in the array a means to derive propagation distances? Could the visual tracks and distance from the array and the 'first becomes audible/now not audible' be used somehow in discussing the likely communication space under each ambient noise scenario.

Maybe take a sentence or two after line 136 to say what these technical specifications mean in terms of the calls received and how they might affect the reception and interpretation of calls to the reader.

Line 139: more description or reference to social sound types is needed here.

Line 140: How was the sample of background taken/selected.

Line 161: How were the files selected? What is deemed as 'wind-dominated' noise. Is wind dominated purely vessel absent?

Line 165: What is the wind speed at the average wind speed noise.

Line 171: used to measure the signal

Line 175: How does using the 1/3 octave band affect the interpretation compared to, say, using the peak frequency when the SNR was at its greatest.

Line 176, Line 184: make sure referencing is complete, consistent and in line with the journal

Lines 184-195: Much of this seems to be information presented elsewhere. Be sure when describing this that you are, then, adding something of note to the initial analysis. Be sure that that the reader is clear that the rSNR you reference is the SNR of a call heard by a receiver

Line 194-195: Not sure what this means

Line 196: run not ran

Line 201: 'R'

Lines 219: Could you draw on your observations here with the inter-group communications rather than randomly placed groups?

Line 230-234: Why combine the two noise environments here, especially as the behavioural response to these noise additions have previously been seen to be different.

Results:

Social, Line 240-245: Is this from the observational or modelled data?

Lines 256-261 – perhaps describe 'typical' wind and vessel noise and how this overlaps with typical social calls

Lines 385-398: Could it be that the low-frequency communication space/low frequency calls/communications are less affected by wind/vessel noise

Is there any difference seen in the calls used/calling behaviours? Were the previous assertions for call/behavioural modifications due to increased ambient noise confirmed in this study?

Discussion

This discussion again could do with more focus and linking back to the research question. Work on the flow of the material presented. The discussion material should show a stronger link back to the results of this study.

Line 426: Typical wind conditions or typical noise conditions – and what percentage of the time of recording does this represent?

Line 425-430: Even the discussion between high and low frequency communication space – and be sure to tie it back to humpback whale communications.

Line 453-455: Again I think it might be possible to draw more out of the observational data than just the random distribution model.

Line 473-474: Develop this point further

There are a few discussion points that could be raised/discussed further:

- Maybe draw out the differences between wind and vessel noise in this discussion.
- More could be made of the implication of the response to vessel noise and presence specifically on social interactions of humpbacks and what this could mean on an individual and population level, even for a robust population
- What are the implications for such changes in communication space

Decision letter (RSOS-190967.R0)

19-Aug-2019

Dear Dr Dunlop,

The editors assigned to your paper ("Mitigating for the effects of vessels on baleen whale signalling from a communication space perspective may not prevent behavioural changes") have now received comments from reviewers. We would like you to revise your paper in accordance with the referee and Associate Editor suggestions which can be found below (not including confidential reports to the Editor). Please note this decision does not guarantee eventual acceptance.

Please submit a copy of your revised paper before 11-Sep-2019. Please note that the revision deadline will expire at 00.00am on this date. If we do not hear from you within this time then it will be assumed that the paper has been withdrawn. In exceptional circumstances, extensions may be possible if agreed with the Editorial Office in advance. We do not allow multiple rounds of revision so we urge you to make every effort to fully address all of the comments at this stage. If deemed necessary by the Editors, your manuscript will be sent back to one or more of the original reviewers for assessment. If the original reviewers are not available, we may invite new reviewers.

- Data accessibility

<http://datadryad.org/submit?journalID=RSOS&manu=RSOS-190967>

- Competing interests

- Authors' contributions

- Acknowledgements

- Funding statement

on behalf of Dr Ari Friedlaender (Associate Editor) and Kevin Padian (Subject Editor)
openscience@royalsociety.org

Associate Editor's comments (Dr Ari Friedlaender):

To the Authors,
This study attempts to provide a novel means to evaluating differences between modeled and measured sound on baleen whales. The subject matter is of interest broadly and there is a substantial amount of new information presented in the submission. That said, both reviewers (and I) have serious concerns with a number of aspects of the writing that require major revision before it can be evaluated for publication. As you will see from the reviewer comments, there is general interest in the work but there are two main areas that require attention. First, there is a lack of information throughout regarding the acoustic modeling that does not allow the reader to fully understand what was done and if it was appropriate. Second, there needs to be more information provided in the main text that is currently in the supplemental material or simply referenced from previous work. I would consider this an opportunity to expand on the writing to make it more broadly consumable to an audience that is not as intimately familiar with acoustics and the literature.

I am confident that both reviewers have provided constructive comments that can, if accounted for, ultimately make this a fine contribution in RSOS.

Thank you.
Ari S. Friedlaender

Subject Editor Comments to Author:

The reviewers are generally positive about the manuscript but believe that it could be considerably clearer in its presentation. Please address all the comments carefully, and best wishes for your revision.

Reviewers' Comments to Author:
Reviewer: 1

Comments to the Author(s)
Please see attached comments.

Reviewer: 2

Comments to the Author(s)

This paper promises to present a worth-while discussion, taking the organism perspective in how increases in ambient noise, driven by vessel noise and wind noise, may change behaviour in vocalising and/or interactions between humpback groups. What it does really is just describe the extent of the communication space based on interactions between groups of whales based on different ambient noise regimes. I can't help but think that more can be made from the data that was collected, both acoustic and observational, which would enlighten the discussion further, and more to what was promised from the title/abstract.

It seems like a lot of material is trying to be drawn from the data presented: that baleen whale hearing capacity might be better determined; that masking might be discussed in context of communication space, and that the implication of masking on behavioural interactions may be determined. More focus given to one of these may help with streamlining the paper and making it more directed. Highlighting the decreased communication space and how that has wider implications in conspecific interaction I think is the best way to go here, and is where the author seems to take it but there needs to be some more definitive statements of the study question and findings. As presented the data and analysis would not answer this question.

The use of visual data in conjunction with acoustic recordings can afford some very interesting results and a strength to the conclusions not possible from acoustics alone. A little more could be drawn out the significance of using these to different types of datasets together. There was a lot left to the reader to follow up on (e.g. social calling/vocalisation types, behavioural categorisation) or that is not clearly defined (like wind dominated ambient noise) or explained (some of the analysis - see comments).

Missing from the discussion is the implications of taking an individual-centric or communication space approach, and how that might play into understanding of behaviour or mitigation actions. Also it could lead to a discussion on how this approach may lead to different/better management actions than are currently in place, but did not.

The paper presents a lot of material, but it feels piecemeal, and like a lot of different pieces of data forced together currently, with it lacking a main focus or directive that is clearly stated to being about cohesiveness.

The paper would benefit from proof-reading and going over to make it more concise, focusing on the follow on of one paragraph to another and the general message to the reader. It doesn't have an easy flow to the material.

Title: suggest rewording the title. The paper presents discussion from vessel and wind noise. The short title almost works better as a descriptor of the material presented (masking of humpback whale signals) as the masking effect is where this paper focuses and how that effects the space over which whales interact.

Abstract

Reads a little awkwardly, some rewording may be helpful here with the flow.

Introduction:

I find the introduction to lack flow within and between sections. There seems to be too much covered. If the how communication space for social sounds, as modeled, in different soundscape settings (wind or vessel noise) affects social interaction between whales became the clear focus

some of the less relevant/irrelevant material could be removed and make the section more concise and easier to follow.

Lines 32-42: Don't think this is relevant to the discussion. That there is an acoustic communication network and this network is defined by the propagation of each call type in ambient noise conditions if the underlying basis of this paper. Then go on to discuss the conditions in ambient noise, and additions to the soundscape (both natural and anthropogenic) that might alter this. A description of sources of noise that comprise the full soundscape may also be worthwhile to the reader too, perhaps describing the differences in natural (geophonic) sources and human-derived (anthrophonic) noise sources.

Again in the second paragraph (lines 45-60) – discuss masking first, and its implications on communication, and then what baleen whales have been noted to do to compensate for increased ambient noise. For this paper, as it references so many previous studies the most succinct way to introduce material, without losing the reader, would be to use these previous works as a jumping off point, i.e we saw compensation mechanisms in increased wind noise and not in vessel noise, knowing this we interpret the reduction of communication space to have implications on socialising behaviours (the focus of the study).

Maybe somewhere explicitly say that you are using communication space as the distance that conspecifics can send and receive acoustic messages, and where there is a observable (social) response in the receiving whale.

Line 64: may be nice to describe briefly under what conditions the propagation distances were calculated. Is there a general description for social sounds (in frequency, length, modulation etc.). Some more details here would likely help the reader.

Lines 82-87: some of this detail may be better in the methods/ is in the methods and doesn't need repeating

Line 89: 'will modelled and compared' should be 'will be modeled and compared to conditions of...'

Materials and Methods

There are some details missing that would help the reader, for example, is there a baseline or 'background' noise level defined. What is average wind speed/noise, and what is the definition of 'wind dominated'. What was the process of comparing the visual and the acoustic data?

Line 106: Should read were received at a base station

Line 111: Were groups only tracked from shore?

Line 113: Be sure it is clear: vocalisations are localised to a position using the array, the calls are then attributed to a group using visual recordings, the group positions are tracked using theodolite. Are all call types used, or just social calls. Are there social and travel calls?

Line 115 onwards: more detail could be given on the theodolite tracking methods.

Line 118: At what interval were these positions taken? Is this in time or space?

Line 120: were the visual and acoustic tracks compared? Were there groups that did not vocalise? Under what conditions may this have happened?

I assume that the comparisons between visual and acoustic are limited to day light hours, but this is not specifically stated. Were the comparisons limited to only social calls when you made them? The points made in Lines 119-121 need to be highlighted and more specifically described. Is the distance between the whales and vessels also calculated?

Line 124: A better description of wind noise would be useful to the reader. Is this over a particular wind speed or is it an increased ambient noise level when vessels are not present.

Line 125: Is there a specific SNR/dB level for 'first becomes audible'. I presume this is derived manually through inspection of spectrograms.

Line 127-129: Unclear

Line 131: Is this distance from the array and within the hydrophones receiving the calls in the array a means to derive propagation distances? Could the visual tracks and distance from the array and the 'first becomes audible/now not audible' be used somehow in discussing the likely communication space under each ambient noise scenario.

Maybe take a sentence or two after line 136 to say what these technical specifications mean in terms of the calls received and how they might affect the reception and interpretation of calls to the reader.

Line 139: more description or reference to social sound types is needed here.

Line 140: How was the sample of background taken/selected.

Line 161: How were the files selected? What is deemed as 'wind-dominated' noise. Is wind dominated purely vessel absent?

Line 165: What is the wind speed at the average wind speed noise.

Line 171: used to measure the signal

Line 175: How does using the 1/3 octave band affect the interpretation compared to, say, using the peak frequency when the SNR was at its greatest.

Line 176, Line 184: make sure referencing is complete, consistent and in line with the journal

Lines 184-195: Much of this seems to be information presented elsewhere. Be sure when describing this that you are, then, adding something of note to the initial analysis. Be sure that that the reader is clear that the rSNR you reference is the SNR of a call heard by a receiver

Line 194-195: Not sure what this means

Line 196: run not ran

Line 201: 'R'

Lines 219: Could you draw on your observations here with the inter-group communications rather than randomly placed groups?

Line 230-234: Why combine the two noise environments here, especially as the behavioural response to these noise additions have previously been seen to be different.

Results:

Social, Line 240-245: Is this from the observational or modelled data?

Lines 256-261 – perhaps describe ‘typical’ wind and vessel noise and how this overlaps with typical social calls

Lines 385-398: Could it be that the low-frequency communication space/low frequency calls/communications are less affected by wind/vessel noise

Is there any difference seen in the calls used/calling behaviours? Were the previous assertions for call/behavioural modifications due to increased ambient noise confirmed in this study?

Discussion

This discussion again could do with more focus and linking back to the research question. Work on the flow of the material presented. The discussion material should show a stronger link back to the results of this study.

Line 426: Typical wind conditions or typical noise conditions – and what percentage of the time of recording does this represent?

Line 425-430: Even the discussion between high and low frequency communication space – and be sure to tie it back to humpback whale communications.

Line 453-455: Again I think it might be possible to draw more out of the observational data than just the random distribution model.

Line 473-474: Develop this point further

There are a few discussion points that could be raised/discussed further:

- Maybe draw out the differences between wind and vessel noise in this discussion.
- More could be made of the implication of the response to vessel noise and presence specifically on social interactions of humpbacks and what this could mean on an individual and population level, even for a robust population
- What are the implications for such changes in communication space

Author's Response to Decision Letter for (RSOS-190967.R0)

See Appendix B.

RSOS-190967.R1 (Revision)

Review form: Reviewer 1

Is the manuscript scientifically sound in its present form?

Yes

Are the interpretations and conclusions justified by the results?

Yes

Is the language acceptable?

Yes

Do you have any ethical concerns with this paper?

No

Have you any concerns about statistical analyses in this paper?

No

Recommendation?

Accept with minor revision (please list in comments)

Comments to the Author(s)

The revision of this paper addressed a majority of the concerns I had previously. I have noted just a few editorial items and 1-2 comments in my attached pdf with notations (Appendix C). This is a nicely done and well times paper. I would have selected Accept As Is, but I think addressing the few minor points will put a polish on the final version.

Review form: Reviewer 2

Is the manuscript scientifically sound in its present form?

Yes

Are the interpretations and conclusions justified by the results?

Yes

Is the language acceptable?

Yes

Do you have any ethical concerns with this paper?

No

Have you any concerns about statistical analyses in this paper?

No

Recommendation?

Accept with minor revision (please list in comments)

Comments to the Author(s)

The paper is much improved. However, I still feel that there could be a little more care in leading the reader and being concise.

Introduction

There is information particularly in the introduction that I do not think adds to the overall message of the piece. In my opinion the 'unintended' receivers is given emphasis, rather than simply stated communications may also be received by eavesdropping whales. I think it detracts

slightly from the message of interacting groups having 2km threshold and is not (rightly) brought up again in the discussion. I suggest the author consider editing.

Methods

A little more on the study site might be helpful. Where was the array, what is the typical ambient noise setting (if not using wind dominated to represent this), what is the topography and setting of the array (to give an idea of propagation pathways) etc.

Some clarification needed to make sure details are consistent. In talking about neighbouring groups you say measurements are between 0 and 6 groups in line 263, and then between 4 groups (max observed) in line 277 used in the model, and then the majority of interactions being between 2 and 4 groups in line 297.

Results

Are there any indications in the surface observations or the acoustic recordings that the groups are in fact communicating/interacting with each other that could be described. Are there calls known to be used within groups and between groups, or changes in calling behaviour seen in different ambient conditions (other than that stated from previous studies).

Discussion

If the physical presence of vessels is influential is there anything in the data that might suggest that this influence is increased as vessels approach etc. Is there anything from the observational data that further supports the statement, or that of the whales avoiding vessels? Were difference behaviours seen compared to addition of vessel number, proximity, type – this could be helpful for management.

What is the typical composition of vessel noise – could you predict this would affect high frequency calls more? Is there more that could be discussed in terms of changes in calling in terms of call type, rate, structure rather than just the Lombard effect?

Somewhere a statement of where this work adds to all the previous studies drawn from is needed to clarify to the reader what knowledge is being added, rather than just assimilating from the other works. Similarly, I think a few sign-post statements of aims, findings and conclusions etc. could aid the reader. Highlight that interactions and their strengths are influenced by several factors, one is the presence of other groups within the communication space and the extent of this, but that presence of vessels/stressors generally may be more influential in whether there are interactions or not.

Line 54 – little is known, however, about hearing capabilities

Line 62 – remove a comma after levels

Line 64 – remove relative

Line 65 – is there a reference here

Line 77-79 – make sure this reads smoothly

Line 99 – would this not be equal to or exceeding 0 dB

Line 136-148- maybe a few more details about the array could help the reader picture it, and the spacings between each of the hydrophones or perhaps a figure? Also a description of the propagating area might also be helpful – flat sandy bottom, rocky etc.

Line 163- I am wondering whether more of your observations could come through in the study rather than just the binary results – for example did the whales move to maintain closer proximities in higher ambient noise, different responses in group travel directions, was the response different with calves present in groups etc.

Line 182- italicise a

Line 184 –italicise b

Line 190-192 – italicise formulae were needed

Line 186 – is there a threshold, and was it first audible or visible as a signal on the spectrogram
 Line 220-221- make sure this makes sense
 Line 245 – plotted or displayed, don't think you need both
 Line 271 – and high frequency signals?
 Lines 328-331 –is this from the observation data – please state
 Line 338 – does not read smoothly
 Line 342-343 – is this taken from the wind-dominated observations
 Line 356 – were there any changes in call usage or call structure in noise. Were low frequency signals favoured if masking potential is lower?
 Line 343 – 344 – does not read smoothly
 Line 358 – I think it should still be clear that these results come from the model and not the observations
 Line 361,384 – 502km should be 50km²

Figure 2 – is this modeled or real data?

Decision letter (RSOS-190967.R1)

04-Oct-2019

Dear Dr Dunlop:

On behalf of the Editors, I am pleased to inform you that your Manuscript RSOS-190967.R1 entitled "The effects of vessel noise on the communication network of humpback whales" has been accepted for publication in Royal Society Open Science subject to minor revision in accordance with the referee suggestions. Please find the referees' comments at the end of this email.

The reviewers and Subject Editor have recommended publication, but also suggest some minor revisions to your manuscript. Therefore, I invite you to respond to the comments and revise your manuscript.

- Ethics statement

- Data accessibility

<http://datadryad.org/submit?journalID=RSOS&manu=RSOS-190967.R1>

- **Competing interests**

- **Authors' contributions**

- **Acknowledgements**

- **Funding statement**

Because the schedule for publication is very tight, it is a condition of publication that you submit the revised version of your manuscript before 13-Oct-2019. Please note that the revision deadline will expire at 00.00am on this date. If you do not think you will be able to meet this date please let me know immediately.

on behalf of Dr Ari Friedlaender (Associate Editor) and Kevin Padian (Subject Editor)
openscience@royalsociety.org

Associate Editor Comments to Author (Dr Ari Friedlaender):

Associate Editor: 1

Comments to the Author:

To the Authors,

Both reviewers and I believe that you have done a very nice job in your revised submission. Nearly all of the previous comments and suggestions have been addressed. The only remaining comment of substance relates to adding some text in the discussion that relates to how this new information is different and/or builds upon previous work. At present it is not completely clear and I believe that this can be done relatively easily. Otherwise, there are a few minor edits/comments that I believe are straight forward to address. I laud you on this work and enjoyed reading your study.

Thanks you.
Ari S. Friedlaender

Reviewer comments to Author:

Reviewer: 2

Comments to the Author(s)

The paper is much improved. However, I still feel that there could be a little more care in leading the reader and being concise.

Introduction

There is information particularly in the introduction that I do not think adds to the overall message of the piece. In my opinion the 'unintended' receivers is given emphasis, rather than simply stated communications may also be received by eavesdropping whales. I think it detracts slightly from the message of interacting groups having 2km threshold and is not (rightly) brought up again in the discussion. I suggest the author consider editing.

Methods

A little more on the study site might be helpful. Where was the array, what is the typical ambient noise setting (if not using wind dominated to represent this), what is the topography and setting of the array (to give an idea of propagation pathways) etc.

Some clarification needed to make sure details are consistent. In talking about neighbouring groups you say measurements are between 0 and 6 groups in line 263, and then between 4 groups (max observed) in line 277 used in the model, and then the majority of interactions being between 2 and 4 groups in line 297.

Results

Are there any indications in the surface observations or the acoustic recordings that the groups are in fact communicating/interacting with each other that could be described. Are there calls known to be used within groups and between groups, or changes in calling behaviour seen in different ambient conditions (other than that stated from previous studies).

Discussion

If the physical presence of vessels is influential is there anything in the data that might suggest that this influence is increased as vessels approach etc. Is there anything from the observational data that further supports the statement, or that of the whales avoiding vessels? Were difference behaviours seen compared to addition of vessel number, proximity, type - this could be helpful for management.

What is the typical composition of vessel noise - could you predict this would affect high frequency calls more? Is there more that could be discussed in terms of changes in calling in terms of call type, rate, structure rather than just the Lombard effect?

Somewhere a statement of where this work adds to all the previous studies drawn from is needed to clarify to the reader what knowledge is being added, rather than just assimilating from the other works. Similarly, I think a few sign-post statements of aims, findings and conclusions etc. could aid the reader. Highlight that interactions and their strengths are influenced by several factors, one is the presence of other groups within the communication space and the extent of this, but that presence of vessels/stressors generally may be more influential in whether there are interactions or not.

Line 54 - little is known, however, about hearing capabilities

Line 62 - remove a comma after levels

Line 64 - remove relative

Line 65 - is there a reference here

Line 77-79 - make sure this reads smoothly

Line 99 – would this not be equal to or exceeding 0 dB

Line 136-148- maybe a few more details about the array could help the reader picture it, and the spacings between each of the hydrophones or perhaps a figure? Also a description of the propagating area might also be helpful – flat sandy bottom, rocky etc.

Line 163- I am wondering whether more of your observations could come through in the study rather than just the binary results – for example did the whales move to maintain closer proximities in higher ambient noise, different responses in group travel directions, was the response different with calves present in groups etc.

Line 182- italicise a

Line 184 –italicise b

Line 190-192 – italicise formulae were needed

Line 186 – is there a threshold, and was it first audible or visible as a signal on the spectrogram

Line 220-221- make sure this makes sense

Line 245 – plotted or displayed, don't think you need both

Line 271 – and high frequency signals?

Lines 328-331 –is this from the observation data – please state

Line 338 – does not read smoothly

Line 342-343 – is this taken from the wind-dominated observations

Line 356 – were there any changes in call usage or call structure in noise. Were low frequency signals favoured if masking potential is lower?

Line 343 – 344 – does not read smoothly

Line 358 – I think it should still be clear that these results come from the model and not the observations

Line 361,384 – 502km should be 50km²

Figure 2 – is this modeled or real data?

Reviewer: 1

Comments to the Author(s)

The revision of this paper addressed a majority of the concerns I had previously. I have noted just a few editorial items and 1-2 comments in my attached pdf with notations. This is a nicely done and well times paper. I would have selected Accept As Is, but I think addressing the few minor points will put a polish on the final version.

Author's Response to Decision Letter for (RSOS-190967.R1)

See Appendices D & E.

Decision letter (RSOS-190967.R2)

24-Oct-2019

Dear Dr Dunlop,

I am pleased to inform you that your manuscript entitled "The effects of vessel noise on the

communication network of humpback whales" is now accepted for publication in Royal Society Open Science.

Kind regards,

on behalf of Dr Ari Friedlaender (Associate Editor) and Professor Kevin Padian (Subject Editor)
openscience@royalsociety.org

Follow Royal Society Publishing on Twitter: [@RSocPublishing](https://twitter.com/RSocPublishing)
Follow Royal Society Publishing on Facebook:
<https://www.facebook.com/RoyalSocietyPublishing.FanPage/>
Read Royal Society Publishing's blog: <https://blogs.royalsociety.org/publishing/>

Appendix A

Mitigating for the effect of vessels on baleen whale signaling from a communication space perspective may not prevent behavioral changes

Author: Rebecca Dunlop

Review

Summary:

This paper filled a much needed gap between modeling the effects of noise and measuring the effects of noise. This was a well-done combination of integrating communication space modelling with behavioral analysis. The modelling guided the assessment of specific targeted behavioral interactions, and the knowledge of the interactions allowed for verification and application of the modelling results. For this aspect alone, the paper will be a value to the published literature. A second highlight of the results of this work was the examination of the assumption that a 0 dB SNR limit on defining communication space is an accurate assumption. Many papers and modelling efforts make this assumption, but this is the first time I've seen it actually tested with behavior. Well done, and a valuable contribution.

To make the leap from the current manuscript to a publishing version will require some revision. There are a few places in the current manuscript that require more detail to be fully understood. There are many cases where the Supplemental material is referenced, as well as previously published Dunlop papers (e.g. Dunlop et al 2013, 2016, 2018), where a summary or summary figure would help provide enough info to the reader of this paper to continue on without having to go back and read other papers to understand this one. These items are pointed out in the detailed comments below. Second, the word model and modelling is used throughout the paper to talk about acoustic modelling, communication space modelling, and statistical modelling. Some care to better differentiate between the difference types of modelling with qualifiers or descriptor would be helpful.

One technical item that requires addressing concerns the acoustic modelling. What model was used for TL? I'm assuming spherical spreading because a circular communication space is referenced to in many places of the paper. The acoustic model was not explicitly stated. Use of a range dependent model would be more applicable in this case (if spherical spreading was indeed used), as range dependent bathymetry is crucial and would not produce circular communication spaces. I'm not advocating for a reworking of the data with a new model, but the details of what was used and its assumptions and limitations should be addressed.

Detailed comments:

Title – I found the title wording and not straightforward. Suggest rewording. Possible suggestion “Is mitigating for the effects of vessel noise on baleen whale communication effective?”

Line 15 – Suggest “coastlines, they”

Line 16 – Suggest “Since there are no empirical data”, since data are plural.

Lines 22-23 – The hearing information is not the emphasis or focus of the analysis. Suggest removing it from the abstract.

Line 26 – Suggest “additional confounding effect of”

Line 53 – Duplicate “to compensate”.

Line 83 – Suggest “condition, and the distance”

Line 89 – Suggest “whales’ or whale’s”

Line 93 – Can you clarify whether the receiver in this line of text refers to the intended receiver, unintended receivers, or both? The assumptions made for each may be different for the signaler and should be addressed.

Lines 111-121 – Positioning of individual vessel transits were discussed in this paragraph and linked to their contribution of vessel noise. How have you dealt with long distance shipping coming in from distant, unobserved ships? This would be a component of the low frequency noise more than the high frequency component. Was this included in the “Wind Only” measurements and modelling? Does “Wind Only” just really mean the absence of close transiting vessels?

Line 127 – What is “These” referring to? The social sounds detected on only 3 of 5 hydrophones, or did you select the best arrivals from 3 of 5 hydrophones?

Line 140 – Where did the background noise measurement come from? 1-sec preceding the call? At another time related to the call? Did it include other background song?

Lines 140-141 – Confused on what this sentence is trying to convey? Please clarify.

Line 144 – Please clarify the use of DFT and FFT in the same sentence.

Line 149 – Not enough information is given about the TL modelling here. A summary from Dunlop et al (2013) would be helpful. What TL model was used?

Lines 160-161 – When did these 10-min files occur in relation to the signals? On the same day, within the same hour?

Line 171 – Suggest “hydrophone used to measure”

Line 176 – What is Dunlop (31)? It would be helpful to see a distribution or histogram for the 1/3 octave bands to identify the dominant band.

Line 181 – It would be helpful to summarize the info related to frequency category definitions from Dunlop (2018). At least provide the frequency range of low and high categories used in this paper.

Line 185 – Which models are you referring to? Statistical, acoustic, propagation, overall communication space? This should be clarified for many uses of “model” or “modelling” throughout the text.

Line 202 – Do you broadband noise or background ambient ocean sound?

Line 220 – More information is needed on why you are assuming a circular area around the signaler? Was this because you used spherical spreading? If so, the assumption of a circular area is most likely not a realistic assumption given the bathymetry, and this should be explicitly stated.

Line 222 – Suggest “randomly distributed receiver groups” if that is what is intended.

Line 224 – Missing the link here between the communication model results and the number of groups left in the area. Is this related to behavioral observations?

Lines 232-234 – A figure showing the 2 groups for comparison would be helpful.

Line 381 – The word compared implies a direct comparison, but there is no comparison in this sentence. Suggest “The loss of total communication area in vessel noise and wind-dominated noise was approximately 75% for both signal types.”

Line 395 – Can you please verify “by” or “to”?

Line 443 – circular area from signaler is again an assumption that warrants clarification.

Appendix B

Mitigating for the effect of vessels on baleen whale signaling from a communication space perspective may not prevent behavioral changes

Letter of response

Associate editor

First, there is a lack of information throughout regarding the acoustic modeling that does not allow the reader to fully understand what was done and if it was appropriate.

Response: this has been addressed in response to reviewer 1 (see below). Note a site-specific empirical measures were used to generate the acoustic model and more details are now included (though noting journal space limitations we attempted to balance the detail included with referring to this in previously published works).

Second, there needs to be more information provided in the main text that is currently in the supplemental material or simply referenced from previous work. I would consider this an opportunity to expand on the writing to make it more broadly consumable to an audience that is not as intimately familiar with acoustics and the literature.

Response: The paper has been further focussed towards a goal that relates to an audience interested in animal communication networks and the effects of noise. This following statement has been added to both the intro and discussion:

This goal of this study was to define the communication space and network of signalling humpback whales and determine if, and how, this was reduced during periods of vessel noise.

The introduction has been edited following comments by reviewer 2. Here, more is included about the relationship between communication space, network, and masking effects. This puts the work into the perspective of animal communication (communication space and networks) as well as the effects of noise on this aspect of communication. This was further re-iterated in the last paragraph of the introduction to illustrate how the study informs current knowledge on animal communication networks and changes in these networks due to noise (using humpback whales as the target species).

The methods were expanded as per reviewer 2 to further explain concepts to an audience not familiar with acoustics. To follow this through the results, more references are made back to communication networks and the implications these results have on the size of the communication network of humpback whales.

Finally, in the discussion, this goal has been reiterated and applicable results discussed. The discussion has been revised and re-ordered. Any previous work referenced has had additional sentences added to further explain results to non-acoustic readers.

To address the material presented in supplementary. This is the background modelling used to generate the outputs for the 0 SNR relationship between noise and distance. Including this in the main paper would add a substantial amount of background analysis which may detract, rather than add to how consumable the paper is to a broad audience. To help with this, I have added some more information on the results of this modelling, rather than the full modelling procedure. This should negate readers having to go to the supplementary results unless interested.

The behavioural data presented in the supplementary material is mainly background information on dealing with various behavioural scenarios. As above, more information on this has been included in the main manuscript to balance overloading the readers with background analyses versus providing enough information that the supplementary materials do not need to be consulted unless interested.

Reviewer 1

There are many cases where the Supplemental material is referenced, as well as previously published Dunlop papers (e.g. Dunlop et al 2013, 2016, 2018), where a summary or summary figure would help provide enough info to the reader of this paper to continue on without having to go back and read other papers to understand this one. These items are pointed out in the detailed comments below.

Response: addressed as per the specific comments below

Second, the word model and modelling is used throughout the paper to talk about acoustic modelling, communication space modelling, and statistical modelling. Some care to better differentiate between the difference types of modelling with qualifiers or descriptor would be helpful.

Response: the terms suggested by the reviewer have been used throughout the paper to differentiate.

One technical item that requires addressing concerns the acoustic modelling. What model was used for TL? I'm assuming spherical spreading because a circular communication space is referenced to in many places of the paper. The acoustic model was not explicitly stated. Use of a range dependent model would be more applicable in this case (if spherical spreading was indeed used), as range dependent bathymetry is crucial and would not produce circular communication spaces. I'm not advocating for a reworking of the data with a new model, but the details of what was used and its assumptions and limitations should be addressed.

Response: a site-specific empirically-derived transmission loss model was used. This model was developed and published in Dunlop et al. 2013, and has been used in many other studies at the site. As per the comments below, more detailed information about this model is included, though only briefly, as the author is aware of taking up journal space with material already published.

Detailed comments:

Title – I found the title wording and not straightforward. Suggest rewording. Possible suggestion “Is mitigating for the effects of vessel noise on baleen whale communication effective?”

Response: changed to

The effects of vessel noise on the communication network of humpback whales

Line 15 – Suggest “coastlines, they”

Response: done

Line 16 – Suggest “Since there are no empirical data”, since data are plural.

Response: done

Lines 22-23 – The hearing information is not the emphasis or focus of the analysis. Suggest removing it from the abstract.

Response: done

Line 26 – Suggest “additional confounding effect of”

Response: done

Line 53 – Duplicate “to compensate”.

Response: done

Line 83 – Suggest “condition, and the distance”

Response: done

Line 89 – Suggest “whales’ or whale’s”

Response: done

Line 93 – Can you clarify whether the receiver in this line of text refers to the intended receiver, unintended receivers, or both? The assumptions made for each may be different for the signaler and should be addressed.

Response: this refers to any generic receiver (in this case the quantification of SNR at a hydrophone receiver, which then could be used to inform SNRs at a generic receiver within the communication space. Changed ‘the’ to ‘a’.

Lines 111-121 – Positioning of individual vessel transits were discussed in this paragraph and linked to their contribution of vessel noise. How have you dealt with long distance shipping coming in from distant, unobserved ships? This would be a component of the low frequency noise more than the high-frequency component. Was this included in the “Wind Only” measurements and modelling? Does “Wind Only” just really mean the absence of close transiting vessels?

Response: Previous papers in this study site termed wind-only noise, wind-dominated as, given the tight correlation with wind speed (even at low frequencies), and the fact that the offshore shipping lane is within view and ships are only present intermittently. We cannot fully exclude a noise contribution from offshore shipping, but we are confident that the wind noise measures are wind dominated due to results of Dunlop et al. 2010. The section here relates only to fishing vessels traversing the areas. A sentence was added later in the methods to address this comment:

Wind-dependent noise was measured when there were no audible and/or visually tracked vessels in the area (including shipping vessels in the within-sight shipping channel). A previous study at this site found the measured noise in the absence of singer and vessel noise similar to that generally observed for wind-dependent noise in Australian waters (Cato 1996)

and found wind speed to be significantly correlated with noise levels in all third octave frequency bands (Dunlop et al. 2010).

Line 127 – What is “These” referring to? The social sounds detected on only 3 of 5 hydrophones, or did you select the best arrivals from 3 of 5 hydrophones?

Response: added in the word ‘hydrophone’

Line 140 – Where did the background noise measurement come from? 1-sec preceding the call? At another time related to the call? Did it include other background song?

Response: specified it was taken just before, or just after, the sound. Also included the sentence “This sample included background singing only if this sound source was also present within the social sound sample.” to clarify

Lines 140-141 – Confused on what this sentence is trying to convey? Please clarify.

Response: added in ‘the’ to clarify

Line 144 – Please clarify the use of DFT and FFT in the same sentence.

Response: changed and clarified with a reference - From the wavfile, an FFT was used to calculate the frequency content of the sound (FFT size of 16 384, 1.35 Hz resolution, 50% overlap) and its power spectrum extracted. The power spectrum was calibrated to account for the sensitivity of the hydrophone and the gain of the system (Girola et al. 2019).

Line 149 – Not enough information is given about the TL modelling here. A summary from Dunlop et al (2013) would be helpful. What TL model was used?

Response: added in some more details but as stated in the opening remarks, the author was cognisant of taking up journal space for something that has been previously published. Information added:

The boat conducted runs along lines radiating from the array, from distances of 100m out to about 10 km. The boat speed was kept constant. Wind speed for that day ranged from 10 to 15 knots, swell was less than 1m, and there were no significant currents in the area.

The value of a may have varied with the direction of the boat (approaching or going away from the receivers) as the noise radiated forward likely differed from the noise radiated aft. Even so, for most frequencies, b varied with distance but could be well approximated by two values; one applying to distances less than, and the other greater than, a cross over value. Absolute values of TL were determined by measuring received levels of octave band limited noise, projected using a J11 source suspended from a boat (estimating the source level using a hydrophone suspended at a distance of 3 m and corrected to 1 m assuming spherical spreading). This source was projected at three distances (between 200 and 1000 m) from the array. The trend in loss, $b \log(x)$, from the boat runs was then fitted to the absolute values of loss from the J11 measurements to determine the value of a for each octave band, by minimising the sum of the squares of the differences between $a + b \log(x)$ and the data points from the J11 measurements.

Lines 160-161 – When did these 10-min files occur in relation to the signals? On the same day, within the same hour?

Response: yes, added in a line to clarify. "As recordings were divided into 10-minute files for storage, the file containing the social sound of interest was used for the background noise measurement."

Line 171 – Suggest "hydrophone used to measure"

Response: done

Line 176 – What is Dunlop (31)? It would be helpful to see a distribution or histogram for the 1/3 octave bands to identify the dominant band.

Response: should be Dunlop et al., 2013. Changed. Did not add in a histogram as there is no one dominant band. Each sound type had a different dominant 1/3 band.

Line 181 – It would be helpful to summarize the info related to frequency category definitions from Dunlop (2018). At least provide the frequency range of low and high categories used in this paper.

Response: done

Line 185 – Which models are you referring to? Statistical, acoustic, propagation, overall communication space? This should be clarified for many uses of "model" or "modelling" throughout the text.

Response: all models referred to are statistical. Clarified.

Line 202 – Do you broadband noise or background ambient ocean sound?

Response: broadband as stated

Line 220 – More information is needed on why you are assuming a circular area around the signaler? Was this because you used spherical spreading? If so, the assumption of a circular area is most likely not a realistic assumption given the bathymetry, and this should be explicitly stated.

Response: agreed, but this was to generate the most simplistic representation of communication space and one that is likely to be used if no other info is available. Added in sentence to clarify.

Line 222 – Suggest "randomly distributed receiver groups" if that is what is intended.

Response: done

Line 224 – Missing the link here between the communication model results and the number of groups left in the area. Is this related to behavioral observations?

Response: clarified as loss of radial distance

Lines 232-234 – A figure showing the 2 groups for comparison would be helpful.

Response: not included due to journal space and duplicity. The first paragraph of results (next paragraph) details what would be in the figure.

Line 381 – The word compared implies a direct comparison, but there is no comparison in this sentence.

Suggest “The loss of total communication area in vessel noise and wind-dominated noise was approximately 75% for both signal types.”

Response: this is not what the sentence meant, communication area was 75% less in vessel noise as compared to wind noise. Re-worded to make clearer.

Line 395 – Can you please verify “by” or “to”?

Response: verified as ‘by’, not changed

Line 443 – circular area from signaler is again an assumption that warrants clarification.

Response: done

Reviewer 2

This paper promises to present a worth-while discussion, taking the organism perspective in how increases in ambient noise, driven by vessel noise and wind noise, may change behaviour in vocalising and/or interactions between humpback groups. What it does really is just describe the extent of the communication space based on interactions between groups of whales based on different ambient noise regimes. I can’t help but think that more can be made from the data that was collected, both acoustic and observational, which would enlighten the discussion further, and more to what was promised from the title/abstract.

Response: the author respectfully disagrees with this statement. It uses measured received levels of social sounds, along with an empirical propagation model, to estimate at communication range based on the assumption of SNR = 0 to delimit the range. In then uses interactions between groups of whales (given that is the likely to be at least one of the functions of the social sounds) to **inform** this estimation of communication range (i.e. these interactions are not used to describe). In all cited references, there is no inclusion of behavioural data to inform estimated communication space.

It seems like a lot of material is trying to be drawn from the data presented: that baleen whale hearing capacity might be better determined;

Response: removed as recommended by reviewer 1

that masking might be discussed in context of communication space, and that the implication of masking on behavioural interactions may be determined.

Response: included as per the comment below, where focus to one of them was given

More focus given to one of these may help with streamlining the paper and making it more directed.

Highlighting the decreased communication space and how that has wider implications in conspecific interaction I think is the best way to go here, and is where the author seems to take it but there needs to be some more definitive statements of the study question and findings. As presented the data and analysis would not answer this question.

Response: addressed within comments further down

The use of visual data in conjunction with acoustic recordings can afford some very interesting results and a strength to the conclusions not possible from acoustics alone. A little more could be drawn out the significance of using these to different types of datasets together. There was a lot left to the reader to follow up on (e.g. social calling/vocalisation types, behavioural categorisation) or that is not clearly defined (like wind dominated ambient noise) or explained (some of the analysis – see comments).

Response: addressed within comments further down

Missing from the discussion is the implications of taking an individual-centric or communication space approach, and how that might play into understanding of behaviour or mitigation actions. Also it could lead to a discussion on how this approach may lead to different/better management actions than are currently in place, but did not.

Response: the problem with this is that the study did not take an individual-centric approach, therefore it cannot state how this approach would lead to different management actions. It also cannot, as stated in the discussion, say if these effects would lead to any population level effects, without much further study.

The paper presents a lot of material, but it feels piecemeal, and like a lot of different pieces of data forced together currently, with it lacking a main focus or directive that is clearly stated to being about cohesiveness.

Response:

The paper would benefit from proof-reading and going over to make it more concise, focusing on the follow on of one paragraph to another and the general message to the reader. It doesn't have an easy flow to the material.

Response: addressed within comments further down

Title: suggest rewording the title. The paper presents discussion from vessel and wind noise The short title almost works better as a descriptor of the material presented (masking of humpback whale signals) as the masking effect is where this paper focuses and how that effects the space over which whales interact.

Response: done, title suggested by reviewer 1

Abstract

Reads a little awkwardly, some rewording may be helpful here with the flow.

Introduction:

I find the introduction to lack flow within and between sections. There seems to be too much covered. If the how communication space for social sounds, as modeled, in different soundscape settings (wind or vessel noise) affects social interaction between whales became the clear focus some of the less relevant/irrelevant material could be removed and make the section more concise and easier to follow.

Lines 32-42: Don't think this is relevant to the discussion. That there is an acoustic communication network and this network is defined by the propagation of each call type in ambient noise conditions if the underlying basis of this paper. Then go on to discuss the conditions in ambient noise, and additions to the soundscape (both natural and anthropogenic) that might alter this.

A description of sources of noise that comprise the full soundscape may also be worthwhile to the reader too, perhaps describing the differences in natural (geophonic) sources and human-derived (anthrophonic) noise sources.

Response: it is important that non-cetacean readers understand that the communication network in cetaceans, like humpbacks, is likely to be large. This provides context in that a reduction in communication space, is likely to reduce their communication network, which then will have implications on social behaviour (like those found here). To make it flow better with the second paragraph (comment below), have added in linking sentences to the second paragraph (linking to the first) and re-arranged the second paragraph as per the comment below. In addition, the associate editor recommended making this paper readable to a wider audience, therefore animal communication networks have been left in.

Again in the second paragraph (lines 45-60) – discuss masking first, and its implications on communication, and then what baleen whales have been noted to do to compensate for increased ambient noise. For this paper, as it references so many previous studies the most succinct way to introduce material, without losing the reader, would be to use these previous works as a jumping off point, i.e we saw compensation mechanisms in increased wind noise and not in vessel noise, knowing this we interpret the reduction of communication space to have implications on socialising behaviours (the focus of the study).

Response: done

Maybe somewhere explicitly say that you are using communication space as the distance that conspecifics can send and receive acoustic messages, and where there is a observable (social) response in the receiving whale.

Response: done

Line 64: may be nice to describe briefly under what conditions the propagation distances were calculated. Is there a general description for social sounds (in frequency, length, modulation etc.). Some more details here would likely help the reader.

Response: done. Included the fact that the environment was a shallow-water environment (<50 m), plus added in:

Their repertoire of social vocalisations is diverse, spans a fundamental frequency range of less than 40 Hz to to over 2 kHz and includes sound types ranging from low-frequency “grumbles”, to high-frequency bird-like “chirps” as well as both modulated (e.g. “moans”, “trumpets” and “cries”), amplitude-modulated (e.g. “purrs” and “growls”), and broadband (e.g. “roars” and “underwater blows”) sounds (Dunlop et al., 2017).

Lines 82-87: some of this detail may be better in the methods/ is in the methods and doesn't need repeating

Response: respectfully disagree, this setting up the background to the work, that we rely on modelling the communication space because we have not empirical data on receiver hearing

Line 89: 'will modelled and compared' should be 'will be modeled and compared to conditions of...'

Response: done

Materials and Methods

There are some details missing that would help the reader, for example, is there a baseline or 'background' noise level defined.

Response: this was already included in the manuscript

These varied from approximately 90 dB *re* 1 μ Pa to 112 dB *re* 1 μ Pa in wind speeds ranging from <5 knots to approximately 15 knots. The median wind-dominated noise was 100 dB *re* 1 μ Pa for this area.

And further down:

Measured noise levels were above 100 dB *re* 1 μ Pa with mean and modal measured levels being 105 dB *re* 1 μ Pa (vessels were approximately 4 km from the array).

What is average wind speed/noise, and what is the definition of 'wind dominated'.

Response: added but wind speed was already included

What was the process of comparing the visual and the acoustic data?

Response: this has been covered in the manuscript though a sentence has been added in to make clearer the link between the visual and acoustic data (highlighted)

Acoustic recordings were made of groups of migrating (from their breeding ground in the Great Barrier Reef, towards their Antarctic feeding grounds) of humpback whales (September/October of 2002 – 2004 and 2008). The array comprised of five hydrophone-buoy systems anchored in 20 – 28 m of water in a 'T' configuration. Buoys 1 to 3 were anchored in a line 1.5 km offshore, with buoys 4 and 5 extended a further 1 km offshore. Each system included a High Tech HTI-96-MIN hydrophone with built-in +40 dB pre-amplifier, a pre-amplifier (+20 dB) and VHF radio transmitter. Real-time radio transmissions from the buoys were received a base station using a Yagi antenna attached to a four channel, low noise, VHF receiver (type 8101) and a Winradio receiver. Sound source locations were calculated by *Ishmael* where the mean position of several estimates were calculated over a brief period. A shore-based theodolite survey of the surface buoys was used to obtain accurate positions. For detailed validation experiments see Noad et al. (2004).

Groups were recorded in water depths ranging from 25 to 35 m. Simultaneous land-based tracking of the groups occurred to obtain visual data of migrating humpback whale groups within a 10 km offshore radius from the acoustic array. All groups were tracked regardless of whether or not they were vocalising. Group positions were recorded using a theodolite linked to *Cyclopes* (developed by Eric Kniest, University of Newcastle) and additional observations on group social composition (number of adults within the group and whether or not there was a calf) and social behaviour (splitting of animals from, and joining of animals to, the group) were recorded at each position. The

position of traversing vessels was also recorded by the land-based observation team. The visual and acoustic positions were overlaid onto the one Cyclopes file, meaning the acoustic tracking could be matched to groups being visually tracked. Using this combination of visual and acoustic tracking data, the distance of each signalling group to each hydrophone-buoy receiver was measured for each recorded social sound.

Line 106: Should read were received at a base station

Response: done

Line 111: Were groups only tracked from shore?

Response: yes

Line 113: Be sure it is clear: vocalisations are localised to a position using the array, the calls are then attributed to a group using visual recordings, the group positions are tracked using theodolite.

Response: edited to make clear

Are all call types used, or just social calls. Are there social and travel calls?

Response: we have don't know the function of these calls so prefer not to say

Line 115 onwards: more detail could be given on the theodolite tracking methods.

Response: some further details added

Line 118: At what interval were these positions taken? Is this in time or space?

Response: at each surfacing, added

Line 120: were the visual and acoustic tracks compared? Were there groups that did not vocalise? Under what conditions may this have happened?

Response: sentence added in to address this (see highlighted sentence above)

I assume that the comparisons between visual and acoustic are limited to day light hours, but this is not specifically stated.

Response: sentence added

Were the comparisons limited to only social calls when you made them? The points made in Lines 119-121 need to be highlighted and more specifically described.

Response: done

Is the distance between the whales and vessels also calculated?

Response: not for this study, a vessel just had to be in the area, also, the noise was measured at the array, not the whale

Line 124: A better description of wind noise would be useful to the reader. Is this over a particular wind speed or is it an increased ambient noise level when vessels are not present.

Response: done

Line 125: Is there a specific SNR/dB level for 'first becomes audible'. I presume this is derived manually through inspection of spectrograms.

Response: yes and added

Line 127-129: Unclear

Response: changed to make clearer

Line 131: Is this distance from the array and within the hydrophones receiving the calls in the array a means to derive propagation distances?

Response: yes, but with the caveat the SNR = 0 defines the distance (as stated in the intro)

Could the visual tracks and distance from the array and the 'first becomes audible/now not audible' be used somehow in discussing the likely communication space under each ambient noise scenario.

Response: yes, but this equates to an approximate SNR of 0 so the answer would be the same

Maybe take a sentence or two after line 136 to say what these technical specifications mean in terms of the calls received and how they might affect the reception and interpretation of calls to the reader.

Response: done

Line 139: more description or reference to social sound types is needed here.

Response: these were described in the intro (in response to reviewer 1) and further details provided in the opening lines of the communication space section

Line 140: How was the sample of background taken/selected.

Response: not sure what the reviewer is referring to here?

Line 161: How were the files selected? What is deemed as 'wind-dominated' noise. Is wind dominated purely vessel absent?

Response: more details added in response to reviewer 1

Line 165: What is the wind speed at the average wind speed noise.

Response: added

Line 171: used to measure the signal

Response: changed

Line 175: How does using the 1/3 octave band affect the interpretation compared to, say, using the peak frequency when the SNR was at its greatest.

Response: the 1/3 octave band would contain the SNR peak frequency and it is standard now to measure humpback sounds using a 1/3 octave band (added this in to make clear)

Line 176, Line 184: make sure referencing is complete, consistent and in line with the journal

Response: done

Lines 184-195: Much of this seems to be information presented elsewhere. Be sure when describing this that you are, then, adding something of note to the initial analysis. Be sure that the reader is clear that the rSNR you reference is the SNR of a call heard by a receiver

Response: the author disagrees. The addition is using the SNR = 0 to statistically define the communication space which hasn't been mentioned previous to this. Also not made clear as the receiver, in this case, is the array, not the whale – added in statement further up to clarify.

Line 194-195: Not sure what this means

Response: changed

Line 196: run not ran

Response: done

Line 201: 'R'

Response: done

Lines 219: Could you draw on your observations here with the inter-group communications rather than randomly placed groups?

Response: the problem with this is that we do not know which group is the intended receiver, versus unintended. Plus, this line refers to randomly place simulated groups therefore we have no inter-group communications to comment on?

Line 230-234: Why combine the two noise environments here, especially as the behavioural response to these noise additions have previously been seen to be different.

Response: as stated, for this model, to re-create the distribution of the other groups relative to the receiver, there was no significant difference between the group distribution in wind noise compared to vessel noise. This analysis is only to generate a robust representation of group distribution, not to determine if there are differences (which we know there are not) in distribution between the 2 environments.

Results:

Social, Line 240-245: Is this from the observational or modelled data?

Response: clarified as observational

Lines 256-261 – perhaps describe ‘typical’ wind and vessel noise and how this overlaps with typical social calls

Response: this was already described in the first sentence of the paragraph

Lines 385-398: Could it be that the low-frequency communication space/low frequency calls/communications are less affected by wind/vessel noise

Response: no, low-frequency calls are generally louder so propagate further. Clarified

Is there any difference seen in the calls used/calling behaviours? Were the previous assertions for call/behavioural modifications due to increased ambient noise confirmed in this study?

Response: no, this was tested in Dunlop 2016b (in terms of switching from vocal to surface-generated) but not here. This manuscript is focussed on communication space in terms of signal level, rather than sounds used and I argue that there is enough content in here without further analyses. In response to the second point, this has been covered in the discussion.

Discussion

This discussion again could do with more focus and linking back to the research question. Work on the flow of the material presented. The discussion material should show a stronger link back to the results of this study.

Response: changed as suggested. The first two paragraphs now re-iterate the original goals of the study (as found in the last paragraph of the intro), and re-iterate and then the discuss the results which address these goals.

Line 426: Typical wind conditions or typical noise conditions – and what percentage of the time of recording does this represent?

Response: wind noise conditions as stated, these were defined in the results. Unclear what the reviewer means by the second comment, wind speeds were unlikely to change over the 20 to 80 minute recording, so this would be the whole recording?

Line 425-430: Even the discussion between high and low frequency communication space – and be sure to tie it back to humpback whale communications.

Response: unclear what the reviewer means by this. The acoustic communication space and matching behavioural observations were stated for both low and high frequency sounds so the discussion was even. To address the second point, we can only hypothesise how these sounds are used within their communication. Added in two sentences to speculate on the function of each of these sound groups as well as sentence to state that further work is needed to clarify their function.

Line 453-455: Again I think it might be possible to draw more out of the observational data than just the random distribution model.

Response: included more discussion on the behavioural data as compared to the random model and the receiver distribution model.

Line 473-474: Develop this point further

Response: done - sentence added below

Behavioural interactions were most likely to occur when groups were separated by 1 km or less where most signals, even high-frequency signals, were still well above received vessel noise and unlikely to be masked.

Appendix C**ROYAL SOCIETY
OPEN SCIENCE****The effects of vessel noise on the communication network
of humpback whales**

Journal:	Royal Society Open Science
Manuscript ID	RSOS-190967.R1
Article Type:	Research
Date Submitted by the Author:	31-Aug-2019
Complete List of Authors:	Dunlop, Rebecca; University of Queensland, Veterinary Science
Subject:	behaviour < BIOLOGY, ecology < BIOLOGY, environmental science < CROSS-DISCIPLINARY SCIENCES
Keywords:	masking, humpback whale, anthropogenic noise, acoustic communication, active space
Subject Category:	Biology (whole organism)

Author-supplied statements

Relevant information will appear here if provided.

Ethics

Does your article include research that required ethical approval or permits?:

Yes

Statement (if applicable):

The project operated under animal ethics approval from the University of Queensland Animal Ethics Committee, a Queensland Government Environment, and Heritage Protection permit and an Australian Government Access to Biological Resources in a Commonwealth Area for Non-Commercial Purposes permit.

Data

It is a condition of publication that data, code and materials supporting your paper are made publicly available. Does your paper present new data?:

Yes

Statement (if applicable):

The article's supporting data are available on the Dryad Digital Repository.
<https://datadryad.org/review?doi=doi:10.5061/dryad.65j00m3>

Conflict of interest

I/We declare we have no competing interests

Statement (if applicable):

CUST_STATE_CONFLICT :No data available.

Authors' contributions

I am the only author on this paper

Statement (if applicable):

CUST_AUTHOR_CONTRIBUTIONS_TEXT :No data available.

**The effects of vessel noise on the communication network of humpback whales**

Rebecca A Dunlop

*Cetacean Ecology and Acoustics Laboratory, School of Veterinary Science, University of*

*Queensland, Gatton Campus, Queensland, QLD 4343, Australia*

Electronic mail: *r.dunlop@uq.edu.au*

Short title: Masking of humpback whale signals

Humpback whales likely rely on social sounds to mediate social interactions. The distance to
which these social signals propagate from the signaller will define its communication space,
and therefore communication network (number of receivers). As humpback whales migrate
along populated coastlines, they are likely to encounter noise from vessel traffic. Since no
empirical data exists on baleen whale hearing, the consequences of this are usually assumed,
being the modelled reduction in their communication space. Here, the communication space
and network of migrating humpback whales was compared in increasing wind-dominated and
vessel noise. Behavioural data on their social interactions were then used to inform these
models. In typical wind noise, a signaller's communication space was estimated to extend to
4 km, which agreed with the observed extent of their social interactions. An increase in vessel
noise reduced the modelled communication area by a factor of four, with a significant
reduction in group social interactions implying a reduction in their communication network.
However, signal masking did not fully explain this change in social behaviour, implying there
was also an additional effect of the physical presence of the vessel on signaller and receiver
behaviour. Though these observed changes in communication space and social behaviour
were likely to be short-term and localised, an increase in vessel activity due to tourism and
coastal population growth may cause more sustained changes along the humpback whale
migration paths.

**Introduction**

Vocal communication networks involve a signaller, one or more intended receivers, and one
or more ‘unintended receivers’. Unintended receivers, referred to as the ‘audience’, as
‘bystanders’, or as ‘eavesdroppers’ (McGregor, 1993; McGregor et al., 2001; Otter et al.,
1999; Peake et al., 2001), can gather and use information from this signalling interaction to
their advantage. The size of this communication network is usually defined as the distance
radius from the signaller at which any other receiver can detect and interpret the signal. This
depends on the signal level and structure, the propagation environment, levels of background
noise, and the receiver’s ability to detect, recognise and discriminate the signal within noise
(Wiley and Richards, 1978). In the marine environment, as sound propagates over large
distances, cetacean communication networks can be extensive and involve multiple intended
and unintended receivers. Cetaceans are likely to utilise these networks within feeding (e.g.
Janik, 2000; Miller, 2006) and breeding interactions (e.g. Dunlop, 2016a; Dunlop and Noad,
2016), meaning effective communication exchanges are essential for these life functions.

In the underwater environment there are several natural sources of noise, such as
noise from surface breaking waves during periods of high wind. Noise sources can also be
anthropogenic in origin, such as from vessel activity, oil and gas exploration, naval sonar
activity, or construction. These noise sources have the potential to mask signals, meaning
they become inaudible to receivers within the network as the noise levels go up. This will
reduce the size of the communication network and, in turn, may have detrimental effects on
feeding or breeding interactions within these networks. Important data for predicting the
potential impacts of masking include, for the receiver, basic audiograms, critical ratios (CRs)
of hearing, and identification of mechanisms that allow receivers some release from masking
(Erbe et al., 2016). Little is known, however, hearing capabilities in baleen whales therefore
it is difficult to predict how receiving whales cope with increased noise (Erbe et al., 2016).

For the signaller, important data for predicting masking includes which anti-masking
strategies are used (Erbe et al., 2016). Signalling whales, for example, use strategies such as
increasing their signal level (Dunlop et al., 2014; Parks et al., 2011), changing their signalling
behaviour (Dunlop et al., 2010), and changing the frequency content (Heiler et al., 2016;
Lesage et al., 1999; Parks et al., 2007) and duration of their signal (Miller et al., 2000) . It is
likely, however, that there is a limit to the success of these strategies. If the noise source is
anthropogenic in origin, for example, and noise levels are above natural noise levels,,-
signaller anti-masking strategies may not be as effective. Therefore, the receiver’s ability to
detect and interpret relative sounds may be reduced over and above that in natural noise, with
a higher risk of loss of communication space and reduction in communication network.

Acoustic communication in humpback whales (*Megaptera novaeangliae*) has been
extensively studied making this a good model species for cetacean masking studies. The
communication space of social sounds (‘vocalisations’, blow-hole associated sounds, and
percussive signals generated at the surface) is in the order of km’s (Dunlop, 2018) in a
shallow-water environment (<50 m) and includes a network of intended and unintended
receivers (Dunlop, 2016a; Dunlop and Noad, 2016). Their repertoire of social vocalisations is
diverse, spans a fundamental frequency range of less than 40 Hz to to over 2 kHz and
includes sound types ranging from low-frequency “grumbles”, to high-frequency bird-like
“chirps” as well as both modulated (e.g. “moans”, “trumpets” and “cries”), amplitude-
modulated (e.g. “purrs” and “growls”), and broadband (e.g. “roars” and “underwater blows”)
sounds (Dunlop et al., 2007). These social sounds are commonly heard during breeding,
(Dunlop, 2016a; Dunlop and Noad, 2016; Rekdahl et al., 2015; Silber, 1986) and feeding,
group interactions (D’Vincent et al., 1985; Parks et al., 2014; Sharpe et al., 1998; Thompson
et al., 1986) and likely function to provide information on group membership, changes in
group membership, signaller sex, size, location (Dunlop et al., 2008) and motivation (Dunlop,

2017). Previous studies found that the communication space of humpback whale signallers
extends to approximately 4 km (Dunlop, 2018) and that signallers maintain this space in
increased wind-dominated noise by both switching from vocal sounds to surface-generated
sounds (Dunlop et al., 2010) and increasing their vocal source level (Dunlop et al., 2014).
Interestingly, in response to increased vessel-dominated noise, there was no evidence that the
signaller used these anti-masking strategies (Dunlop, 2016b). This implies that when vessels
are traversing the area, the communication space of a humpback whale signaller is
significantly reduced. The consequences of this within the context of breeding interactions, if
any, remain unknown.

In baleen whales, apart from anatomical modelling, there is no empirical data on their
hearing in noise. Therefore, if predicting the impact of masking on the receiver, the signal
propagation is usually modelled in various noise conditions and the distance at which the
signal disappears below noise is assumed to equate to the receiver's audible limits. In other
words, an assumption of receiver hearing in noise, and/or the receivers's ability to recognise
and discriminate the signal must be made. Cetacean communication space is therefore usually
modelled using a presumed limit in receiver audibility (e.g. Tennessen and Parks, 2016), and
in some cases, a presumed ability of the receiver to recognise the signal (e.g. Cholewiak et
al., 2018; Hatch et al., 2016). These studies assume the distance at which the measured
received level of a signal-to-noise ratio, or signal excess, is of a certain value (e.g. 0 dB)
delimits the signaller's communication distance. This study will follow the approach outlined
above to define the communication space of signalling humpback whales but will use
behavioural observations to further inform results. Therefore, in this study, communication
space is defined as distance that conspecifics can potentially send and receive acoustic

[revised manuscript text omitted]

**Communication space**

The communication space for the low-frequency vocalisation category was estimated to be 4
306 km in typical (received noise levels of 100 dB *re* 1 μ Pa) wind noise, reducing to 3 km in the
307 same received level of vessel noise (Fig. 1a). These noise levels equated to typical wind

conditions and relatively low received levels of vessel noise. The maximum interaction
distance of groups was a separation of 4 km, where a neighbour turned to head towards a
signalling group from this distance to eventually join together ($n = 2$). This separation
distance matched the likely extent of the communication space for low-frequency sounds in
typical wind-dominated noise. The size of the communication network for migrating
humpback whales, in this shallow-water environment (<50 m), was therefore likely to be
between two and four other groups based on the distribution of groups from the signaller.

For high-frequency vocalisations, the communication space was 2.5 km in typical
wind noise, given they were lower in level (mean of 149 dB *re* 1 μ Pa @ 1m compared to 156
317 dB *re* 1 μ Pa @ 1m in low frequency sounds). This would reduce the number of receiving
groups within the network to one or two. Groups, in wind noise, had a +50% chance of
interacting if separated by less than 2 km (Fig. 2) suggesting this distance was likely to be
important in terms of acoustically-mediated group social behaviour (joining together). Even
in relatively low received levels of vessel noise (100 dB *re* 1 μ Pa), the communication space
of high-frequency vocalisations reduced to 1.5 km (Fig. 1b).

**Effect of increased noise**

As received noise levels increased to 105 dB *re* 1 μ Pa (mean received vessel noise and above
average wind noise), the communication space decreased to approximately 2 km and 1 km
for low- and high-frequency vocalisations respectively (Fig. 2a), meaning the network would
likely be reduced to one receiver. By received noise levels of 110 dB *re* 1 μ Pa, which is
considered to be high wind noise, or noise from a close-by (within 2 km) vessel, the
communication space for both signal types was reduced to only 1 km, which may or may not
include one receiver depending on its distance from the signaller. However, although the
reduction in communication space was the same for both noise sources, a neighbour was

significantly less likely to interact with the signalling group when there was a vessel in the
area ($t = 2.19$, $p = 0.03$; Fig. 2a) compared to if the noise source was wind-dominated. As
signalling groups were most likely to interact with their nearest neighbour, the position of this
receiving group within the signaller's communication space was the most important to note
(Fig. 2b and c). One explanation for this reduction in interaction behaviour would be
increased signal masking, as received noise would be likely be elevated above typical wind
noise if a vessel was in the area. However, noise level was not a significant predictor in that
increased noise did not explain any reduction in the likelihood of groups to interact.

Signalling groups had an 80% chance of interacting with another group if separated
by 1 km as long as there was no vessel in the area. This reduced to a 30% chance if a vessel
was present (Fig. 2). Even in vessel noise, many received signals would have been above
noise at this close-by separation distance (Fig. 1) suggesting masking was not a significant
issue. If using 105 *re* 1 μPa as a generic (average) received vessel noise level, though the
modelled communication space of both vocalisation types was significantly reduced, it was
still beyond 1 km (Fig. 1). Using signals at which the $rSNR = 0$ does not include any measure
of variance, in that, signals received by a whale at 1 km are likely to range in $rSNRs$. If
choosing all signals emitted from groups 1 km positioned at 1 km from the receiver, the $rSNR$
of low-frequency signals, in received vessel noise of 105 *re* 1 μPa , was up to + 13 dB (Fig.
3a), and for high-frequency signals, up to + 12 dB (Fig. 3b). The decrease in the $rSNR$ with
increasing measured noise was also significantly less in vessel noise compared to wind-
dominated noise (Table 1; Fig. 3b). In other words, low-frequency signals were less masked
by vessel noise as long as the signaller was 1 km from the receiver. Therefore, at close group-
to-group separation distances, the observed reduction in interaction behaviour was likely not
due to signal masking.

**Communication area and group distribution**

The loss of total communication area in vessel noise was 75% for both signal types (Table 1)
as compared to communication area in wind-dominated noise. Assuming a random
distribution of groups within that area (4 groups per 5m), this equated to an approximate
77 to 78% reduction in available receivers within the network (Table 1) for both signal types.
However, the distribution of groups at this study site suggests a bias towards the signalling
group, where the density of groups (available receivers) was higher closer to the signalling
group (Fig. 1). Using this known distribution, the reduction in available receivers due to loss
in communication space was estimated as 47% and 33% for the low and high-frequency
signal communication area respectively (Table 1). Behavioural data from this study suggests
the reduction in low-frequency signal communication space was unlikely to lead to a
significant reduction in social interactions, given most of these interactions occurred within 2
370 km of the signalling group. It was also clear that the probability of a receiver group
interacting with a signaller group was dependent on its position and distance from the
signaller (Fig. 2b and c). In wind noise, the signalling group had a 70% chance of interacting
with the nearest neighbour if the separation distance was less than 2 km, reducing by 50% in
vessel noise. Therefore, this reduction in interaction behaviour was greater than that predicted
by using known group distribution as a proxy for the likelihood of social interactions, but less
than using a random distribution of groups within the communication space assuming a
simplistic spherical communication area.

Table 1. The modelled communication area based on the distance from the signaller at which
 the $rSNR = 0$. The group numbers within the wind and vessel noise communication areas
 were estimated assuming a random distribution (of 4 groups per 50^2km) and known
 distribution, using visual data of groups around signalling groups. Standard deviations are in
 brackets.

Signal type	Distance km	Area ^2km	Random distribution		Known distribution	
			No. groups	% reduction	No. groups	% reduction
Low-f	4	50	4		2.2 (1.0)	
	2	50	0.9 (0.9)	77 (22)	1.0 (0.8)	47 (36)
High-f	2	12	0.9 (0.8)		1.0 (0.8)	
	1	12	0.3 (0.5)	78 (35)	0.5 (0.6)	33 (44)

Discussion

This goal of this study was to define the communication space and network of
 signalling humpback whales and determine if, and how, this was reduced during periods of
 vessel noise. Here, communication space was defined as distance that conspecifics could
 potentially send and receive acoustic signals (assuming a signal excess of 0 dB delimited this
 distance), and where there was an observable (social) response in the receiving whale. First
 the communication space and network was defined in wind-dominated (natural) noise.
 Results suggest that humpback whale low-frequency social signals had a communication
 space of approximately 4 km with a communication network of two to four other groups.
 Behavioural data found that humpback whale receivers responded to, and eventually joined, a
 signalling group from a maximum separation distance of 4 km. Therefore, these sounds may
 function to mediate longer-range interactions between networks of dispersed groups.
 However, the local neighbourhood of signalling humpback whales, where sighted
 behavioural interactions were most likely to occur, was more similar to the communication

space of higher frequency signals (2 km). Higher frequency signals were of a lower
amplitude compared to low-frequency signals meaning they may be used mediate closer-
range joining interactions between a signaling and receiving groups. There was usually only
one other group within this network; the group participating in the joining interaction.

To test for effects of anthropogenic noise, the likely extent of a signalling whale's
communication space was compared in natural (wind) and anthropogenic (vessel) noise.
In average vessel noise (105 dB *re* 1 μ Pa), the communication space of humpback whale
low-frequency vocalisations reduced, by half, to 2 km. These results are similar to modelled
loss in communication space of humpback whale social sounds in vessel noise in Cholewiak
(et al., 2018). The communication space of high-frequency sounds was also reduced by half
in vessel noise to only 1 km. This translates to a reduction of four times the communication
area, assuming a circular area from the signaller for both signal types. To assess the
consequences of increased vessel noise on the communication network, the magnitude of the
reduction depended on the method use. If assuming a random distribution of groups around
the signaller, the number of available receivers within the network reduced from four to one.
However, there are two problems with this assumption. First, the receiving groups were not
randomly distributed from the signaller, and second, signallers were more likely to interact
with the closest receiver as long as it was within 2 km. After accounting for these two
problems, the reduction in available receivers decreased to 47% and 33% respectively.
Behavioural observations found a 50% reduction in the likelihood of a signalling group
interacting with a receiver group within the 2 km interaction zone. Therefore, using the
random distribution of receivers over-estimated the reduction in communication network
whereas accounting for receiver distribution, but not receiver behaviour, under-estimated
network consequences. However, as explained below, there was an additional factor to
consider, being the physical presence of the vessel.

Though this study focused on the acoustic environment and behaviour of receivers,
potential signaller anti-masking strategies were also considered. The estimated
communication space of low-frequency signals in wind-dominated noise was maintained at
approximately 4 km up to received levels of 100 dB *re* 1 μ Pa. This was likely due to the
Lombard response found humpback whales, where signallers increased their vocal source
level in response to increased wind-dominated noise (Dunlop et al., 2014). In the same
received levels of vessel noise, the communication space was found to be smaller. A previous
study found no signaller Lombard response to increasing vessel-dominated noise (Dunlop,
2016b). Vessel noise levels were estimated at the signalling group (in Dunlop, 2016b) and
were comparable to those measured at the array in this study. In other words, received levels
of vessel noise should have been high enough to cause a Lombard response in signalling
whales but did not. One explanation put forward in this study was that whales have release
from masking, in that they can focus their attention on the vocalising conspecific, rather than
the vessel noise. This implies the receiver groups, if possessing the ability for masking relief,
would continue to behave as if there was no masking noise source in the area. Behavioural
results presented here showed this was not the case. Further, the reduction in humpback
group interaction behaviour in vessel noise was not fully explained by signal masking.
Behavioural interactions were most likely to occur when groups were separated by 1 km or
less where most signals, even high-frequency signals, were still well above received vessel
noise and unlikely to be masked. Therefore both the lack of signaller Lombard response
found in a previous study (Dunlop, 2016b), and the significant change in receiver behaviour
found here, are more likely to be an avoidance response to the vessel itself. The physical
presence of the vessel has been shown to reduce bottlenose dolphin (Bas et al., 2017) and
humpback whale (Blair et al., 2016) foraging activity as well as bottlenose dolphin resting
and socialising behaviour (Bas et al., 2017). Here, the combination of communication space

and behavioural analysis illustrates the need to inform one result with the other. This provides
a powerful tool with which to fully assess the consequences of anthropogenic noise on
cetacean social behaviour which does not rely on communication space modelling and
assumptions of group distribution and social behaviour.

The population used for this study is robust in that it is increasing at 11% per year
(Noad et al., 2019). These behavioural interruptions within the communication network, if
localised and rare, are unlikely to have a significant effect on such a robust population. As
many of these social interactions are likely to be breeding interactions (Dunlop et al., 2008;
Dunlop 2016a, Dunlop and Noad, 2016), understanding the population consequences of
disturbance, in this case reduced potential breeding opportunities, would be the next step in
understanding whale and vessel interactions. Given this population is also well studied in
terms of population dynamics, a population consequences model would not be difficult to
produce once the link between reduced group interactions within the communication network
and reduced mating opportunities has been established. These models could be extended to
other humpback populations and other baleen whale species, such as the highly endangered
right whale (*Eubalaena spp.*), where less is known about their breeding behaviour. For now,
the most comprehensive model to estimate loss of communication space for baleen whales in
vessel noise would be to account for signal type, receiver group distribution with some
knowledge of the local signalling neighbourhood, and any additional (e.g. avoidance) effect
of the presence of the vessel.

Along the eastern coast of Australia, and likely other populated coastlines, vessel
activity is increasing due to the growth in the tourism industry. For cetaceans occupying a
coastal habitat, vessel interactions are becoming more of an issue. Current Australian
mitigation measures regarding the interaction of vessels and whales include caution zones
(within 300 m of the whale) and exclusion zones (within 100 m). The assumption with these

measures is that staying more than 300 m away “reduces the risk of disturbance to natural
behaviours”. The results of this study suggests otherwise. What is clear is that the deleterious
effects of vessels on whales are not limited to collisions and increased signal masking but
potential changes in vital function behaviours, such as breeding interactions, should also be
considered. In summary, mitigating vessel activity from a noise and communication space
perspective may not mitigate for behavioural changes within their communication network.

Ethics and permits

The project operated under animal ethics approval from the University of Queensland Animal
Ethics Committee, a Queensland Government Environment, and Heritage Protection permit
and an Australian Government Access to Biological Resources in a Commonwealth Area for
Non-Commercial Purposes permit.

Data accessibility

The article’s supporting data are available on the Dryad Digital Repository (Dunlop, 2019):
<https://datadryad.org/review?doi=doi:10.5061/dryad.65j00m3>). Wind and vessel noise
communication models are uploaded as part of the supplementary material.

Competing interests

There are no competing interests in that funding agencies had no input in to experimental
design, data collection, data analysis or interpretation or the writing of the paper.

Author contributions

RD conceived the study, designed the study, carried out the data analysis and wrote the
manuscript. Raw data were collected as part of a previous collaborative project (HARC, see
acknowledgements).

**Acknowledgements**

The author would like to thank everyone involved in the Humpback Acoustic Research
Collaboration (HARC; funded by the U.S. Office of Naval Research and the Australian
Antarctic Division), in particular the numerous volunteers who donated their time and
energy to this project. I also thank David Paton for his invaluable field expertise and Eric
Kniest for his continued support in the development of *Cyclopes*. The author would
particularly like to acknowledge Associate Professor Michael Noad for leading the HARC
work (without which, this study would not have been possible) and Dr. Douglas Cato for
his continued support and mentorship.

Bas AA, Christiansen F, Ozturk B, *et al.* 2017. Marine vessels alter the behaviour of
bottlenose dolphins *Tursiops truncatus* in the Istanbul Strait, Turkey. *Endanger Species Res*
**34**:1-14.

Blair HB, Merchant ND, Friedlaender AS, Wiley DN, Parks SE. 2016. Evidence for ship
noise impacts on humpback whale foraging behaviour. *Biol Lett* **12**(8).

Cato, DH. 1996. Ambient sea noise in waters near Australia. *J. Acoust. Soc. Am* **60**:320–328.

Cholewiak D, Clark CW, Ponirakis D, *et al.* 2018. Communicating amidst the noise:
modeling the aggregate influence of ambient and vessel noise on baleen whale
communication space in a national marine sanctuary. *Endanger Species Res* **36**:59-75.

D'Vincent CG, Nilson RM, Hanna, RE (1985). Vocalization and coordinated feeding
behaviour of the humpback whale in southeastern Alaska. *Sci Rep Whales Res Inst, Tokoyo,*
**36**:41–47.

Dunlop R. 2019. Data from: Mitigations for the effects of vessels on baleen whale signaling
from a communication space perspective may not prevent behavioural changes. Dryad Digital
Repository. <http://doi.org/10.5061/dryad.65j00m3>.

Dunlop RA. 2016a. Changes in vocal parameters with social context in humpback whales:
considering the effect of bystanders. *Behav Ecol Sociobiol* **70**(6):857-870.

Dunlop RA. 2016b. The effect of vessel noise on humpback whale, *Megaptera*
*novaeangliae*, communication behaviour. *Anim Behav* **111**:13-21.
- Dunlop RA (2017) Potential motivational information encoded within humpback whale non-
song vocal sounds. *J Acoust Soc Amer* **141**(3):2204-2213.
- 10 538 Dunlop RA. 2018. The communication space of humpback whale social sounds in wind-
11 539 dominated noise. *J Acoust Soc Amer* **144**(2):540-551.
- 13 540 Dunlop RA. The communication space of humpback whale social sounds in vessel noise.
14 541 *Proceedings of Meetings on Acoustics*, In press, accepted 20/11/18.
- 17 542 Dunlop RA, Cato DH, Noad MJ. 2008. Non-song acoustic communication in migrating
18 543 humpback whales (*Megaptera novaeangliae*). *Mar Mamm Sci* **24**(3):613-629.
- 20 544 Dunlop RA, Cato DH, Noad MJ. 2010. Your attention please: increasing ambient noise
21 545 levels elicits a change in communication behaviour in humpback whales (*Megaptera*
22 546 *novaeangliae*). *Proc Roy Soc B* **277**(1693):2521-2529.
- Dunlop RA, Cato DH, Noad MJ. 2014. Evidence of a Lombard response in migrating
humpback whales (*Megaptera novaeangliae*). *J Acoust Soc Amer*, **136**(1):430-437.
- Dunlop RA, Noad MJ. 2016. The "risky" business of singing: tactical use of song during
joining by male humpback whales. *Behav Ecol Sociobiol* **70**(12):2149-2160.
- Dunlop, RA., Noad, MJ, Cato, DH, Stokes, D. 2007. The social vocalization repertoire of
east Australian migrating humpback whales (*Megaptera novaeangliae*). *J Acoust Soc Amer*
**122**(5): 2893-2905.
- Erbe C, Reichmuth C, Cunningham K, *et al.* 2016. Communication masking in marine
mammals: A review and research strategy. *Mar Poll Bull* **103**(1-2):15-38.
- Girola, E., Noad, MJ, Dunlop, RA, Cato DH. 2019. Source levels of humpback whales
decrease with frequency suggesting an air-filled resonator is used in sound production. *J*
*Acoust Soc Amer* **145**(2):869-880.
- Hatch LT, Clark CW, Van Parijs SM, *et al.* 2012. Quantifying Loss of Acoustic
Communication Space for Right Whales in and around a U.S. National Marine
Sanctuary. *Con Biol* **26**(6):983-994.
- Heiler J, Elwen SH, Kriesell HJ, Gridley T (2016) Changes in bottlenose dolphin whistle
parameters related to vessel presence, surface behaviour and group composition. *Anim Behav*
**117**:167-177.
- Janik VM. 2000. Source levels and the estimated active space of bottlenose dolphin
(*Tursiops truncatus*) whistles in the Moray Firth, Scotland. *J Comp Physiol A* **186**(7-8):673-
680.

Lesage V, Barrette C, Kingsley MCS, Sjare B (1999) The effect of vessel noise on the vocal
behavior of Belugas in the St. Lawrence River estuary, Canada. *Mar Mamm Sci* **15**(1):65-84.
McGregor PK (1993) Signaling in territorial systems – a context for individual
identification, ranging and eavesdropping. *Philosophical Transactions of the Royal Society*
*of London Series B-Biological Sciences* **340**(1292):237-244.
McGregor PK, Peake TM, Lampe HM (2001) Fighting fish *Betta splendens* extract relative
information from apparent interactions: what happens when what you see is not what you
get. *Anim Behav* **62**:1059-1065.
Miller PJO. 2006. Diversity in sound pressure levels and estimated active space of resident
killer whale vocalizations. *J Comp Physiol A* **192**(5):449-459.
Miller PJO, Biassoni N, Samuels A, Tyack P L (2000) Whale songs lengthen in response to
sonar. *Nat* **405**(6789):903-903.
Noad MJ, Cato DH, Stokes MD (2004) Acoustic tracking of humpback whales: measuring
interactions with the acoustic environment. Proceedings of Acoustics 2004, Annual
Conference of the Australian Acoustical Society, Gold Coast, 3-5 November 2004, pp 353–
358.
Noad MJ, Dunlop RA, Kniest E. Boom to bust? Implications for the continued rapid growth
of the eastern Australian humpback whale population despite recovery. *Pop Ecol* **61**(2):198-
209.
Otter K, McGregor PK, Terry AMR, Burford FRL, Peake, TM, Dabelsteen T (1999) Do
female great tits (*Parus major*) assess males by eavesdropping? A field study using
interactive song playback. *Proc Roy Soc B* **266**(1426):1305-1309.
Peake TM, Terry AMR, McGregor PK, Dabelsteen T (2001) Male great tits eavesdrop on
simulated male-to-male vocal interactions. *Proc Roy Soc B* **268**(1472):1183-1187.
Parks SE, Clark CW, Tyack PL (2007) Short- and long-term changes in right whale calling
behavior: The potential effects of noise on acoustic communication. *J Acoust Soc Amer*
**122**(6):3725-3731.
Parks SE, Cusano DA, Stimpert AK, Weinrich MT, Friedlaender AS, Wiley DN (2014)
Evidence for acoustic communication among bottom foraging humpback whales. *Sci Rep* **4**.
Parks SE, Johnson M, Nowacek D, *et al.* 2011. Individual right whales call louder in
increased environmental noise. *Biol Letters* **7**(1):33-35.
Rekdahl ML, Dunlop RA, Goldizen AW, Garland EC, Biassoni N, Miller P, Noad MJ
(2015) Non-song social call bouts of migrating humpback whales. *J Acoust Soc Amer*
**137**(6):3042-3053.
Sharpe FA, Dill LM, Beaver V, Spellman B. (1998) Killing me softly: feeding calls of the
Alaskan humpback whale. Abstracts of the World Marine Mammal Science Conference,
Monaco, January, pp 20 – 24.

Silber GK (1986) The relationship of social vocalizations to surface behavior and aggression
in the Hawaiian humpback whale (*Megaptera novaeangliae*). *Can J Zoo* **64**(10):2075-2080.
Tennessen JB and Parks SE. 2016. Acoustic propagation modeling indicates vocal
compensation in noise improves communication range for North Atlantic right
whales. *Endanger Species Res* **30**:225-237.
Tervo OM, Christoffersen MF, Simon M, *et al.* 2012. High Source Levels and Small Active
Space of High-Pitched Song in Bowhead Whales (*Balaena mysticetus*). *Plos One* **7**(12).
Thompson POW, Cummings WC, Ha SJ (1986) Sounds, source levels and associated
behavior of humpback whales, Southeast Alaska. *J Acoust Soc Amer* **80**(3):735 - 740.
Wiley RH, Richards DG. 1978. Physical constraints on acoustic communication in
atmosphere – implications for evolution of animal vocalizations. *Behav Ecol Sociobiol*
**3**(1):69-94.

Fig. 1. The relationship of the communication space of humpback whale low- (a) and high-
(b) frequency signals in wind-dominated and vessel noise. Smoothing functions are from the
generalised estimate equation outputs (Table S1) for the $rSNR = 0$ relationship (from Figs. S1
and S2). Raw data-points at which the signal $rSNR$ at the array was between 1 and -1 are also
included.

Fig. 2. The probability of the signalling group interacting with a neighbouring group as a
function of the distance of the neighbour from the signaller. Relationships are shown for the
wind and vessel noise environments and include 95% confidence intervals (a) and separated
according to the neighbour position relative to the signalling group (b and c in wind and
vessel noise respectively).

Fig. 3. The modelled relationship, and significance of the predictor variables, of the $rSNR$ of
humpback whale low- (a) and high- (b) frequency signals in wind and vessel-dominated noise
received noise. Data-points from signalling whales between 900 and 1100 m (approximately
1 km) from the array are included.

Mitigating for the effect of vessels on baleen whale signaling from a communication space perspective may not prevent behavioral changes

Letter of response

Associate editor

First, there is a lack of information throughout regarding the acoustic modeling that does not allow the reader to fully understand what was done and if it was appropriate.

Response: this has been addressed in response to reviewer 1 (see below). Note a site-specific empirical measures were used to generate the acoustic model and more details are now included (though noting journal space limitations we attempted to balance the detail included with referring to this in previously published works).

Second, there needs to be more information provided in the main text that is currently in the supplemental material or simply referenced from previous work. I would consider this an opportunity to expand on the writing to make it more broadly consumable to an audience that is not as intimately familiar with acoustics and the literature.

Response: The paper has been further focussed towards a goal that relates to an audience interested in animal communication networks and the effects of noise. This following statement has been added to both the intro and discussion:

This goal of this study was to define the communication space and network of signalling humpback whales and determine if, and how, this was reduced during periods of vessel noise.

The introduction has been edited following comments by reviewer 2. Here, more is included about the relationship between communication space, network, and masking effects. This puts the work into the perspective of animal communication (communication space and networks) as well as the effects of noise on this aspect of communication. This was further re-iterated in the last paragraph of the introduction to illustrate how the study informs current knowledge on animal communication networks and changes in these networks due to noise (using humpback whales as the target species).

The methods were expanded as per reviewer 2 to further explain concepts to an audience not familiar with acoustics. To follow this through the results, more references are made back to communication networks and the implications these results have on the size of the communication network of humpback whales.

Finally, in the discussion, this goal has been reiterated and applicable results discussed. The discussion has been revised and re-ordered. Any previous work referenced has had additional sentences added to further explain results to non-acoustic readers.

To address the material presented in supplementary. This is the background modelling used to generate the outputs for the 0 SNR relationship between noise and distance. Including this in the main paper would add a substantial amount of background analysis which may detract, rather than add to how consumable the paper is to a broad audience. To help with this, I have added some more information on the results of this modelling, rather than the full modelling procedure. This should negate readers having to go to the supplementary results unless interested.

The behavioural data presented in the supplementary material is mainly background information on
dealing with various behavioural scenarios. As above, more information on this has been included in
the main manuscript to balance overloading the readers with background analyses versus providing
enough information that the supplementary materials do not need to be consulted unless
interested.

11 12 **Reviewer 1**

There are many cases where the Supplemental material is referenced, as well as previously
published Dunlop papers (e.g. Dunlop et al 2013, 2016, 2018), where a summary or summary figure
would help provide enough info to the reader of this paper to continue on without having to go back
and read other papers to understand this one. These items are pointed out in the detailed
comments below.

**Response:** addressed as per the specific comments below

Second, the word model and modelling is used throughout the paper to talk about acoustic
modelling, communication space modelling, and statistical modelling. Some care to better
differentiate between the difference types of modelling with qualifiers or descriptor would be
helpful.

**Response:** the terms suggested by the reviewer have been used throughout the paper to
differentiate.

One technical item that requires addressing concerns the acoustic modelling. What model was used
for TL? I'm assuming spherical spreading because a circular communication space is referenced to in
many places of the paper. The acoustic model was not explicitly stated. Use of a range dependent
model would be more applicable in this case (if spherical spreading was indeed used), as range
dependent bathymetry is crucial and would not produce circular communication spaces. I'm not
advocating for a reworking of the data with a new model, but the details of what was used and its
assumptions and limitations should be addressed.

**Response:** a site-specific empirically-derived transmission loss model was used. This model was
developed and published in Dunlop et al. 2013, and has been used in many other studies at the site.
As per the comments below, more detailed information about this model is included, though only
briefly, as the author is aware of taking up journal space with material already published.

Detailed comments:

Title – I found the title wording and not straightforward. Suggest rewording. Possible suggestion “Is
mitigating for the effects of vessel noise on baleen whale communication effective?”

**Response:** changed to

**The effects of vessel noise on the communication network of humpback whales**

Line 15 – Suggest “coastlines, they”

**Response:** done

Line 16 – Suggest “Since there are no empirical data”, since data are plural.

**Response:** done

Lines 22-23 – The hearing information is not the emphasis or focus of the analysis. Suggest removing
it from the abstract.

**Response:** done

Line 26 – Suggest “additional confounding effect of”

**Response:** done

Line 53 – Duplicate “to compensate”.

**Response:** done

Line 83 – Suggest “condition, and the distance”

**Response:** done

Line 89 – Suggest “whales’ or whale’s”

**Response:** done

Line 93 – Can you clarify whether the receiver in this line of text refers to the intended receiver,
unintended receivers, or both? The assumptions made for each may be different for the signaler and
should be addressed.

**Response:** this refers to any generic receiver (in this case the quantification of SNR at a hydrophone
receiver, which then could be used to inform SNRs at a generic receiver within the communication
space. Changed ‘the’ to ‘a’.

Lines 111-121 – Positioning of individual vessel transits were discussed in this paragraph and linked
to their contribution of vessel noise. How have you dealt with long distance shipping coming in from
distant, unobserved ships? This would be a component of the low frequency noise more than the
high-frequency component. Was this included in the “Wind Only” measurements and modelling?
Does “Wind Only” just really mean the absence of close transiting vessels?

**Response:** Previous papers in this study site termed wind-only noise, wind-dominated as, given the
tight correlation with wind speed (even at low frequencies), and the fact that the offshore shipping
lane is within view and ships are only present intermittently. We cannot fully exclude a noise
contribution from offshore shipping, but we are confident that the wind noise measures are wind
dominated due to results of Dunlop et al. 2010. The section here relates only to fishing vessels
traversing the areas. A sentence was added later in the methods to address this comment:

Wind-dependent noise was measured when there were no audible and/or visually tracked
vessels in the area (including shipping vessels in the within-sight shipping channel). A
previous study at this site found the measured noise in the absence of singer and vessel noise
similar to that generally observed for wind-dependent noise in Australian waters (Cato 1996)

and found wind speed to be significantly correlated with noise levels in all third octave
frequency bands (Dunlop et al. 2010).

Line 127 – What is “These” referring to? The social sounds detected on only 3 of 5 hydrophones, or
did you select the best arrivals from 3 of 5 hydrophones?

**Response:** added in the word ‘hydrophone’

Line 140 – Where did the background noise measurement come from? 1-sec preceding the call? At
another time related to the call? Did it include other background song?

**Response:** specified it was taken just before, or just after, the sound. Also included the sentence
“This sample included background singing only if this sound source was also present within the social
sound sample.” to clarify

Lines 140-141 – Confused on what this sentence is trying to convey? Please clarify.

**Response:** added in ‘the’ to clarify

Line 144 – Please clarify the use of DFT and FFT in the same sentence.

**Response:** changed and clarified with a reference - From the wavfile, an FFT was used to
calculate the frequency content of the sound (FFT size of 16 384, 1.35 Hz resolution, 50%
overlap) and its power spectrum extracted. The power spectrum was calibrated to account for
the sensitivity of the hydrophone and the gain of the system (Girola et al. 2019).

Line 149 – Not enough information is given about the TL modelling here. A summary from Dunlop et
al (2013) would be helpful. What TL model was used?

**Response:** added in some more details but as stated in the opening remarks, the author was
cognisant of taking up journal space for something that has been previously published. Information
added:

The boat conducted runs along lines radiating from the array, from distances of 100m out to
about 10 km. The boat speed was kept constant. Wind speed for that day ranged from 10 to
15 knots, swell was less than 1m, and there were no significant currents in the area.

The value of a may have varied with the direction of the boat (approaching or going away
from the receivers) as the noise radiated forward likely differed from the noise radiated aft.
Even so, for most frequencies, b varied with distance but could be well approximated by two
values; one applying to distances less than, and the other greater than, a cross over value.
Absolute values of TL were determined by measuring received levels of octave band limited
noise, projected using a J11 source suspended from a boat (estimating the source level using a
hydrophone suspended at a distance of 3 m and corrected to 1 m assuming spherical
spreading). This source was projected at three distances (between 200 and 1000 m) from the
array. The trend in loss, $b \log(x)$, from the boat runs was then fitted to the absolute values of
loss from the J11 measurements to determine the value of a for each octave band, by
minimising the sum of the squares of the differences between $a + b \log(x)$ and the data points
from the J11 measurements.

Lines 160-161 – When did these 10-min files occur in relation to the signals? On the same day, within
the same hour?

**Response:** yes, added in a line to clarify. "As recordings were divided into 10-minute files for storage,
the file containing the social sound of interest was used for the background noise measurement."

Line 171 – Suggest "hydrophone used to measure"

**Response:** done

Line 176 – What is Dunlop (31)? It would be helpful to see a distribution or histogram for the 1/3
octave bands to identify the dominant band.

**Response:** should be Dunlop et al., 2013. Changed. Did not add in a histogram as there is no one
dominant band. Each sound type had a different dominant 1/3 band.

Line 181 – It would be helpful to summarize the info related to frequency category definitions from
Dunlop (2018). At least provide the frequency range of low and high categories used in this paper.

**Response:** done

Line 185 – Which models are you referring to? Statistical, acoustic, propagation, overall
communication space? This should be clarified for many uses of "model" or "modelling" throughout
the text.

**Response:** all models referred to are statistical. Clarified.

Line 202 – Do you broadband noise or background ambient ocean sound?

**Response:** broadband as stated

Line 220 – More information is needed on why you are assuming a circular area around the signaler?
Was this because you used spherical spreading? If so, the assumption of a circular area is most likely
not a realistic assumption given the bathymetry, and this should be explicitly stated.

**Response:** agreed, but this was to generate the most simplistic representation of communication
space and one that is likely to be used if no other info is available. Added in sentence to clarify.

Line 222 – Suggest "randomly distributed receiver groups" if that is what is intended.

**Response:** done

Line 224 – Missing the link here between the communication model results and the number of
groups left in the area. Is this related to behavioral observations?

**Response:** clarified as loss of radial distance

Lines 232-234 – A figure showing the 2 groups for comparison would be helpful.

**Response:** not included due to journal space and duplicity. The first paragraph of results (next
paragraph) details what would be in the figure.

Line 381 – The word compared implies a direct comparison, but there is no comparison in this
sentence.

Suggest “The loss of total communication area in vessel noise and wind-dominated noise was
approximately 75% for both signal types.”

**Response:** this is not what the sentence meant, communication area was 75% less in vessel noise as
compared to wind noise. Re-worded to make clearer.

Line 395 – Can you please verify “by” or “to”?

**Response:** verified as ‘by’, not changed

Line 443 – circular area from signaler is again an assumption that warrants clarification.

**Response:** done

Reviewer 2

This paper promises to present a worth-while discussion, taking the organism perspective in how
increases in ambient noise, driven by vessel noise and wind noise, may change behaviour in
vocalising and/or interactions between humpback groups. What it does really is just describe the
extent of the communication space based on interactions between groups of whales based on
different ambient noise regimes. I can’t help but think that more can be made from the data that
was collected, both acoustic and observational, which would enlighten the discussion further, and
more to what was promised from the title/abstract.

**Response:** the author respectfully disagrees with this statement. It uses measured received levels of
social sounds, along with an empirical propagation model, to estimate at communication range
based on the assumption of SNR = 0 to delimit the range. In then uses interactions between groups
of whales (given that is the likely to be at least one of the functions of the social sounds) to **inform**
this estimation of communication range (i.e. these interactions are not used to describe). In all cited
references, there is no inclusion of behavioural data to inform estimated communication space.

It seems like a lot of material is trying to be drawn from the data presented: that baleen whale
hearing capacity might be better determined;

**Response:** removed as recommended by reviewer 1

that masking might be discussed in context of communication space, and that the implication of
masking on behavioural interactions may be determined.

**Response:** included as per the comment below, where focus to one of them was given

More focus given to one of these may help with streamlining the paper and making it more directed.

Highlighting the decreased communication space and how that has wider implications in conspecific
interaction I think is the best way to go here, and is where the author seems to take it but there
needs to be some more definitive statements of the study question and findings. As presented the
data and analysis would not answer this question.

**Response:** addressed within comments further down

The use of visual data in conjunction with acoustic recordings can afford some very interesting
results and a strength to the conclusions not possible from acoustics alone. A little more could be
drawn out the significance of using these to different types of datasets together. There was a lot left
to the reader to follow up on (e.g. social calling/vocalisation types, behavioural categorisation) or
that is not clearly defined (like wind dominated ambient noise) or explained (some of the analysis –
see comments).

**Response:** addressed within comments further down

Missing from the discussion is the implications of taking an individual-centric or communication
space approach, and how that might play into understanding of behaviour or mitigation actions. Also
it could lead to a discussion on how this approach may lead to different/better management actions
than are currently in place, but did not.

**Response:** the problem with this is that the study did not take an individual-centric approach,
therefore it cannot state how this approach would lead to different management actions. It also
cannot, as stated in the discussion, say if these effects would lead to any population level effects,
without much further study.

The paper presents a lot of material, but it feels piecemeal, and like a lot of different pieces of data
forced together currently, with it lacking a main focus or directive that is clearly stated to being
about cohesiveness.

**Response:**

The paper would benefit from proof-reading and going over to make it more concise, focusing on the
follow on of one paragraph to another and the general message to the reader. It doesn't have an
easy flow to the material.

**Response:** addressed within comments further down

Title: suggest rewording the title. The paper presents discussion from vessel and wind noise The
short title almost works better as a descriptor of the material presented (masking of humpback
whale signals) as the masking effect is where this paper focuses and how that effects the space over
which whales interact.

**Response:** done, title suggested by reviewer 1

Abstract

Reads a little awkwardly, some rewording may be helpful here with the flow.

Introduction:

I find the introduction to lack flow within and between sections. There seems to be too much
covered. If the how communication space for social sounds, as modeled, in different soundscape
settings (wind or vessel noise) affects social interaction between whales became the clear focus
some of the less relevant/irrelevant material could be removed and make the section more concise
and easier to follow.

Lines 32-42: Don't think this is relevant to the discussion. That there is an acoustic communication
network and this network is defined by the propagation of each call type in ambient noise conditions
if the underlying basis of this paper. Then go on to discuss the conditions in ambient noise, and
additions to the soundscape (both natural and anthropogenic) that might alter this.

A description of sources of noise that comprise the full soundscape may also be worthwhile to the
reader too, perhaps describing the differences in natural (geophonic) sources and human-derived
(anthrophonic) noise sources.

**Response:** it is important that non-cetacean readers understand that the communication network in
cetaceans, like humpbacks, is likely to be large. This provides context in that a reduction in
communication space, is likely to reduce their communication network, which then will have
implications on social behaviour (like those found here). To make it flow better with the second
paragraph (comment below), have added in linking sentences to the second paragraph (linking to
the first) and re-arranged the second paragraph as per the comment below. In addition, the
associate editor recommended making this paper readable to a wider audience, therefore animal
communication networks have been left in.

Again in the second paragraph (lines 45-60) – discuss masking first, and its implications on
communication, and then what baleen whales have been noted to do to compensate for increased
ambient noise. For this paper, as it references so many previous studies the most succinct way to
introduce material, without losing the reader, would be to use these previous works as a jumping off
point, i.e we saw compensation mechanisms in increased wind noise and not in vessel noise,
knowing this we interpret the reduction of communication space to have implications on socialising
behaviours (the focus of the study).

**Response:** done

Maybe somewhere explicitly say that you are using communication space as the distance that
conspecifics can send and receive acoustic messages, and where there is a observable (social)
response in the receiving whale.

**Response:** done

Line 64: may be nice to describe briefly under what conditions the propagation distances were
calculated. Is there a general description for social sounds (in frequency, length, modulation etc.).
Some more details here would likely help the reader.

**Response:** done. Included the fact that the environment was a shallow-water environment (<50 m),
plus added in:

Their repertoire of social vocalisations is diverse, spans a fundamental frequency range of less than
40 Hz to to over 2 kHz and includes sound types ranging from low-frequency “grumbles”, to high-
frequency bird-like “chirps” as well as both modulated (e.g. “moans”, “trumpets” and “cries”),
amplitude-modulated (e.g. “purrs” and “growls”), and broadband (e.g. “roars” and “underwater
blows”) sounds (Dunlop et al., 2017).

Lines 82-87: some of this detail may be better in the methods/ is in the methods and doesn't need
repeating

**Response:** respectfully disagree, this setting up the background to the work, that we rely on
modelling the communication space because we have not empirical data on receiver hearing

Line 89: 'will modelled and compared' should be 'will be modeled and compared to conditions of...'

**Response:** done

Materials and Methods

There are some details missing that would help the reader, for example, is there a baseline or
'background' noise level defined.

**Response:** this was already included in the manuscript

These varied from approximately 90 dB *re* 1 μ Pa to 112 dB *re* 1 μ Pa in wind speeds ranging from <5
knots to approximately 15 knots. The median wind-dominated noise was 100 dB *re* 1 μ Pa for this
area.

And further down:

Measured noise levels were above 100 dB *re* 1 μ Pa with mean and modal measured levels being 105
26 dB *re* 1 μ Pa (vessels were approximately 4 km from the array).

What is average wind speed/noise, and what is the definition of 'wind dominated'.

**Response:** added but wind speed was already included

What was the process of comparing the visual and the acoustic data?

**Response:** this has been covered in the manuscript though a sentence has been added in to make
clearer the link between the visual and acoustic data (highlighted)

Acoustic recordings were made of groups of migrating (from their breeding ground in the Great
Barrier Reef, towards their Antarctic feeding grounds) of humpback whales (September/October of
2002 – 2004 and 2008). The array comprised of five hydrophone-buoy systems anchored in 20 – 28
42 m of water in a 'T' configuration. Buoys 1 to 3 were anchored in a line 1.5 km offshore, with buoys 4
and 5 extended a further 1 km offshore. Each system included a High Tech HTI-96-MIN hydrophone
with built-in +40 dB pre-amplifier, a pre-amplifier (+20 dB) and VHF radio transmitter. Real-time
radio transmissions from the buoys were received a base station using a Yagi antenna attached to a
four channel, low noise, VHF receiver (type 8101) and a Winradio receiver. Sound source locations
were calculated by *Ishmael* where the mean position of several estimates were calculated over a
brief period. A shore-based theodolite survey of the surface buoys was used to obtain accurate
positions. For detailed validation experiments see Noad et al. (2004).

Groups were recorded in water depths ranging from 25 to 35 m. Simultaneous land-based tracking
of the groups occurred to obtain visual data of migrating humpback whale groups within a 10 km
offshore radius from the acoustic array. All groups were tracked regardless of whether or not they
were vocalising. Group positions were recorded using a theodolite linked to *Cyclopes* (developed by
Eric Kniest, University of Newcastle) and additional observations on group social composition
(number of adults within the group and whether or not there was a calf) and social behaviour
(splitting of animals from, and joining of animals to, the group) were recorded at each position. The

position of traversing vessels was also recorded by the land-based observation team. The visual and
acoustic positions were overlaid onto the one Cyclopes file, meaning the acoustic tracking could be
matched to groups being visually tracked. Using this combination of visual and acoustic tracking
data, the distance of each signalling group to each hydrophone-buoy receiver was measured for
each recorded social sound.

Line 106: Should read were received at a base station

**Response:** done

Line 111: Were groups only tracked from shore?

**Response:** yes

Line 113: Be sure it is clear: vocalisations are localised to a position using the array, the calls are then
attributed to a group using visual recordings, the group positions are tracked using theodolite.

**Response:** edited to make clear

Are all call types used, or just social calls. Are there social and travel calls?

**Response:** we have don't know the function of these calls so prefer not to say

Line 115 onwards: more detail could be given on the theodolite tracking methods.

**Response:** some further details added

Line 118: At what interval were these positions taken? Is this in time or space?

**Response:** at each surfacing, added

Line 120: were the visual and acoustic tracks compared? Were there groups that did not vocalise?
Under what conditions may this have happened?

**Response:** sentence added in to address this (see highlighted sentence above)

I assume that the comparisons between visual and acoustic are limited to day light hours, but this is
not specifically stated.

**Response:** sentence added

Were the comparisons limited to only social calls when you made them? The points made in Lines
119-121 need to be highlighted and more specifically described.

**Response:** done

Is the distance between the whales and vessels also calculated?

**Response:** not for this study, a vessel just had to be in the area, also, the noise was measured at the
array, not the whale

Line 124: A better description of wind noise would be useful to the reader. Is this over a particular
wind speed or is it an increased ambient noise level when vessels are not present.

**Response:** done

Line 125: Is there a specific SNR/dB level for 'first becomes audible'. I presume this is derived
manually through inspection of spectrograms.

**Response:** yes and added

Line 127-129: Unclear

**Response:** changed to make clearer

Line 131: Is this distance from the array and within the hydrophones receiving the calls in the array a
means to derive propagation distances?

**Response:** yes, but with the caveat the SNR = 0 defines the distance (as stated in the intro)

Could the visual tracks and distance from the array and the 'first becomes audible/now not audible'
be used somehow in discussing the likely communication space under each ambient noise scenario.

**Response:** yes, but this equates to an approximate SNR of 0 so the answer would be the same

Maybe take a sentence or two after line 136 to say what these technical specifications mean in
terms of the calls received and how they might affect the reception and interpretation of calls to the
reader.

**Response:** done

Line 139: more description or reference to social sound types is needed here.

**Response:** these were described in the intro (in response to reviewer 1) and further details provided
in the opening lines of the communication space section

Line 140: How was the sample of background taken/selected.

**Response:** not sure what the reviewer is referring to here?

Line 161: How were the files selected? What is deemed as 'wind-dominated' noise. Is wind
dominated purely vessel absent?

**Response:** more details added in response to reviewer 1

Line 165: What is the wind speed at the average wind speed noise.

**Response:** added

Line 171: used to measure the signal

**Response:** changed

Line 175: How does using the 1/3 octave band affect the interpretation compared to, say, using the
peak frequency when the SNR was at its greatest.

**Response:** the 1/3 octave band would contain the SNR peak frequency and it is standard now to
measure humpback sounds using a 1/3 octave band (added this in to make clear)

Line 176, Line 184: make sure referencing is complete, consistent and in line with the journal

**Response:** done

Lines 184-195: Much of this seems to be information presented elsewhere. Be sure when describing
this that you are, then, adding something of note to the initial analysis. Be sure that that the reader
is clear that the rSNR you reference is the SNR of a call heard by a receiver

**Response:** the author disagrees. The addition is using the SNR = 0 to statistically define the
communication space which hasn't been mentioned previous to this. Also not made clear as the
receiver, in this case, is the array, not the whale – added in statement further up to clarify.

Line 194-195: Not sure what this means

**Response:** changed

Line 196: run not ran

**Response:** done

Line 201: 'R'

**Response:** done

Lines 219: Could you draw on your observations here with the inter-group communications rather
than randomly placed groups?

**Response:** the problem with this is that we do not know which group is the intended receiver, versus
unintended. Plus, this line refers to randomly place simulated groups therefore we have no inter-
group communications to comment on?

Line 230-234: Why combine the two noise environments here, especially as the behavioural
response to these noise additions have previously been seen to be different.

**Response:** as stated, for this model, to re-create the distribution of the other groups relative to the
receiver, there was no significant difference between the group distribution in wind noise compared
to vessel noise. This analysis is only to generate a robust representation of group distribution, not to
determine if there are differences (which we know there are not) in distribution between the 2
environments.

Results:

Social, Line 240-245: Is this from the observational or modelled data?

**Response:** clarified as observational

Lines 256-261 – perhaps describe ‘typical’ wind and vessel noise and how this overlaps with typical
social calls

**Response:** this was already described in the first sentence of the paragraph

Lines 385-398: Could it be that the low-frequency communication space/low frequency
calls/communications are less affected by wind/vessel noise

**Response:** no, low-frequency calls are generally louder so propagate further. Clarified

Is there any difference seen in the calls used/calling behaviours? Were the previous assertions for
call/behavioural modifications due to increased ambient noise confirmed in this study?

**Response:** no, this was tested in Dunlop 2016b (in terms of switching from vocal to surface-
generated) but not here. This manuscript is focussed on communication space in terms of signal
level, rather than sounds used and I argue that there is enough content in here without further
analyses. In response to the second point, this has been covered in the discussion.

Discussion

This discussion again could do with more focus and linking back to the research question. Work on
the flow of the material presented. The discussion material should show a stronger link back to the
results of this study.

**Response:** changed as suggested. The first two paragraphs now re-iterate the original goals of the
study (as found in the last paragraph of the intro), and re-iterate and then the discuss the results
which address these goals.

Line 426: Typical wind conditions or typical noise conditions – and what percentage of the time of
recording does this represent?

**Response:** wind noise conditions as stated, these were defined in the results. Unclear what the
reviewer means by the second comment, wind speeds were unlikely to change over the 20 to 80
minute recording, so this would be the whole recording?

Line 425-430: Even the discussion between high and low frequency communication space – and be
sure to tie it back to humpback whale communications.

**Response:** unclear what the reviewer means by this. The acoustic communication space and
matching behavioural observations were stated for both low and high frequency sounds so the
discussion was even. To address the second point, we can only hypothesise how these sounds are
used within their communication. Added in two sentences to speculate on the function of each of
these sound groups as well as sentence to state that further work is needed to clarify their function.

Line 453-455: Again I think it might be possible to draw more out of the observational data than just
the random distribution model.

**Response:** included more discussion on the behavioural data as compared to the random model and
the receiver distribution model.

Line 473-474: Develop this point further

**Response:** done - sentence added below

Behavioural interactions were most likely to occur when groups were separated by 1 km or less
where most signals, even high-frequency signals, were still well above received vessel noise and
unlikely to be masked.

~~Mitigating for the~~ **The effects of vessels** vessel noise ~~on baleen whale signalling from a~~
the communication space perspective may not prevent behavioural changes network of
humpback whales

Rebecca A Dunlop

*Cetacean Ecology and Acoustics Laboratory, School of Veterinary Science, University of*

*Queensland, Gatton Campus, Queensland, QLD 4343, Australia*

Electronic mail: *r.dunlop@uq.edu.au*

Short title: Masking of humpback whale signals

Humpback whales likely rely on social sounds to mediate social interactions. The distance to
which these social signals propagate from the signaller will define its communication space,
and therefore communication network (number of receivers). As humpback whales migrate
along populated coastlines, they are likely to encounter noise from vessel traffic. Since ~~there~~
~~is~~ no empirical data exists on baleen whale hearing, the consequences of this are usually
assumed, being the modelled reduction in their communication space. Here, the
communication space and network of migrating humpback whales was compared in
increasing wind-dominated and vessel noise ~~to determine the effects of signal masking.~~
Behavioural data on their social interactions were then used to inform these models. In
typical wind noise, a signaller's communication space was ~~modelled at~~
approximately estimated to extend to 4 km, which agreed with the observed extent of their
social interactions. ~~This result provides novel information on the hearing capabilities of~~
~~baleen whales in noise.~~ An increase in vessel noise reduced the modelled communication area
by a factor of four. ~~Within that space, the likelihood of,~~ with a significant reduction in group
social interactions ~~also significantly decreased implying a reduction in their communication~~
network. However, signal masking did not fully explain this change in social behaviour,
implying there was also an additional effect of the physical presence of the vessel on ~~both~~
signaller and receiver behaviour. Though these observed changes in communication space
and social behaviour were likely to be short-term and localised, an increase in vessel activity
due to tourism and coastal population growth may cause more sustained changes along the
humpback whale migration paths.

**Introduction**

Vocal communication networks involve a signaller, one or more intended receivers, and one
or more ‘unintended receivers’. Unintended receivers, referred to as the ‘audience’, as
‘bystanders’, or as ‘eavesdroppers’ (McGregor, 1993; McGregor et al., 2001; Otter et al.,
1999; Peake et al., 2001), can gather and use information from this signalling interaction to
their advantage. The size of this communication network is usually defined as the distance
radius from the signaller at which any other receiver can detect and interpret the signal. This
depends on the signal level and structure, the propagation environment, levels of background
noise, and the receiver’s ability to detect, recognise and discriminate the signal within noise
(Wiley and Richards, 1978). In the marine environment, as sound propagates over large
distances, cetacean communication networks can be extensive and involve multiple intended
and unintended receivers. Cetaceans are likely to utilise these networks within feeding (e.g.
Janik, 2000; Miller, 2006) and breeding interactions (e.g. Dunlop, 2016a; Dunlop and Noad,
2016), meaning effective communication exchanges are essential for these life functions.

~~Masking is the process by which the receiver’s threshold of hearing of a signal is~~
~~raised by a masking sound (noise). Increased masking can reduce the signaller’s~~
~~communication space.~~In the underwater environment there are several natural sources of
noise, such as noise from surface breaking waves during periods of high wind. ~~Though this~~
~~increase in natural noise has~~Noise sources can also be anthropogenic in origin, such as from
vessel activity, oil and gas exploration, naval sonar activity, or construction. These noise
sources have the potential to mask signals, meaning they become inaudible to receivers
within the network as the noise levels go up. This will reduce the size of the communication
network and, in turn, may have detrimental effects on feeding or breeding interactions within
these networks. Important data for predicting the potential impacts of masking include, for
the receiver, basic audiograms, critical ratios (CRs) of hearing, and identification of

mechanisms that allow receivers some release from masking (Erbe et al., cetacean
 communication signals, they have evolved various anti-2016). Little is known, however,
 hearing capabilities in baleen whales therefore it is difficult to predict how receiving whales
 cope with increased noise (Erbe et al., 2016). For the signaller, important data for predicting
 masking includes which anti-masking strategies are used (Erbe et al., 2016). strategies.
 BaleenSignalling whales, for example, increaseuse strategies such as increasing their signal
 level (Dunlop et al., 2014; Parks et al., 2011), changechanging their signalling behaviour
 (Dunlop et al., 2010), and changechanging the frequency content (Heiler et al., 2016; Lesage
 et al., 1999; Parks et al., 2007) and duration of thetheir signal (Miller et al., 2000) to
 compensate to compensate. It is likely, however, that there is a limit to the success of these
 strategies. If the noise source is anthropogenic in origin, such as from vessel activity, oil and
 gas exploration, naval sonar activity, or construction,for example, and noise levels are above
 natural noise levels., signaller anti-masking strategies may not be as effective. Therefore, the
 receiver's ability to detect and interpret relative sounds may be reduced over and above that
 in natural noise.One way for the receiver to compensate is to focus its attention on the signal
 and audibly separate it from the masking noise, known as masking release. Little is known,
 however, about this strategy in cetacean receivers (Erbe et al., 2016) given that little is known
 about their hearing capabilities., with a higher risk of loss of communication space and
 reduction in communication network.

Acoustic communication in humpback whales (*Megaptera novaeangliae*) has been
 extensively studied making this a good model species for cetacean masking studies. The
 communication space of social sounds ('vocalisations', blow-hole associated sounds, and
 percussive signals generated at the surface) is in the order of km's (Dunlop, 2018) in a
 shallow-water environment (<50 m) and includes a network of intended and unintended
 receivers (Dunlop, 2016a; Dunlop and Noad, 2016). SœialTheir repertoire of social

vocalisations is diverse, spans a fundamental frequency range of less than 40 Hz to to over 2
88 kHz and includes sound types ranging from low-frequency “grumbles”, to high-frequency
bird-like “chirps” as well as both modulated (e.g. “moans”, “trumpets” and “cries”),
amplitude-modulated (e.g. “purrs” and “growls”), and broadband (e.g. “roars” and
“underwater blows”) sounds (Dunlop et al., 2007). These social sounds are commonly heard
during breeding, (Dunlop, 2016a; Dunlop and Noad, 2016; Rekdahl et al., 2015; Silber, 1986)
and feeding, group interactions (D’Vincent et al., 1985; Parks et al., 2014; Sharpe et al.,
1998; Thompson et al., 1986) and likely function to provide information on group
membership, changes in group membership, signaller sex, size, location (Dunlop et al., 2008)
and motivation (Dunlop, 2017). Previous studies found that the communication space of
humpback whale signallers ~~compensate for~~ extends to approximately 4 km (Dunlop, 2018)
and that signallers maintain this space in increased wind-dominated noise by both switching
from vocal sounds to surface-generated sounds (Dunlop et al., 2010) and increasing their
vocal source level (Dunlop et al., 2014). In Interestingly, in response to increased vessel-
dominated noise, there was no evidence that the signaller used these anti-masking strategies
(Dunlop, 2016b) implying). This implies that when vessels are traversing the signaller’s area,
the communication space ~~was~~ of a humpback whale signaller is significantly reduced. The
consequences of this within the context of breeding interactions, if any, remain unknown.
~~Important data for predicting the potential impacts of masking include, for the~~
~~receiver, basic audiograms, critical ratios (CRs) of hearing, and identification of mechanisms~~
~~that allow receivers some release from masking (Erbe et al., 2016). For the signaller,~~
~~important information includes which anti-masking strategies are used (Erbe et al., 2016). In~~
baleen whales, apart from anatomical modelling, there is no empirical data on their hearing in
noise. Therefore, if predicting the impact of masking on the receiver, the signal propagation
is usually modelled in various noise conditions and the distance at which the signal

disappears below noise is assumed to equate to the receiver's audible limits. In other words,
an assumption of receiver hearing in noise, and/or the receivers's ability to recognise and
discriminate the signal must be made. This Cetacean communication space is then
used therefore usually modelled using a presumed limit in receiver audibility (e.g. Tennesen
and Parks, 2016), and in some cases, a presumed ability of the receiver to define the
recognise the signal (e.g. Cholewiak et al., 2018; Hatch et al., 2016). These studies assume
the distance at which the measured received level of a signal-to-noise ratio, or signal excess,
is of a certain value (e.g. 0 dB) delimits the signaller's communication space distance. This
study will follow the approach outlined above to define the communication space of
signalling humpback whales but will use behavioural observations to further inform results.
Firstly Therefore, in this study, communication space is defined as distance that conspecifics
can potentially send and receive acoustic signals, and where there is an observable (social)
response in the receiving whale. To test for effects of anthropogenic noise on this space,
firstly, the likely extent of a signalling whales whale's communication space will be modelled
and compared in natural (wind) and anthropogenic (vessel) noise. Second, the study will
assess the behavioural consequences of increased vessel noise within the signaller's local
neighbourhood; (communication network), in terms of changes in the number of available
receivers and changes in acoustically-mediated group interactions. By combining signal-to-
noise measurements at the receiver, with group behaviour within this local communication
network, a more informed representation of their important communication space is
presented.

1
2
3 133
45 134 **Materials and Methods**6
**Visual and acoustic data collection**

Acoustic recordings were made of groups of migrating (from their breeding ground in the
Great Barrier Reef, towards their Antarctic feeding grounds) of humpback whales
(September/October of 2002 – 2004 and 2008). The array comprised of five hydrophone-
buoy systems anchored in 20 – 28 m of water in a ‘T’ configuration. Buoys 1 to 3 were
anchored in a line 1.5 km offshore, with buoys 4 and 5 extended a further 1 km offshore.
Each system included a High Tech HTI-96-MIN hydrophone with built-in +40 dB pre-
amplifier, a pre-amplifier (+20 dB) and VHF radio transmitter. Real-time radio transmissions
from the buoys were received at a base station using a Yagi antenna attached to a four
channel, low noise, VHF receiver (type 8101) and a Winradio receiver. Sound source
locations were calculated by *Ishmael* where the mean position of several estimates were
calculated over a brief period. A shore-based theodolite survey of the surface buoys was used
to obtain accurate positions. Sounds were then attributed to migrating groups (see below).
For detailed validation experiments see Noad et al. (2004).

~~Groups were recorded in water depths ranging from 25 to 35 m.~~ Simultaneous land-
based tracking of the groups occurred to obtain visual data of migrating humpback whale
groups within a 10 km offshore radius from the acoustic array. All groups were tracked
regardless of whether or not they were vocalising. Group positions were recorded using a
theodolite linked to *Cyclopes* (a tracking programme developed by Eric Kniest, University of
Newcastle) ~~and additional~~ each time the group surfaced. Additional observations on group
social composition (number of adults within the group and whether or not there was a calf)
and social behaviour (splitting of animals from, and joining of animals to, the group) were
recorded at each position: using binoculars. The ~~position~~ positions of traversing vessels

~~was~~ were also recorded by the land-based observation team. ~~Using this combination of visual~~
~~using the theodolite meaning vessels were tracked as they moved through the study site.~~
Visual and acoustic ~~tracking data~~ positions were then overlaid onto the one Cyclopes file,
~~meaning the acoustic tracking could be matched to groups being visually tracked. Using this~~
~~overlaid file~~, the distance of each signalling group to each hydrophone-buoy receiver was
~~then~~ measured for each recorded social sound.

Vocalising groups were recorded out to approximately 5 km

~~Sound measurements~~

~~Social sounds (n = 1268 vocalisations) were measured from 31 groups (15 the array in~~
~~wind noise and 16 all directions, in vessel noise) starting when they first became audible on~~
~~the array and ending when they were no longer audible as the group migrated out of the area.~~
~~Each social sound was measured using three of the five hydrophones. These were selected to~~
~~maximise the range of distances from the water depths of between 20 and 35 m. However,~~
~~these signalling group whilst still having enough signal above noise to allow measurement~~
~~(i.e. at least some of the signal was visible in the spectrogram). Group recordings lasted~~
~~between 20 minutes and 80 minutes per group depending on their distance offshore from the~~
~~acoustic array. Signalling groups were continuously visually tracked when in visual range (10~~
~~km offshore radius from the array)), meaning the visual track was longer than the acoustic~~
~~track (up to approximately 3 hrs). The full recording chain was calibrated post-field. Full~~
~~system sensitivity varied by 1.5 dB over the frequency range 40 to 10 000 Hz and~~
~~hydrophone sensitivity was -164 dB re 1 V μ Pa⁻¹, which included +40 dB gain for the built in~~
~~preamplifier.~~

Sound measurements

The following analysis was limited to daylight hours, given that is when the visual
observations were taking place. Spectrograms of ~~signals of interest~~ acoustic recordings were
produced and viewed using Raven 1.2 (Cornell Lab of Ornithology) with the DFT size set at
4096 samples, Hamming window, and 80% overlap. Group recordings started when they first
became audible on the array (determined by manual inspection of the spectrogram) and ended
when they were no longer audible as the group migrated out of the area (between 20 and 80
minutes). Social sounds (n = 1268 vocalisations) were measured from 31 groups (15 in wind
noise and 16 in vessel noise). Each social sound was measured using three of the five
hydrophones. These hydrophones were selected to maximise the range of distances from the
signalling group at which the same social sound was recorded, whilst still having enough
signal above noise to allow measurement (i.e. at least some of the signal was visible in the
spectrogram). ~~Each social sound was~~ This meant the same sound could be measured at an
increasing range from the receiver to better quantify the loss in signal energy with increasing
distance.

Each social sound was then isolated from three different channels and saved as a
separate file along with a sample of background noise: either just before, or just after the
sound. This sample included background singing only if this sound source was also present
within the social sound sample. If a sound was not visible on one of the spectrogram, only
two channels were used in the analysis.

Estimates of received and source level were made using a custom-made Matlab script.
~~Each~~ (Girola et al., 2019). From the wavefile, an FFT was used to calculate the frequency
content of the sound ~~was divided into 743 millisecond segments with a 50% overlap,~~
multiplied by a Hanning window. A discrete fourier transform (DFT) was calculated (FFT
size of 16,384 ~~with a~~ 1.35-Hz resolution), 50% overlap) and its power spectrum extracted.

The ~~Power Spectrum (power spectrum was~~ calibrated to account for the sensitivity of the
 hydrophone and the gain of the system) ~~for each segment.~~ The full recording chain was
 extracted as the squared magnitude of the DFT and a noise correction calibrated post-field.
 All sounds measured were made on each segment. The resulting Power Spectrum within the
 dynamic limits of the system. Hydrophone sensitivity was the sound received level for each
 1.35-164 dB re 1 V μPa^{-1} , which included +40 dB gain for the built-in preamplifier. Full
 system sensitivity varied by 1.5 dB over the frequency range 40 to 10 000 Hz frequency band
 for each segment.

Transmission loss (TL) was measured at the site as described in (Dunlop et al., 2013).
 A boat and playback of octave band limited white noise were used as the sources. The boat
 conducted runs along lines radiating to and from the array, from distances of 100m out to
 about 10 km. The boat speed was kept constant. Wind speed for that day ranged from 10 to
 15 knots, swell was less than 1 m, and there were no significant currents in the area. This
 gave various regression lines of received levels as a function of distance from the sources:

$$220 \quad TL = a + b \log(x) \quad (1)$$

where b is the slope of the regression line, x is distance (meters) and a is a constant (which
 was frequency dependent). ~~For~~ The value of a may have varied with the direction of the boat
 (approaching or going away from the receivers) as the noise radiated forward likely differed
 from the noise radiated aft. Even so, for most frequencies, b varied with distance but could be
 well approximated by two values; one applying to distances less than, and the other greater
 than, a cross over value ~~where the slope changed.~~ Absolute values of TL were determined by
 measuring received levels of octave band limited noise, projected using a J11 source
 suspended from a boat (estimating the source level using a hydrophone suspended at a
 distance of 3 m and corrected to 1 m assuming spherical spreading). This source was
 projected at three distances (between 200 and 1000 m) from the array. The trend in loss, b
 $\log(x)$, from the boat runs was then fitted to the absolute values of loss from the J11

measurements to determine the value of a for each octave band, by minimising the sum of the
squares of the differences between $a + b \log(x)$ and the data points from the J11
measurements.

Equation (1) values were then calculated for each octave band.

The signal was then reconstructed and filtered from 35 to 5623 Hz to eliminate low-
frequency turbulence noise and high-frequency system noise.

Broadband background noise levels during this time (NL_{bb} : actual band 36 Hz to 2.8
239 kHz) were estimated using 10-minute files isolated from each hydrophone-buoy. As
recordings were divided into 10-minute files for storage, the file containing the social sound
of interest was used for the background noise measurement. Wind-dependent noise was
measured when there were no audible and/or visually tracked vessels in the area (including
shipping vessels in the within-sight shipping channel). A previous study at this site found the
measured noise in the absence of singer and vessel noise similar to that generally observed
for wind-dependent noise in Australian waters (Cato, 1996). In addition, this study found
wind speed (ranging from <5 to 15 knots) to be significantly correlated with noise levels in
all third octave frequency bands (Dunlop et al., 2010). Wind-dominated noise measurements
could be considered to be similar to that at the vocalising whales and were therefore similar
across hydrophones. These varied from approximately 90 dB *re* 1 μ Pa to 112 dB *re* 1 μ Pa in
wind speeds ranging from <5 knots to approximately 15 knots. The median wind-dominated
noise was 100 dB *re* 1 μ Pa for this area: at a wind speed of 12 knots. The vessel noise
environment was due to passing fishing vessels and usually audible on the array for 10 to 40
minutes depending on the vessel's trajectory, distance from the array, speed, and size.
Measured noise levels were above 100 dB *re* 1 μ Pa with mean and modal measured levels
being 105 dB *re* 1 μ Pa (vessels were approximately 4 km from the array). Given the distance
of the vessel, and therefore measured noise, varied across the three hydrophones, the

hydrophone used ~~theto~~ measure the signal was also the hydrophone used to measure the
 noise. It should be noted that vessel noise may not have dominated the noise environment for
 the entire recording given the vessels would have faded in and faded out as they traversed the
 area.

The 1/3 octave band containing the most energy was used as the array received level
 (RL_{rms} for) and noise levels at the where then used to calculate array signal-to-noise analysis
 ($rSNR$ levels ($rSNR$)) as per Dunlop (~~34~~ et al., 2013). From the received level of each sound
 (RL_{rms} over 35 to 5623 Hz), the source level was estimated as:

$$265 \quad SL_{rms} = RL_{rms} + TL \quad (2)$$

**Communication space models**

First, signals were grouped into “low-frequency” or “high-frequency” (separated by
 frequency content) using the results of the classification and regression tree analysis (CART)
 analysis presented in Dunlop (2018). To summarise, low-frequency sounds generally had a
 centre frequency below 158 Hz and a minimum frequency below 126 Hz.

The communication space has been previously ~~modeled~~ statistically modelled for
 wind-dominated (2018) and vessel (Dunlop, 2018) noise ~~and further~~. Further details on the
 ~~modeling~~ statistical modelling procedure and model outputs are presented in the
 supplementary methods (Figs. S1, S2 and Table S1). ~~Here, it~~ In brief, a two-dimensional
 smooth surface was fitted to the interaction between background noise and separation
 distance between the signaller and array. The response variable was the received SNR at the
 array (termed $rSNR$). This gave a series of trend lines representing the relationship between
 the distance of the signalling group from the receiver, broadband noise levels, and measured
 $rSNR$ for low- and high-frequency vocal sounds. A separate analysis was carried out for
 groups in wind-dominated noise and groups in vessel-dominated noise. It was assumed that

an $rSNR$ below 0 would equate to reduced ability of receiver humpbacks to discriminate
and/or recognise the signal. Therefore, for this study, $rSNR = 0$ was assumed to delimit the
signalersignaller communication space. To compare the size of the communication space in
wind-dominated noise with vessel noise, statistical model outputs for the 0 dB $rSNR$ trend
lines were selected ~~(with full models presented in supplementary materials). In brief, these~~
~~outputs are the modeled relationships between the decrease in distance from the signaler (y~~
~~axis; response variable) with increased broadband noise (x axis; predictor variable) for both~~
~~noise sources.~~ Raw data-points were also selected and plotted displayed to indicate the
variation around the displayed trend line.

Further statistical models were ~~ran~~run using the $rSNR$ as the response variable to
assess the variation in $rSNRs$ at the receiver. Here, signals from groups between 900 and
1100 m from the receiver were measured to limit the data to signals produced from groups
approximately 1 km from the receiver ($n = 14$ and $n = 6$ groups for low- and high-frequency
models respectively). Response data were normally distributed. A generalised linear mixed
model framework was used in “R” software with the “lmer” package for model fitting.
Predictor variables were broadband noise with the interaction effect of dominant noise source
(wind or vessel). GroupID was included as a random effect as multiple signals were
measured from the same group.

**Behavioural analysis**

The social environment was first quantified and compared between groups measured in wind-
dominated noise, with those measured in vessel noise, to ensure the social environments were
comparable (see supplementary methods). For every 10 minutes of the recording, the mean
distance of each neighbour, being the nearest neighbour, 2nd nearest neighbor, and so on for
every group within a 5 km radius of the signaling group was measured. This gave between

two and eight measured time periods per group, depending on the length of the recording, and
between zero and six neighbours per time period. Social behaviour was then quantified for
each neighbour according to whether (1) or not (0) it interacted with the signalling group as
outlined in supplementary materials. An interaction was defined as being joined by, or
joining, another group, or another group changing direction and approaching to within
approximately 200 m of the signalling group. To determine if groups were less likely to
interact with each other in increasing vessel noise compared to wind noise, the response
variable (neighbour interaction) was modelled using a general linear model, assuming a
binomial regression distribution, and including the interaction effects of neighbour distance,
noise source, and noise level.

**Communication area and receiver group distribution models**

The communication area (circular area around the signaller) was calculated using the radius
from the signaller at which the signal level above noise was equal to 0 dB. This was the most
simplistic representation of communication area and likely to be used in the absence of
information on propagation loss. Four ~~groups~~ randomly distributed receiver groups
(maximum number observed) were placed in the area. The communication space for low-
frequency signals was then reduced according to the communication modelspace results for
vessel noise, (loss of radial distance), and the number of groups left within the area counted.
This simulation was repeated 30 times (approximate signaller group sample size) to give an
estimated mean (with standard deviation) of the percentage loss of receiver groups due to the
reduced area. The same random group distribution of four groups within the area was then
used to simulate (30 times) the percentage loss in receiver groups for the high-frequency
signal communication space.

Next, the known (taken from the social environment data) percentage loss in number
of receiver groups was determined for each signaller group (per 10 minutes) and reported as a
mean and standard deviation for all signallers. Wind and vessel noise data were combined
given there was no significant difference in receiver group distribution between the two noise
environments: (supplementary methods).

**Results**

**Social environment**

~~The~~From the visual observations, the majority of signalling groups ~~had~~were observed to have
between 2 and 4 groups within 5 km ($n = 21$). In both noise environments, the mean distance
of nearest neighbour was 2 km, with 1st and 3rd quartile ranges of 1 to 3 km. The second
nearest neighbour mean distance was approximately 3 km, and the third, approximately 4 km,
from the vocalising group. Signalling groups most commonly interacted with their nearest
neighbour (16 out of 19 interactions), and then may have further interacted with their second
neighbour ($n = 1$) or third neighbour ($n = 2$).

~~Modelled communication~~Communication space

The ~~modelled~~ communication space for the low-frequency ~~vocalisations was~~vocalisation
category was estimated to be 4 km in typical (received noise levels of 100 dB *re* 1 μ Pa) wind
noise, reducing to 3 km in the same received level of vessel noise (Fig. 1a). These noise
levels equated to typical wind conditions and relatively low received levels of vessel noise.
The maximum interaction distance of groups was a separation of 4 km, where a neighbour
turned to head towards a signalling group from this distance to eventually join together ($n =$
2). This separation distance matched the likely extent of the communication space for low-
frequency sounds in typical wind-dominated noise. The size of the communication network

for migrating humpback whales, in this shallow-water environment (<50 m), was therefore
 likely to be between two and four other groups based on the distribution of groups from the
 signaller.

For high-frequency vocalisations, the communication space was 2.5 km in typical
 wind noise-, given they were lower in level (mean of 149 dB *re* 1 μ Pa @ 1m compared to 156
 363 dB *re* 1 μ Pa @ 1m in low frequency sounds). This would reduce the number of receiving
 groups within the network to one or two. Groups, in wind noise, had a +50% chance of
 interacting if separated by less than 2 km (Fig. 2),) suggesting this distance, being the local
 neighbourhood, was was likely to be important in terms of acoustically-mediated group social
 behaviour-: (joining together). Even in relatively low received levels of vessel noise (100 dB
 *re* 1 μ Pa), the communication space of high-frequency vocalisations reduced to 1.5 km (Fig.
 1b).

~~Fig. 1. The modelled relationship of the communication space of humpback whale low- (a)~~
~~and high- (b) frequency signals in wind-dominated and vessel noise. Smoothing functions are~~
390 ~~from the generalised estimate equation outputs (Table S1) for the $rSNR = 0$ relationship (from~~
~~Figs. S1 and S2). Raw data points at which the signal $rSNR$ at the array was between -1 and 1~~
391 ~~are also included.~~

**Effect of increased noise**

As received noise levels increased to 105 dB *re* 1 μ Pa (mean received vessel noise and above
average wind noise), the communication space decreased to approximately 2 km and 1 km
for low- and high-frequency vocalisations respectively (Fig. ~~2a)-2a~~, meaning the network
would likely be reduced to one receiver. By received noise levels of 110 dB *re* 1 μ Pa, which
is considered to be high wind noise, or noise from a close-by (within 2 km) vessel, the
communication space for both signal types was reduced to only 1 km-, which may or may not
include one receiver depending on its distance from the signaller. However, although the
reduction in ~~modelled~~ communication space was the same for both noise sources, a neighbour
was significantly less likely to interact with the signalling group when there was a vessel in
the area ($t = 2.19$, $p = 0.03$; Fig. ~~2a)-2a~~) compared to if the noise source was
wind-dominated. As signalling groups were most likely to interact with their nearest
neighbour, the position of this receiving group within the signaller's communication space
was the most important to note (Fig. 2b and c). One explanation for this reduction in
interaction ~~behavior~~behaviour would be increased signal masking, as received noise would be
likely be elevated above typical wind noise if a vessel was in the area. However, noise level
was not a significant predictor in that increased noise did not explain any reduction in the
likelihood of groups to interact.

Fig. 2. The probability of the signalling group interacting with a neighbouring group as a function of the distance of the neighbour from the signaler. Relationships are shown for the wind and vessel noise environments and include 95% confidence intervals (a) and separated according to the neighbour position relative to the signalling group (b and c in wind and vessel noise respectively).

Signalling groups had an 80% chance of interacting with another group if separated
by 1 km as long as there was no vessel in the area. This reduced to a 30% chance if a vessel
was present (Fig. 2). Even in vessel noise, many received signals would have been above
noise at this close-by separation distance (Fig. 1) suggesting masking was not a significant
issue. If using 105 *re* 1 μ Pa as a generic (average) received vessel noise level, though the
modelled communication space of both vocalisation types was significantly reduced, it was
still beyond 1 km (Fig. 1). Using signals at which the $rSNR = 0$ does not include any measure
of variance, in that, signals received by a whale at 1 km are likely to range in $rSNRs$. If
choosing all signals emitted from groups 1 km positioned at 1 km from the receiver, the $rSNR$
of low-frequency signals, in received vessel noise of 105 *re* 1 μ Pa, was up to + 13 dB (Fig.
3a), and for high-frequency signals, up to + 12 dB (Fig. 3b). The decrease in the $rSNR$ with
increasing measured noise was also significantly less in vessel noise compared to wind-
dominated noise (Table 1; Fig. 3b). In other words, low-frequency signals were less masked
by vessel noise as long as the signaller was 1 km from the receiver. Therefore, at close group-
to-group separation distances, the observed reduction in interaction ~~behavior~~behaviour was
likely not due to signal masking.

Fig. 3. The modelled relationship, and significance of the predictor variables, of the rSNR of humpback whale low- (a) and high- (b) frequency signals in wind and vessel-dominated noise received noise. Data points from signaling whales between 900 and 1100 m (approximately 1 km) from the array are included.

Communication area and group distribution models

The loss of total communication area in vessel noise compared to wind-dominated noise was 75% for both signal types (Table 1), as compared to communication area in wind-dominated noise. Assuming a random distribution of groups within that area (4 groups per 50²km), this equated to an approximate 77 to 78% reduction in available receivers within the network

(Table 1) for both signal types ~~based on 30 simulations~~. However, the distribution of groups
at this study site suggests a bias towards the signalling group, where the density of groups
(available receivers) was higher closer to the signalling group (Fig. 1). Using this known
distribution, the reduction in available receivers due to loss in communication space was
estimated as 47% and 33% for the low and high-frequency signal communication area
respectively (Table 1). Behavioural data from this study suggests the reduction in low-
frequency signal communication space was unlikely to lead to a significant reduction in
social interactions, given most of these interactions occurred within 2 km of the signalling
group. It was also clear that the probability of a receiver group interacting with a signaller
group was dependent on its position and distance from the signaller (Fig. 2b and c). In wind
noise, the signalling group had a 70% chance of interacting with the nearest neighbour if the
separation distance was less than 2 km, reducing by 50% in vessel noise. Therefore, this
reduction in interaction behaviour was greater than that predicted by using known group
distribution as a proxy for the likelihood of social interactions, but less than using a random
distribution of groups within the communication space assuming a simplistic spherical
communication area.

Table 1. The modelled communication area based on the distance from the signaller at which
the $rSNR = 0$. The group numbers within the wind and vessel noise communication areas
were estimated assuming a random distribution (of 4 groups per 50^2km) and known
distribution, using visual data from signaling of groups around signalling groups. Standard
deviations are in brackets.

Signal type	Distance km	Area km^2	Random distribution		Known distribution	
			No. groups	% reduction	No. groups	% reduction
Low-f	4	50	4		2.2 (1.0)	
	2	50	0.9 (0.9)	77 (22)	1.0 (0.8)	47 (36)
High-f	2	12	0.9 (0.8)		1.0 (0.8)	
	1	12	0.3 (0.5)	78 (35)	0.5 (0.6)	33 (44)

Discussion

~~For many cetacean species, especially baleen whales, there is little information on~~
~~how they hear their own signals in noise. Cetacean~~ This goal of this study was to define the
~~communication space is therefore usually modelled using a presumed limit in receiver~~
~~audibility (e.g. Tennessen and Parks, 2016), and in some cases, a presumed ability of the~~
~~receiver to recognise the signal (e.g. Cholewiak et al., 2018; Hatch et al., 2016). These studies~~
~~assume the distance at which the measured received level~~ network of a signal-to-noise ratio,
~~or signal excess, is of a certain value (e.g. 0 dB) delimits the signal's communication~~
~~distance. Here, a received signal-to-noise level of 0 was used to define a signalling humpback~~
~~whale's communication space. Using this value, the signaller communication space for low-~~
~~frequency vocalisations~~ whales and determine if, and how, this was estimated at
~~approximately 4 km in typical wind noise conditions. Behavioural data agreed with these~~
~~modelled limits as reduced during periods of vessel noise. Here, communication space was~~

534 defined as distance that conspecifics could potentially send and receive acoustic signals
(assuming a signal excess of 0 dB delimited this distance), and where there was an observable
(social) response in the receiving whale. First the communication space and network was
defined in wind-dominated (natural) noise. Results suggest that humpback whale low-
frequency social signals had a communication space of approximately 4 km with a
communication network of two to four other groups. Behavioural data found that humpback
whale receivers responded to, and eventually joined, a signalling group from a maximum
sighted separation distance of 4 km. Therefore, these sounds may function to mediate longer-
range interactions between networks of dispersed groups. However, the local neighbourhood
of signalling humpback whales, where sighted behavioural interactions were most likely to
occur, was more similar to the ~~modelled~~ communication space of higher frequency signals (2
545 km). ~~It seems that 0 dB SNR, as measured here, defines both the maximum communication~~
546 ~~space when using~~ Higher frequency signals were of a lower amplitude compared to low-
547 frequency signals, ~~and socially important communication space when using high-frequency~~
548 ~~signals. meaning they may be used mediate closer-range joining interactions between a~~
549 ~~signaling and receiving groups.~~ There is no reason to suggest other species of baleen whales
~~hear signals in noise differently. Therefore, as long as signal type is taken into account, until~~
~~more data becomes available on their hearing in noise, the 0 dB SNR limit seems appropriate~~
~~to define baleen whale receiver capabilities in noise. These results will aid in defining the~~
~~communication of~~ was usually only one other baleen whale species if behavioural data is not
~~available group within this network; the group participating in the joining interaction.~~
~~Here, if continuing to use the 0 dB SNR limit, in~~To test for effects of anthropogenic
~~noise, the likely extent of a signalling whale's communication space was compared in natural~~
~~(wind) and anthropogenic (vessel) noise.~~

In average vessel noise (105 dB *re* 1 μ Pa), the communication space of humpback whale
low-frequency vocalisations reduced, by half, to 2 km. These results are similar to modelled
loss in communication space of humpback whale social sounds in vessel noise in Cholewiak
(et al., 2018). The communication space of high-frequency sounds was also reduced by half
in vessel noise to only 1 km. This translates to a reduction of four times the communication
area (~~i.e. the, assuming a~~ circular area from the signaller) for both signal types. To assess the
consequences of increased vessel noise on the communication network, the magnitude of the
reduction depended on the method use. If assuming ~~receivers were randomly distributed~~
~~within that area, this would equate to a 75% reduction in a random distribution of groups~~
~~around the signaller, the number of~~ available receivers. ~~There~~ within the network reduced
~~from four to one.~~ However, there are two problems with this assumption. ~~Firstly~~First, the
receiving groups were not randomly distributed from the signaller, and ~~secondly~~second,
signallers were more likely to interact with the closest receiver as long as it was ~~no more~~
~~than within~~ 2 km away. ~~Accounting.~~ After accounting for ~~the first problem meant there was~~
~~only 47% these two problems, the~~ reduction in available receivers ~~if reducing the signaller~~
~~distance from 4 km decreased~~ to 2 km. ~~Accounting for the second problem meant there was~~
~~only a 47% and 33% reduction in available receivers, if reducing the distance from 2 km to 1~~
~~km respectively.~~ Behavioural ~~data, however, observations~~ found a 50% reduction in the
likelihood of a signalling group interacting with a receiver group within the 2 km interaction
zone. Therefore, ~~models~~ using the random distribution of receivers ~~may over-~~
~~estimate~~ estimated the ~~consequences, reduction in communication network~~ whereas ~~models~~
accounting for receiver distribution, but not receiver behaviour, ~~may under-estimate~~
~~consequences.~~ ~~Future studies modelling the loss of communication space in increased noise~~
~~for a marine mammal in terms of signaller-receiver consequences should at least account for~~
~~receiver distribution relative to the signaller.~~ under-estimated network consequences.

However, as ~~outlined~~explained below, ~~there was an additional factor to consider, being the~~
physical presence of the vessel ~~may cause additional behavioural effects not accounted for~~
584 ~~within masking models.~~

Though this study focused on the acoustic environment and behaviour of receivers,
potential ~~signalers~~signaller anti-masking strategies were also considered. ~~Here, the~~
~~measured~~The estimated communication space of low-frequency signals in wind-dominated
noise was maintained at approximately 4 km up to received levels of 100 dB *re* 1 μ Pa. This
was likely due to the Lombard response ~~in~~ found humpback whales, where signallers
increased their vocal source level in response to increased wind-dominated noise (Dunlop et
al., 2014). In the same received levels of vessel noise, the communication space was found to
be smaller, ~~likely due to the lack of.~~ A previous study found no signaller Lombard response
to increasing vessel-dominated noise (Dunlop, 2016b). Vessel noise levels were estimated at
the signalling group (in Dunlop, 2016b) and were comparable to those measured at the array
in this study, ~~meaning.~~ In other words, received levels of vessel noise should have been high
enough to cause a Lombard response. ~~in signalling whales but did not.~~ One explanation
~~for put forward in this~~ is study was that whales have release from masking, in that they can
focus their attention on the vocalising conspecific ~~signals~~, rather than the vessel noise. This
implies the receiver groups, if possessing the ability for masking relief, would continue to
behave as if there was no masking noise source in the area. Behavioural results presented
here showed this was not the case. Further, the reduction in humpback group interaction
behaviour in vessel noise was not fully explained by signal masking. ~~Humpback whale~~
~~signallers have been found to increase their source level in response to an increase in~~
~~background noise, where shipping noise formed part of the noise environment (Fournet et al.,~~
~~2018).~~ However, these Behavioural interactions were ~~distant ships, whereas in this study,~~
~~vessels most likely to occur when groups~~ were traversing within the signaller's communication

~~network-separated by 1 km or less where most signals, even high-frequency signals, were still~~
~~well above received vessel noise and unlikely to be masked.~~ Therefore both the lack of
~~signalersignaller~~ Lombard response, ~~and~~ found in a previous study (Dunlop, 2016b), and the
significant change in receiver behaviour ~~seemfound here, are more likely~~ to be ~~the result of a~~
~~response, such as~~ an avoidance response, to the vessel itself. The physical presence of the
vessel has been shown to reduce bottlenose dolphin (Bas et al., 2017) and humpback whale
(Blair et al., 2016) foraging activity as well as bottlenose dolphin resting and socialising
behaviour (Bas et al., 2017). Here, the combination of communication space ~~modelling, with~~
~~theand~~ behavioural analysis, illustrates the need to inform one result with the other. This
provides a powerful tool with which to fully assess the consequences of anthropogenic noise
on cetacean social behaviour ~~thatwhich~~ does not rely on communication space modelling and
assumptions of group distribution and social behaviour.

The population used for this study is robust in that it is increasing at 11% per year
(Noad et al., ~~in press~~2019). These behavioural interruptions within the communication
network, if localised and rare, are unlikely to have a significant effect on such a robust
population. As many of these social interactions are likely to be breeding interactions
(Dunlop et al., 2008; Dunlop 2016a, Dunlop and Noad, 2016), understanding the population
consequences of disturbance, in this case reduced potential breeding opportunities, would be
the next step in understanding whale and vessel interactions. Given this population is also
well studied in terms of population dynamics, a population consequences model would not be
difficult to produce once the link between reduced group interactions within the
communication network and reduced mating opportunities has been established. These
models could be extended to other humpback populations and other baleen whale species,
such as the highly endangered right whale (*Eubalaena spp.*), where less is known about their
breeding behaviour. For now, the most comprehensive model to estimate loss of

communication space for baleen whales in vessel noise would be to account for signal type,
receiver group distribution with some knowledge of the local signalling neighbourhood, and
any additional (e.g. avoidance) effect of the presence of the vessel.

Along the eastern coast of Australia, and likely other populated coastlines, vessel
activity is increasing due to the growth in the tourism industry. For cetaceans occupying a
coastal habitat, vessel interactions are becoming more of an issue. Current Australian
mitigation measures regarding the interaction of vessels and whales include caution zones
(within 300 m of the whale) and exclusion zones (within 100 m). The assumption with these
measures is that staying more than 300 m away “reduces the risk of disturbance to natural
behaviours”. The results of this study suggests otherwise. What is clear is that the deleterious
effects of vessels on whales are not limited to collisions and increased signal masking but
potential changes in vital function behaviours, such as breeding interactions, should also be
considered. In summary, mitigating vessel activity from a noise and ~~modelled~~ communication
space perspective may not mitigate for behavioural changes within their communication
network.

Ethics and permits

The project operated under animal ethics approval from the University of Queensland Animal
Ethics Committee, a Queensland Government Environment, and Heritage Protection permit
and an Australian Government Access to Biological Resources in a Commonwealth Area for
Non-Commercial Purposes permit.

Data accessibility

The article's supporting data are available on the Dryad Digital Repository: [\(Dunlop, 2019\):](https://datadryad.org/review?doi=doi:10.5061/dryad.65j00m3)
<https://datadryad.org/review?doi=doi:10.5061/dryad.65j00m3>). Wind and vessel noise
communication models are uploaded as part of the supplementary material.

Competing interests

There are no competing interests in that funding agencies had no input in to experimental

design, data collection, data analysis or interpretation or the writing of the paper.

Author contributions

RD conceived the study, designed the study, carried out the data analysis and wrote the

manuscript. Raw data were collected as part of a previous collaborative project (HARC, see

acknowledgements).

**Acknowledgements**

The author would like to thank everyone involved in the Humpback Acoustic Research

Collaboration (HARC; funded by the U.S. Office of Naval Research and the Australian

Antarctic Division), in particular the numerous volunteers who donated their time and

energy to this project. I also thank David Paton for his invaluable field expertise and Eric

Kniest for his continued support in the development of *Cyclopes*. The author would

particularly like to acknowledge Associate Professor Michael Noad for leading the HARC

work (without which, this study would not have been possible) and Dr. Douglas Cato for

his continued support and mentorship.

Bas AA, Christiansen F, Ozturk B, *et al.* 2017. Marine vessels alter the behaviour of

bottlenose dolphins *Tursiops truncatus* in the Istanbul Strait, Turkey. *Endanger Species Res*

**34**:1-14.

Blair ~~H. BHB~~, Merchant ND, Friedlaender AS, Wiley DN, Parks SE. 2016. Evidence for
ship noise impacts on humpback whale foraging behaviour. *Biol Lett* **12**(8).
Cato, DH. 1996. Ambient sea noise in waters near Australia. *J. Acoust. Soc. Am* **60**:320–328.
Cholewiak D, Clark CW, Ponirakis D, *et al.* 2018. Communicating amidst the noise:
modeling the aggregate influence of ambient and vessel noise on baleen whale
communication space in a national marine sanctuary. *Endanger Species Res* **36**:59-75.
D'Vincent CG, Nilson RM, Hanna, RE (1985). Vocalization and coordinated feeding
behaviour of the humpback whale in southeastern Alaska. *Sci Rep Whales Res Inst, Tokyo*,
**36**:41—47.
Dunlop R. 2019. Data from: Mitigations for the effects of vessels on baleen whale signaling
from a communication space perspective may not prevent behavioural changes. Dryad Digital
Repository. <http://doi.org/10.5061/dryad.65j00m3>.
Dunlop RA. 2016a. Changes in vocal parameters with social context in humpback whales:
considering the effect of bystanders. *Behav Ecol Sociobiol* **70**(6):857-870.
Dunlop RA. 2016b. The effect of vessel noise on humpback whale, *Megaptera*
*novaeangliae*, communication behaviour. *Anim Behav* **111**:13-21.
Dunlop RA (2017) Potential motivational information encoded within humpback whale non-
song vocal sounds. *J Acoust Soc Amer*, **141**(3):2204-2213.
Dunlop RA. 2018. The communication space of humpback whale social sounds in wind-
dominated noise. *J Acoust Soc Amer* **144**(2):540-551.
Dunlop RA. The communication space of humpback whale social sounds in vessel noise.
*Proceedings of Meetings on Acoustics*, In press, accepted 20/11/18.
Dunlop RA, Cato DH, Noad MJ. 2008. Non-song acoustic communication in migrating
humpback whales (*Megaptera novaeangliae*). *Mar Mamm Sci* **24**(3):613-629.
Dunlop RA, Cato DH, Noad MJ. 2010. Your attention please: increasing ambient noise
levels elicits a change in communication behaviour in humpback whales (*Megaptera*
*novaeangliae*). *Proc Roy Soc B* **277**(1693):2521-2529.
Dunlop RA, Cato DH, Noad MJ. 2014. Evidence of a Lombard response in migrating
humpback whales (*Megaptera novaeangliae*). *J Acoust Soc Amer*, **136**(1):430-437.
Dunlop RA, Noad MJ. 2016. The "risky" business of singing: tactical use of song during
joining by male humpback whales. *Behav Ecol Sociobiol* **70**(12):2149-2160.
Dunlop, RA., Noad, MJ, Cato, DH, Stokes, D. 2007. The social vocalization repertoire of
east Australian migrating humpback whales (*Megaptera novaeangliae*). *J Acoust Soc Amer*
**122**(5): 2893-2905.

Erbe C, Reichmuth C, Cunningham K, *et al.* 2016. Communication masking in marine
mammals: A review and research strategy. *Mar Poll Bull* **103**(1-2):-15-38.
~~Fournet MEH, Matthews LP, Gabriele CM, Haver S, Mellinger DK, Klinck H. 2018.~~
~~Humpback whales *Megaptera novaeangliae* alter calling behavior in response to natural~~
~~sounds and vessel noise. *Mar Ecol Prog Ser* **607**:251-268.~~
Girola, E., Noad, MJ, Dunlop, RA, Cato DH. 2019. Source levels of humpback whales
decrease with frequency suggesting an air-filled resonator is used in sound production. *J*
*Acoust Soc Amer* **145**(2):869-880.
Hatch LT, Clark CW, Van Parijs SM, *et al.* 2012. Quantifying Loss of Acoustic
Communication Space for Right Whales in and around a U.S. National Marine
Sanctuary. *Con Biol* **26**(6):983-994.
Heiler J, Elwen SH, Kriesell HJ, Gridley T (2016) Changes in bottlenose dolphin whistle
parameters related to vessel presence, surface behaviour and group composition. *Anim*
*Behav*; **117**:167-177.
Janik VM. 2000. Source levels and the estimated active space of bottlenose dolphin
(*Tursiops truncatus*) whistles in the Moray Firth, Scotland. *J Comp Physiol A* **186**(7-8):673-
680.
Lesage V, Barrette C, Kingsley MCS, Sjare B (1999) The effect of vessel noise on the vocal
behavior of Belugas in the St. Lawrence River estuary, Canada. *Mar Mamm Sci*; **15**(1):65-
84.
McGregor PK (1993) Signaling in territorial systems – a context for individual
identification, ranging and eavesdropping. *Philosophical Transactions of the Royal Society*
*of London Series B-Biological Sciences*; **340**(1292):237-244.
McGregor PK, Peake TM, Lampe HM (2001) Fighting fish *Betta splendens* extract relative
information from apparent interactions: what happens when what you see is not what you
get. *Anim Behav*; **62**:1059-1065.
Miller PJO. 2006. Diversity in sound pressure levels and estimated active space of resident
killer whale vocalizations. *J Comp Physiol A* **192**(5):449-459.
Miller PJO, Biassoni N, Samuels A, Tyack P L (2000) Whale songs lengthen in response to
sonar. *Nat*; **405**(6789):903-903.
Noad MJ, Cato DH, Stokes MD (2004) Acoustic tracking of humpback whales: measuring
interactions with the acoustic environment. Proceedings of Acoustics 2004, Annual
Conference of the Australian Acoustical Society, Gold Coast, 3-5 November 2004, pp 353–
358.
Noad MJ, Dunlop RA, Kniest E. Boom to bust? Implications for the continued rapid growth
of the eastern Australian humpback whale population despite recovery. *Pop Ecol* **In press**,
**accepted 25/09/2018** **61**(2):198-209.

Otter K, McGregor PK, Terry AMR, Burford FRL, Peake, TM, Dabelsteen T (1999) Do
female great tits (*Parus major*) assess males by eavesdropping? A field study using
interactive song playback. *Proceedings of the Royal Society B-Biological Sciences, Proc Roy*
*Soc B* **266**(1426):1305-1309.
Peake TM, Terry AMR, McGregor PK, Dabelsteen T (2001) Male great tits eavesdrop on
simulated male-to-male vocal interactions. *Proceedings of the Royal Society B-Biological*
*Sciences, Proc Roy Soc B* **268**(1472):1183-1187.
Parks SE, Clark CW, Tyack PL (2007) Short- and long-term changes in right whale calling
behavior: The potential effects of noise on acoustic communication. *J Acoust Soc Amer*;
**122**(6):3725-3731.
Parks SE, Cusano DA, Stimpert AK, Weinrich MT, Friedlaender AS, Wiley DN (2014)
Evidence for acoustic communication among bottom foraging humpback whales. *Sci Rep*; **4**.
Parks SE, Johnson M, Nowacek D, *et al.* 2011. Individual right whales call louder in
increased environmental noise. *Biol Letters* **7**(1):33-35.
Rekdahl ML, Dunlop RA, Goldizen AW, Garland EC, Biassoni N, Miller P, Noad MJ
(2015) Non-song social call bouts of migrating humpback whales. *J Acoust Soc Amer*;
**137**(6):3042-3053.
Sharpe FA, Dill LM, Beaver V, Spellman B. (1998) Killing me softly: feeding calls of the
Alaskan humpback whale. Abstracts of the World Marine Mammal Science Conference,
Monaco, January, pp 20 – 24.
Silber GK (1986) The relationship of social vocalizations to surface behavior and aggression
in the Hawaiian humpback whale (*Megaptera novaeangliae*). *Can J Zoo*; **64**(10):2075-2080.
Tennessen JB and Parks SE. 2016. Acoustic propagation modeling indicates vocal
compensation in noise improves communication range for North Atlantic right
whales. *Endanger Species Res* **30**:225-237.
Tervo OM, Christoffersen MF, Simon M, *et al.* 2012. High Source Levels and Small Active
Space of High-Pitched Song in Bowhead Whales (*Balaena mysticetus*). *Plos One* **7**(12).
Thompson POW, Cummings WC, Ha SJ (1986) Sounds, source levels and associated
behavior of humpback whales, Southeast Alaska. *J Acoust Soc Amer*; **80**(3):735 - 740.
Wiley RH, Richards DG. 1978. Physical constraints on acoustic communication in
atmosphere – implications for evolution of animal vocalizations. *Behav Ecol Sociobiol*;
**3**(1):69-94.

Fig. 1. The relationship of the communication space of humpback whale low- (a) and high-
(b) frequency signals in wind-dominated and vessel noise. Smoothing functions are from the
generalised estimate equation outputs (Table S1) for the $rSNR = 0$ relationship (from Figs. S1
and S2). Raw data-points at which the signal $rSNR$ at the array was between 1 and -1 are also
included.

Fig. 2. The probability of the signalling group interacting with a neighbouring group as a
function of the distance of the neighbour from the signaller. Relationships are shown for the
wind and vessel noise environments and include 95% confidence intervals (a) and separated
according to the neighbour position relative to the signalling group (b and c in wind and
vessel noise respectively).

Fig. 3. The modelled relationship, and significance of the predictor variables, of the $rSNR$ of
humpback whale low- (a) and high- (b) frequency signals in wind and vessel-dominated noise
received noise. Data-points from signalling whales between 900 and 1100 m (approximately
1 km) from the array are included.

Appendix D

Editor comments:

The only remaining comment of substance relates to adding some text in the discussion that relates to how this new information is different and/or builds upon previous work. At present it is not completely clear and I believe that this can be done relatively easily.

Response: this has been carried out as per reviewer 2's comments – statements consolidating past and this work have been added within the discussion

Reviewer comments to Author:

Reviewer: 2

Comments to the Author(s)

The paper is much improved. However, I still feel that there could be a little more care in leading the reader and being concise.

Introduction

There is information particularly in the introduction that I do not think adds to the overall message of the piece. In my opinion the 'unintended' receivers is given emphasis, rather than simply stated communications may also be received by eavesdropping whales. I think it detracts slightly from the message of interacting groups having 2km threshold and is not (rightly) brought up again in the discussion. I suggest the author consider editing.

Response: changed as suggested

Methods

A little more on the study site might be helpful. Where was the array, what is the typical ambient noise setting (if not using wind dominated to represent this), what is the topography and setting of the array (to give an idea of propagation pathways) etc.

Response: further details added

Some clarification needed to make sure details are consistent. In talking about neighbouring groups you say measurements are between 0 and 6 groups in line 263, and then between 4 groups (max observed) in line 277 used in the model, and then the majority of interactions being between 2 and 4 groups in line 297.

Response: the first was the total neighbouring groups within 5 km, the second was maximum observed within the defined communication area (SNR = 0), and the third is was the modal value. Changed to make clearer.

Results

Are there any indications in the surface observations or the acoustic recordings that the groups are in fact communicating/interacting with each other that could be described. Are there calls known to be used within groups and between groups, or changes in calling behaviour seen in different ambient conditions (other than that stated from previous studies).

Response: only what was mentioned in the introduction, which gave a background on the use of these recorded signals in social interactions and changes in these signals with noise. Given this

background info is on the same population of whales at the same study site, this point was covered as extensively as possible in the introduction.

Discussion

If the physical presence of vessels is influential is there anything in the data that might suggest that this influence is increased as vessels approach etc. Is there anything from the observational data that further supports the statement, or that of the whales avoiding vessels? Were difference behaviours seen compared to addition of vessel number, proximity, type – this could be helpful for management.

Response: the simple answer is that this would require another more targeted study (on behaviour around vessels), which is underway but well beyond the scope of this study. Added in some results of previous studies (studies on the effects of seismic on behaviour, but using controls (vessel) so with some idea of their behavioural response to vessels) to address with a caveat that further and more targeted studies are required (which, as mentioned earlier, are underway).

What is the typical composition of vessel noise – could you predict this would affect high frequency calls more? Is there more that could be discussed in terms of changes in calling in terms of call type, rate, structure rather than just the Lombard effect?

Response: other parameters were tested in the Dunlop (2016b) – peak frequency and duration, with no significant change in these. However, these were parameters were only tested in commonly-used signals, which were low frequency calls. The composition of vessel noise suggests the lower frequency calls would be more ‘affected’ but it is difficult to say without doing a lot of further analysis, as it depends on the call structure. Again, much more targeted studies would be required to fully address this which goes beyond the scope of the current study.

Somewhere a statement of where this work adds to all the previous studies drawn from is needed to clarify to the reader what knowledge is being added, rather than just assimilating from the other works. Similarly, I think a few sign-post statements of aims, findings and conclusions etc. could aid the reader. Highlight that interactions and their strengths are influenced by several factors, one is the presence of other groups within the communication space and the extent of this, but that presence of vessels/stressors generally may be more influential in whether there are interactions or not.

Response: statements added throughout the discussion (see track changes).

Line 54 – little is known, however, about hearing capabilities
Line 62 – remove a comma after levels
Done

Line 64 – remove relative
Done

Line 65 – is there a reference here
No

Line 77-79 – make sure this reads smoothly
Done

Line 99 – would this not be equal to or exceeding 0 dB
changed

Line 136-148- maybe a few more details about the array could help the reader picture it, and the spacings between each of the hydrophones or perhaps a figure? Also a description of the propagating area might also be helpful – flat sandy bottom, rocky etc.

Response: Added in more details as per comment above

Line 163- I am wondering whether more of your observations could come through in the study rather than just the binary results – for example did the whales move to maintain closer proximities in higher ambient noise, different responses in group travel directions, was the response different with calves present in groups etc.

Response: this refers more to a behavioural response (change in movement behaviour) which was beyond the scope of the study. Here, the study was focussed specifically on communication space, the extent of this space, and their likelihood of socially interacting, rather than a change in movement behaviour. Movement behaviour analyses has been carried out in other studies (now mentioned in discussion), and their results summarised. To do this sort of analyses in this study would not be possible given we do not have detailed tracks of the movement of groups around vessels.

Line 182- italicise a
Done

Line 184 –italicise b
Done

Line 190-192 – italicise formulae were needed
Done

Line 186 – is there a threshold, and was it first audible or visible as a signal on the spectrogram

Line 220-221- make sure this makes sense
Done

Line 245 – plotted or displayed, don't think you need both
Done

Line 271 – and high frequency signals?
No, just low frequency signals

Lines 328-331 –is this from the observation data – please state
Yes, stated

Line 338 – does not read smoothly
Changed

Line 342-343 – is this taken from the wind-dominated observations
Yes, added

Line 356 – were there any changes in call usage or call structure in noise. Were low frequency signals favoured if masking potential is lower?

See response above

Line 343 – 344 – does not read smoothly

Changed

Line 358 – I think it should still be clear that these results come from the model and not the observations

Done

Line 361,384 – 502km should be 50km²

done

Figure 2 – is this modeled or real data?

Modelled – changed in fig. title

Reviewer: 1

Comments to the Author(s)

The revision of this paper addressed a majority of the concerns I had previously. I have noted just a few editorial items and 1-2 comments in my attached pdf with notations. This is a nicely done and well times paper. I would have selected Accept As Is, but I think addressing the few minor points will put a polish on the final version.

Response: all items addressed

Appendix E

**The effects of vessel noise on the communication network of humpback whales**

Rebecca A Dunlop

*Cetacean Ecology and Acoustics Laboratory, School of Veterinary Science, University of*

*Queensland, Gatton Campus, Queensland, QLD 4343, Australia*

Electronic mail: *r.dunlop@uq.edu.au*

Short title: Masking of humpback whale signals

Humpback whales likely rely on ~~soeial sounds~~acoustic communication to mediate social
interactions. The distance to which these social signals propagate from the signaller ~~will~~
defines its communication space, and therefore communication network (number of
receivers). As humpback whales migrate along populated coastlines, they are likely to
encounter noise from vessel traffic which will –mask their social signals. Since no empirical
data exists on baleen whale hearing, the consequences of this are usually assumed, being the
modelled reduction in their communication space. Here, the communication space and
network of migrating humpback whales was compared in increasing wind-dominated and
vessel noise. Behavioural data on their social interactions were then used to inform these
models. In typical wind noise, a signaller’s communication space was estimated to extend to
4 km, which agreed with the ~~observed extent of their~~maximum separation distance between
groups that social interactions socially interacted. An increase in vessel noise reduced the
modelled communication area by a factor of four, along with a significant reduction in group
social interactions, implying a reduction in their communication network. However, signal
masking did not fully explain this change in social behaviour, implying there was also an
additional effect of the physical presence of the vessel on signaller and receiver behaviour.
Though these observed changes in communication space and social behaviour were likely to
be short-term and localised, an increase in vessel activity due to tourism and coastal
population growth may cause more sustained changes along the humpback whale migration
paths.

**Introduction**

Vocal communication networks involve a signaller, one or more intended receivers, and one
or more ‘unintended receivers’ ~~Unintended receivers, referred to as the ‘audience’, as~~
~~‘bystanders’, or as such as~~ ‘eavesdroppers’ (McGregor, 1993; McGregor et al., 2001; Otter et
al., 1999; Peake et al., 2001), ~~can gather and use information from this signalling interaction~~
~~to their advantage~~. The size of this communication network is usually defined as the distance
radius from the signaller at which any other receiver can detect and interpret the signal. This
depends on the signal level and structure, the propagation environment, levels of background
noise, and the receiver’s ability to detect, recognise and discriminate the signal within noise
(Wiley and Richards, 1978). In the marine environment, as sound propagates over large
distances, cetacean communication networks can be extensive and involve multiple intended
and unintended receivers. Cetaceans are likely to utilise these networks within feeding (e.g.
Janik, 2000; Miller, 2006) and breeding interactions (e.g. Dunlop, 2016a; Dunlop and Noad,
2016), meaning effective communication exchanges are essential for these life functions.

In the underwater environment there are several natural sources of noise, such as
noise from surface breaking waves during periods of high wind. Noise sources can also be
anthropogenic in origin, such as from vessel activity, oil and gas exploration, naval sonar
activity, or construction. These noise sources have the potential to mask signals, meaning
they become inaudible to receivers ~~within the network, as the noise levels go up~~. This will
reduce the size of the communication network and, in turn, may have detrimental effects on
feeding or breeding interactions within these networks. Important data for predicting the
potential impacts of masking include, for the receiver, basic audiograms, critical ratios (CRs)
of hearing, and identification of mechanisms that allow receivers some release from masking
(Erbe et al., 2016). ~~Little is known, however, hearing capabilities in baleen whales therefore~~
~~it is difficult to predict how receiving whales cope with increased noise (Erbe et al., 2016).~~

For the signaller, important data for predicting masking includes ~~which~~ determining which
anti-masking strategies are used and how successful these strategies are (Erbe et al., 2016).
Signalling whales, ~~for example, use strategies such as~~ increasing ~~increase~~ their signal level
(Dunlop et al., 2014; Parks et al., 2011), ~~changing~~ change their signalling behaviour (Dunlop
et al., 2010), ~~and changing the~~ frequency content (Heiler et al., 2016; Lesage et al., 1999;
Parks et al., 2007), and duration of their signal (Miller et al., 2000). It is likely, however, that
there is a limit to the success of these strategies, especially i. If the noise source is
anthropogenic in origin, ~~for example,~~ and noise levels are above natural noise levels, ~~;~~
~~signaller anti-masking strategies may not be as effective~~. Therefore, in anthropogenic noise,
the receiver's ability to detect and interpret ~~relative~~ sounds may be reduced over and above
that in natural noise, with a higher risk of loss of communication space and reduction in
communication network.

Acoustic communication in humpback whales (*Megaptera novaeangliae*) has been
extensively studied making this a good model species for cetacean masking studies. The
communication space of social sounds ('vocalisations', blow-hole associated sounds, and
percussive signals generated at the surface) is in the order of km's (Dunlop, 2018) in a
shallow-water environment (<50 m) and includes a network of likely intended and
unintended receivers (Dunlop, 2016a; Dunlop and Noad, 2016). Their repertoire of social
vocalisations is diverse and, spans a fundamental frequency range of less than 40 Hz to ~~to~~
over 2 kHz. S and includes sound types ranging range from low-frequency "grumbles", to
high-frequency bird-like "chirps" ~~as well as~~ and includes both modulated (e.g. "moans",
"trumpets" and "cries"), amplitude-modulated (e.g. "purrs" and "growls"), and broadband
(e.g. "roars" and "underwater blows") sounds (Dunlop et al., 2007). These social sounds are
commonly heard during breeding, (Dunlop, 2016a; Dunlop and Noad, 2016; Rekdahl et al.,
2015; Silber, 1986) and feeding; group social interactions (D'Vincent et al., 1985; Parks et

al., 2014; Sharpe et al., 1998; Thompson et al., 1986). ~~They~~ and likely function to provide
information on group membership, changes in group membership (social information), as
well as signaller sex, size, location (Dunlop et al., 2008), and motivation (Dunlop, 2017).
Previous studies found that the communication space of humpback whale signallers extends
to approximately 4 km (Dunlop, 2018). ~~and that s~~ Signallers maintain this space in increased
wind-dominated noise by ~~both~~ switching from vocal sounds to surface-generated sounds
(Dunlop et al., 2010) and increasing their vocal source level (Dunlop et al., 2014).
Interestingly, in response to increased vessel-dominated noise, there was no evidence that the
signaller used either of these anti-masking strategies (Dunlop, 2016b). This implies that when
vessels are traversing the area, the communication space of a humpback whale signaller is
significantly reduced. The consequences of this within the context of breeding interactions, if
any, remain unknown.

In baleen whales, apart from anatomical modelling, there is no empirical data on their
hearing in noise. Therefore, if predicting the impact of masking on the receiver, the signal
propagation is usually modelled in various noise conditions and the distance at which the
signal disappears below noise is assumed to equate to the receiver's audible limits. In other
words, ~~an~~ assumptions of receiver hearing in noise, and/or the receivers's ability to hear (e.g.
Tennessen and Parks, 2016) or recognise ~~(e.g. Cholewiak et al., 2018; Hatch et al., 2016)~~ and
~~discriminate the signal~~ the signal must be made. ~~Cetacean communication space is therefore~~
~~usually modelled using a presumed limit in receiver audibility (e.g. Tennessen and Parks,~~
~~2016), and in some cases, a presumed ability of the receiver to recognise the signal (e.g.~~
~~Cholewiak et al., 2018; Hatch et al., 2016).~~ To do this, ~~t~~ These studies assume the distance at
which the measured received level of a signal-to-noise ratio, or signal excess, ~~becomes~~ is of a
certain value (e.g. 0 dB) delimits the signaller's communication distance. This study will
follow the approach outlined above to define the communication space of signalling

humpback whales but will use behavioural observations of social interactions to further
inform results. ~~Therefore, in this study, e~~Here, communication space is defined as distance

[revised manuscript text omitted]

~~signalling groups were most likely to interact with their nearest neighbour, the position of this~~
~~receiving group within the signaller's communication space was the most important to~~
~~note~~ This was mainly due to a reduction in the probability that a signalling group would join
with its nearest neighbour in the presence of a vessel (Fig. 2b and c).

One explanation for this reduction in interaction behaviour would be increased signal
masking, ~~as received noise would be likely be elevated above typical wind noise if a vessel~~
~~was in the area~~. However, noise level was not a significant predictor in that increased noise

did not explain any reduction in the likelihood of groups to interact. Even in vessel noise,
many received signals would have been above noise at this close-by separation distance (Fig.
1) suggesting masking was not a significant issue. If using 105 *re* 1 μPa as a generic
(average) received vessel noise level, though the modelled communication space of both
vocalisation types was significantly reduced, it was still beyond 1 km (Fig. 1). Using signals
at which the $rSNR = 0$ does not include any measure of variance, in that, signals received by a
whale at 1 km are likely to range in $rSNRs$. If choosing all signals emitted from groups 1 km
positioned at 1 km from the receiver, the $rSNR$ of low-frequency signals, in received vessel
noise of 105 *re* 1 μPa , was up to + 13 dB (Fig. 3a), and for high-frequency signals, up to + 12
368 dB (Fig. 3b). In other words, many of the signals at the receiver would have been well above
369 noise. Further, tThe modelled decrease in the $rSNR$ with increasing measured noise was also
significantly less in vessel noise compared to wind-dominated noise (Table 1; Fig. 3b). In
other words, low-frequency signals were less masked by vessel noise compared to wind noise
as long as the signaller was 1 km from the receiver. Therefore, at close group-to-group
separation distances, the observed reduction in interaction behaviour in vessel noise cannot
be explained by modelled signal masking.

**Communication area and group distribution**

~~When compared to communication area in wind-dominated noise, there was a 75% loss in~~
~~vessel noise was 75% for both signal types (Table 1). To determine the loss of available~~
~~receiver groups within the communication area in vessel noise, first Assuming a random~~
distribution of groups within that area (4 groups per 50²km²) was assumed using a simple
spherical communication area with the signalling group at the centre. In vessel noise, this
equated to an approximate 77 to 78% reduction in available receivers within the network
(Table 1) for both signal types. However, the actual distribution of groups at this study site

suggests a bias towards the signalling group, where the density of groups (available receivers)
was higher closer to the signalling group (Fig. 1). Using this known distribution, the
reduction in available receivers due to loss in communication space was estimated as 47%
and 33% for the low and high-frequency signal communication area respectively (Table 1).
Behavioural data ~~of observed social interactions from this study suggests the reduction in~~
~~low-frequency signal communication space was unlikely to lead to a significant reduction in~~
~~social interactions, given most of these interactions occurred within 2 km of the signalling~~
~~group. It was also clear that the probability of a receiver group interacting with a signaller~~
~~group was dependent on its position and distance from the signaller (Fig. 2b and c). found~~
~~that, in~~ wind noise, the signalling group had a 70% chance of interacting with the nearest
neighbour if the separation distance was less than 2 km, reducing by 50% in vessel noise.
Therefore, this reduction in interaction behaviour was greater than that predicted by using
known group distribution as a proxy for the likelihood of social interactions, but less than
using a random distribution of groups within the communication space ~~assuming a simplistic~~
~~spherical communication area.~~

Table 1. The modelled communication area based on the distance from the signaller at which
the $rSNR = 0$. The group numbers within the wind and vessel noise communication areas
were estimated assuming a random distribution (of 4 groups per 50²km) and known
distribution, using visual data of groups around signalling groups. Standard deviations are in
brackets.

Signal type	Distance km	Area ² km	Random distribution		Known distribution	
			No. groups	% reduction	No. groups	% reduction
Low-f	4	50	4		2.2 (1.0)	
	2	50	0.9 (0.9)	77 (22)	1.0 (0.8)	47 (36)
High-f	2	12	0.9 (0.8)		1.0 (0.8)	
	1	12	0.3 (0.5)	78 (35)	0.5 (0.6)	33 (44)

Discussion

This goal of this study was to acoustically and behaviourally define the
communication space and network of signalling humpback whales and determine if, and
how, this was reduced during periods of vessel noise. Here, modelled communication space
was defined as distance that conspecifics could potentially send and receive acoustic signals
(assuming a signal excess of 0 dB delimited this distance). However the significance of this
study lies in the use of, and where there was an observable (social) responses (from
behavioural data) in the receiving whale groups to further inform these acoustic models. First
the communication space and network was defined in wind-dominated (natural) noise.

Acoustic rResults suggest that humpback whale low-frequency social signals had a modelled
communication space of approximately 4 km with a communication network of two to four
other groups. BBehavioural data found that humpback whale receivers responded to, and
eventually joined, a signalling group from a maximum separation distance of 4 km suggesting
.Therefore, these low-frequency sounds may function to mediate longer-range interactions
between networks of dispersed groups. However, the local neighbourhood of signalling
humpback whales, where sighted behavioural interactions were most likely to occur, was
within 2 km, which was the modelled more similar to the communication space of higher
frequency signals (2 km). Higher frequency signals were of a lower amplitude compared to
low-frequency signals meaning they may be used mediate closer-range joining interactions
between a signaling and receiving groups. There was usually only one other group within this
network; the group participating in the joining interaction. By matching the predictions of
communication space from the acoustic data, with observed social behaviour, fewer
assumptions on receiver hearing in noise are required. This provides a much more accurate
tool with which to fully assess the consequences of increased anthropogenic noise.

To test for effects of anthropogenic noise, the likely extent of a signalling whale's
communication space was compared in natural (wind) and anthropogenic (vessel) noise. In
average vessel noise (105 dB *re* 1 μ Pa), the communication space of humpback whale low-
frequency vocalisations reduced, by half, to 2 km. These results are similar to modelled loss
in communication space of humpback whale social sounds in vessel noise in Cholewiak (et
al., 2018). The communication space of high-frequency sounds was also reduced by half in
vessel noise to only 1 km. This translates to a reduction of four times the communication
area, assuming a circular area from the signaller for both signal types. This should, in theory,
reduce the number of available receivers and therefore the number of observed social
interactions. To assess this, many studies do not have access to data on receiver group
distribution and social behaviour within the communication area. Rather, a random
distribution of receiver groups around the signaller is assumed. Here, assuming a random
distribution, the consequences of increased vessel noise on the communication network, the
magnitude of the reduction depended on the method use. If assuming a random distribution of
groups around the signaller, the number of available receivers within the network reduced
reduced from four to one. However, there are two problems with this assumption. First,
observational data found that the receiving groups were not randomly distributed from the
signaller, and second, behavioural data found that signallers were more highly likely to
interact with the closest receiver as long as it was within 2 km. After accounting for these two
problems, the reduction in available receivers within the communication space decreased to
47% and 33% for low and high-frequency signals respectively. Behavioural
observations Observed social interactions in vessel noise, however, found a 50% reduction in
the likelihood of a signalling group interacting with a receiver group within the 2 km
interaction zone. Therefore, using the random distribution of receivers over-estimated the
likely consequences of the reduction in communication network, whereas accounting for

observed receiver distribution, but not receiver behaviour, under-estimated network
consequences. This discrepancy, as explained below, is likely to be due to However, as
explained below, there was an additional factor to consider, being the physical presence of
the vessel.

Though this study focused on the acoustic environment and behaviour of receivers,
potential signaller anti-masking strategies were also considered. The estimated
communication space of low-frequency signals in wind-dominated noise was maintained at
approximately 4 km up to received levels of 100 dB *re* 1 μ Pa. This was likely due to the
Lombard response found humpback whales, where signallers increased their vocal source
level in response to increased wind-dominated noise (Dunlop et al., 2014). In the same
received levels of vessel noise, the communication space was found to be smaller. A previous
study found no signaller Lombard response to increasing vessel-dominated noise (Dunlop,
2016b). Vessel noise levels were estimated at the signalling group (in Dunlop, 2016b) and
were comparable to those measured at the array in this study. In other words, received levels
of vessel noise should have been high enough to cause a Lombard response in signalling
whales but did not. One explanation put forward in this study was that whales have release
from masking, in that they can focus their attention on the vocalising conspecific, rather than
the vessel noise. This implies the receiver groups, if possessing the ability for masking relief,
would continue to behave as if there was no masking noise source in the area. Behavioural
results presented here showed this was not the case. Further, the reduction in humpback
group interaction behaviour in vessel noise was not fully explained by signal masking.
Behavioural interactions were most likely to occur when groups were separated by 1 km or
less where most signals, even high-frequency signals, were still well above received vessel
noise and unlikely to be masked. Again, this shows the value of using behavioural data to
inform acoustic models.

~~An alternative explanation for the~~ ~~Therefore both the~~ lack of signaller Lombard
response ~~put forward in~~ ~~found in a previous study~~ (Dunlop, (2016b) ~~was that humpback~~
~~whales were emitting signals at a relatively low level to avoid detection, similar to how they~~
~~behave around unintended receivers~~ (Dunlop, 2016a). ~~In other words, the lack of Lombard~~
~~response was an acoustic avoidance response to the presence of the vessel.~~ ~~and the~~
~~significant change in receiver behaviour found here, are more likely to be an avoidance~~
~~response to the vessel itself.~~ The physical presence of the vessel has been shown to reduce
bottlenose dolphin (Bas et al., 2017) and humpback whale (Blair et al., 2016) foraging
activity as well as bottlenose dolphin resting and socialising behaviour (Bas et al., 2017).
~~Other studies, carried out on this population of humpback whales at this study site, found that~~
~~humpback whale groups significantly changed their migration speed and direction to avoid a~~
~~slowly (4 knots) moving vessel towing (silent) air gun arrays~~ (Dunlop et al. 2015; 2017).
~~Therefore, this body of work suggests that not only do migrating humpback whales~~
~~physically avoid vessels, but also change their acoustic and social behaviour when vessels are~~
~~present. Masking may not be a significant issue when groups are within close proximity to~~
~~each other. Rather, migrating groups physically and acoustically avoid the vessel and, whilst~~
~~doing so, are less likely to socially interact with each other.~~

The population used for this study is robust in that it is increasing at 11% per year
(Noad et al., 2019). These behavioural interruptions within the communication network, if
localised and rare, are unlikely to have a significant effect on such a robust population. As
many of these social interactions are likely to be breeding interactions (Dunlop et al., 2008;
Dunlop 2016a, Dunlop and Noad, 2016), understanding the population consequences of
disturbance, in this case reduced potential breeding opportunities, would be the next step in
understanding whale and vessel interactions. Given this population is also well studied in
terms of population dynamics, a population consequences model would not be difficult to

produce once the link between reduced group interactions within the communication network
and reduced mating opportunities has been established. These models could be extended to
other humpback populations and other baleen whale species, such as the highly endangered
right whale (*Eubalaena spp.*), where less is known about their breeding behaviour. For now,
the most comprehensive model to estimate loss of communication space for baleen whales in
vessel noise would be to account for signal type, receiver group distribution with some
knowledge of the local signalling neighbourhood communication network, and any additional
(e.g. avoidance) effect of the presence of the vessel.

Along the eastern coast of Australia, and likely other populated coastlines, vessel
activity is increasing due to the growth in the tourism industry. For cetaceans occupying a
coastal habitat, vessel interactions are becoming more of an issue. Current Australian
mitigation measures regarding the interaction of vessels and whales include caution zones
(within 300 m of the whale) and exclusion zones (within 100 m). The assumption with these
measures is that staying more than 300 m away “reduces the risk of disturbance to natural
behaviours”. The results of this study suggests otherwise. What is clear is that the deleterious
effects of vessels on whales are not limited to collisions and increased signal masking but
potential changes in vital function behaviours, such as breeding interactions, should also be
considered. In summary, mitigating vessel activity from a noise and communication space
perspective may not mitigate for behavioural changes within their communication network.

Ethics and permits

The project operated under animal ethics approval from the University of Queensland Animal
Ethics Committee, a Queensland Government Environment, and Heritage Protection permit
and an Australian Government Access to Biological Resources in a Commonwealth Area for
Non-Commercial Purposes permit.

Data accessibility

The article's supporting data are available on the Dryad Digital Repository (Dunlop, 2019):

<https://datadryad.org/review?doi=doi:10.5061/dryad.65j00m3>). Wind and vessel noise

communication models are uploaded as part of the supplementary material.

Competing interests

There are no competing interests in that funding agencies had no input in to experimental

design, data collection, data analysis or interpretation or the writing of the paper.

Author contributions

RD conceived the study, designed the study, carried out the data analysis and wrote the

manuscript. Raw data were collected as part of a previous collaborative project (HARC, see

acknowledgements).

**Acknowledgements**

The author would like to thank everyone involved in the Humpback Acoustic Research

Collaboration (HARC; funded by the U.S. Office of Naval Research and the Australian

Antarctic Division), in particular the numerous volunteers who donated their time and

energy to this project. I also thank David Paton for his invaluable field expertise and Eric

Kniest for his continued support in the development of *Cyclopes*. The author would

particularly like to acknowledge Associate Professor Michael Noad for leading the HARC

work (without which, this study would not have been possible) and Dr. Douglas Cato for

his continued support and mentorship.

- Bas AA, Christiansen F, Ozturk B, *et al.* 2017. Marine vessels alter the behaviour of
bottlenose dolphins *Tursiops truncatus* in the Istanbul Strait, Turkey. *Endanger Species Res*
**34**:1-14.
- Blair HB, Merchant ND, Friedlaender AS, Wiley DN, Parks SE. 2016. Evidence for ship
noise impacts on humpback whale foraging behaviour. *Biol Lett* **12**(8).
- Cato, DH. 1996. Ambient sea noise in waters near Australia. *J. Acoust. Soc. Am* **60**:320–328.
- Cholewiak D, Clark CW, Ponirakis D, *et al.* 2018. Communicating amidst the noise:
modeling the aggregate influence of ambient and vessel noise on baleen whale
communication space in a national marine sanctuary. *Endanger Species Res* **36**:59-75.
- D'Vincent CG, Nilson RM, Hanna, RE (1985). Vocalization and coordinated feeding
behaviour of the humpback whale in southeastern Alaska. *Sci Rep Whales Res Inst, Tokoyo,*
**36**:41–47.
- Dunlop R. 2019. Data from: Mitigations for the effects of vessels on baleen whale signaling
from a communication space perspective may not prevent behavioural changes. Dryad Digital
Repository. <http://doi.org/10.5061/dryad.65j00m3>.
- Dunlop RA. 2016a. Changes in vocal parameters with social context in humpback whales:
considering the effect of bystanders. *Behav Ecol Sociobiol* **70**(6):857-870.
- Dunlop RA. 2016b. The effect of vessel noise on humpback whale, *Megaptera*
*novaeangliae*, communication behaviour. *Anim Behav* **111**:13-21.
- Dunlop RA (2017) Potential motivational information encoded within humpback whale non-
song vocal sounds. *J Acoust Soc Amer* **141**(3):2204-2213.
- Dunlop RA. 2018. The communication space of humpback whale social sounds in wind-
dominated noise. *J Acoust Soc Amer* **144**(2):540-551.
- Dunlop RA. The communication space of humpback whale social sounds in vessel noise.
*Proceedings of Meetings on Acoustics*, In press, accepted 20/11/18.
- Dunlop RA, Cato DH, Noad MJ. 2008. Non-song acoustic communication in migrating
humpback whales (*Megaptera novaeangliae*). *Mar Mamm Sci* **24**(3):613-629.
- Dunlop RA, Cato DH, Noad MJ. 2010. Your attention please: increasing ambient noise
levels elicits a change in communication behaviour in humpback whales (*Megaptera*
*novaeangliae*). *Proc Roy Soc B* **277**(1693):2521-2529.
- Dunlop RA, Cato DH, Noad MJ. 2014. Evidence of a Lombard response in migrating
humpback whales (*Megaptera novaeangliae*). *J Acoust Soc Amer*, **136**(1):430-437.
- Dunlop RA, Noad MJ. 2016. The "risky" business of singing: tactical use of song during
joining by male humpback whales. *Behav Ecol Sociobiol* **70**(12):2149-2160.

- Dunlop, RA., Noad, MJ, Cato, DH, Stokes, D. 2007. The social vocalization repertoire of
east Australian migrating humpback whales (*Megaptera novaeangliae*). *J Acoust Soc Amer*
**122**(5): 2893-2905.
- Dunlop RA, Noad MJ, McCauley R, et al. 2015. The Behavioural Response of Humpback
Whales (*Megaptera novaeangliae*) to a 20 cubic inch Air Gun. *Aquat Mamm* **41**(4): 412 –
433.
- Dunlop RA, Noad MJ, McCauley RD, et al. 2017. The behavioural response of migrating
humpback whales to a full seismic airgun array. *Proc Roy Soc B* **284**(1689): 20171901. doi:
10.1098/rspb.2017.1901.
- Erbe C, Reichmuth C, Cunningham K, et al. 2016. Communication masking in marine
mammals: A review and research strategy. *Mar Poll Bull* **103**(1-2):15-38.
- Girola, E., Noad, MJ, Dunlop, RA, Cato DH. 2019. Source levels of humpback whales
decrease with frequency suggesting an air-filled resonator is used in sound production. *J*
*Acoust Soc Amer* **145**(2):869-880.
- Hatch LT, Clark CW, Van Parijs SM, et al. 2012. Quantifying Loss of Acoustic
Communication Space for Right Whales in and around a U.S. National Marine
Sanctuary. *Con Biol* **26**(6):983-994.
- Heiler J, Elwen SH, Kriesell HJ, Gridley T (2016) Changes in bottlenose dolphin whistle
parameters related to vessel presence, surface behaviour and group composition. *Anim Behav*
**117**:167-177.
- Janik VM. 2000. Source levels and the estimated active space of bottlenose dolphin
(*Tursiops truncatus*) whistles in the Moray Firth, Scotland. *J Comp Physiol A* **186**(7-8):673-
680.
- Lesage V, Barrette C, Kingsley MCS, Sjare B (1999) The effect of vessel noise on the vocal
behavior of Belugas in the St. Lawrence River estuary, Canada. *Mar Mamm Sci* **15**(1):65-84.
- McGregor PK (1993) Signaling in territorial systems – a context for individual
identification, ranging and eavesdropping. *Philosophical Transactions of the Royal Society*
*of London Series B-Biological Sciences* **340**(1292):237-244.
- McGregor PK, Peake TM, Lampe HM (2001) Fighting fish *Betta splendens* extract relative
information from apparent interactions: what happens when what you see is not what you
get. *Anim Behav* **62**:1059-1065.
- Miller PJO. 2006. Diversity in sound pressure levels and estimated active space of resident
killer whale vocalizations. *J Comp Physiol A* **192**(5):449-459.
- Miller PJO, Biassoni N, Samuels A, Tyack P L (2000) Whale songs lengthen in response to
sonar. *Nat* **405**(6789):903-903.
- Noad MJ, Cato DH, Stokes MD (2004) Acoustic tracking of humpback whales: measuring
interactions with the acoustic environment. Proceedings of Acoustics 2004, Annual

- Conference of the Australian Acoustical Society, Gold Coast, 3-5 November 2004, pp 353–
358.
- Noad MJ, Dunlop RA, Kniest E. Boom to bust? Implications for the continued rapid growth
of the eastern Australian humpback whale population despite recovery. *Pop Ecol* **61**(2):198-
209.
- Otter K, McGregor PK, Terry AMR, Burford FRL, Peake, TM, Dabelsteen T (1999) Do
female great tits (*Parus major*) assess males by eavesdropping? A field study using
interactive song playback. *Proc Roy Soc B* **266**(1426):1305-1309.
- Peake TM, Terry AMR, McGregor PK, Dabelsteen T (2001) Male great tits eavesdrop on
simulated male-to-male vocal interactions. *Proc Roy Soc B* **268**(1472):1183-1187.
- Parks SE, Clark CW, Tyack PL (2007) Short- and long-term changes in right whale calling
behavior: The potential effects of noise on acoustic communication. *J Acoust Soc Amer*
**122**(6):3725-3731.
- Parks SE, Cusano DA, Stimpert AK, Weinrich MT, Friedlaender AS, Wiley DN (2014)
Evidence for acoustic communication among bottom foraging humpback whales. *Sci Rep* **4**.
- Parks SE, Johnson M, Nowacek D, *et al.* 2011. Individual right whales call louder in
increased environmental noise. *Biol Letters* **7**(1):33-35.
- Rekdahl ML, Dunlop RA, Goldizen AW, Garland EC, Biassoni N, Miller P, Noad MJ
(2015) Non-song social call bouts of migrating humpback whales. *J Acoust Soc Amer*
**137**(6):3042-3053.
- Sharpe FA, Dill LM, Beaver V, Spellman B. (1998) Killing me softly: feeding calls of the
Alaskan humpback whale. Abstracts of the World Marine Mammal Science Conference,
Monaco, January, pp 20 – 24.
- Silber GK (1986) The relationship of social vocalizations to surface behavior and aggression
in the Hawaiian humpback whale (*Megaptera novaeangliae*). *Can J Zoo* **64**(10):2075-2080.
- Tennessen JB and Parks SE. 2016. Acoustic propagation modeling indicates vocal
compensation in noise improves communication range for North Atlantic right
whales. *Endanger Species Res* **30**:225-237.
- Tervo OM, Christoffersen MF, Simon M, *et al.* 2012. High Source Levels and Small Active
Space of High-Pitched Song in Bowhead Whales (*Balaena mysticetus*). *Plos One* **7**(12).
- Thompson POW, Cummings WC, Ha SJ (1986) Sounds, source levels and associated
behavior of humpback whales, Southeast Alaska. *J Acoust Soc Amer* **80**(3):735 - 740.
- Wiley RH, Richards DG. 1978. Physical constraints on acoustic communication in
atmosphere – implications for evolution of animal vocalizations. *Behav Ecol Sociobiol*
**3**(1):69-94.

Fig. 1. The relationship of the communication space of humpback whale low- (a) and high-
(b) frequency signals in wind-dominated and vessel noise. Smoothing functions are from the
generalised estimate equation outputs (Table S1) for the $rSNR = 0$ relationship (from Figs. S1
and S2). Raw data-points at which the signal $rSNR$ at the array was between 1 and -1 are also
included.

Fig. 2. The modelled probability of the signalling group interacting with a neighbouring
group as a function of the distance of the neighbour from the signaller. Relationships are
shown for the wind and vessel noise environments and include 95% confidence intervals (a)
and separated according to the neighbour position relative to the signalling group (b and c in
wind and vessel noise respectively).

Fig. 3. The modelled relationship, and significance of the predictor variables, of the $rSNR$ of
humpback whale low- (a) and high- (b) frequency signals in wind and vessel-dominated noise
received noise. Data-points from signalling whales between 900 and 1100 m (approximately
1 km) from the array are included.
